# Plant diversity dynamics over space and time in a warming Arctic

Mariana García Criado[1 ✉], Isla H. Myers-Smith[1,2], Anne D. Bjorkman[3,4], Sarah C. Elmendorf[5], Signe Normand[6], Peter Aastrup[7,8], Rien Aerts[9], Juha M. Alatalo[10], Lander Baeten[11], Robert G. Björk[4,12], Mats P. Björkman[3,4], Noémie Boulanger-Lapointe[13], Ethan E. Butler[14], Elisabeth J. Cooper[15], J. Hans C. Cornelissen[9], Gergana N. Daskalova[16], Belen Fadrique[17,18], Bruce C. Forbes[19], Greg H. R. Henry[20], Robert D. Hollister[21], Toke Thomas Høye[8,22], Ida Bomholt Dyrholm Jacobsen[23], Annika K. Jägerbrand[24], Ingibjörg S. Jónsdóttir[25], Elina Kaarlejärvi[26], Olga Khitun[3], Kari Klanderud[27], Tiina H. M. Kolari[28,29], Simone I. Lang[30], Nicolas Lecomte[31], Jonathan Lenoir[32], Petr Macek[33,34], Julie Messier[35], Anders Michelsen[36], Ulf Molau[3,53], Robert Muscarella[37], Marie-Louise Nielsen[8,22], Matteo Petit Bon[30,38], Eric Post[39], Katrine Raundrup[23], Riikka Rinnan[36], Christian Rixen[40,41], Ingvild Ryde[36], Josep M. Serra-Diaz[42,43], Gabriela Schaepman-Strub[44], Niels M. Schmidt[7,8], Franziska Schrodt[45], Sofie Sjögersten[46], Manuel J. Steinbauer[47,48], Lærke Stewart[49], Beate Strandberg[22], Anne Tolvanen[50], Craig E. Tweedie[51] & Mark Vellend[52]

The Arctic is warming four times faster than the global average[1] and plant communities are responding through shifts in species abundance, composition and distribution[2–4]. However, the direction and magnitude of local changes in plant diversity in the Arctic have not been quantified. Using a compilation of 42,234 records of 490 vascular plant species from 2,174 plots across the Arctic, here we quantified temporal changes in species richness and composition through repeat surveys between 1981 and 2022. We also identified the geographical, climatic and biotic drivers behind these changes. We found greater species richness at lower latitudes and warmer sites, but no indication that, on average, species richness had changed directionally over time. However, species turnover was widespread, with 59% of plots gaining and/or losing species. Proportions of species gains and losses were greater where temperatures had increased the most. Shrub expansion, particularly of erect shrubs, was associated with greater species losses and decreasing species richness. Despite changes in plant composition, Arctic plant communities did not become more similar to each other, suggesting no biotic homogenization so far. Overall, Arctic plant communities changed in richness and composition in different directions, with temperature and plant–plant interactions emerging as the main drivers of change. Our findings demonstrate how climate and biotic drivers can act in concert to alter plant composition, which could precede future biodiversity changes that are likely to affect ecosystem function, wildlife habitats and the livelihoods of Arctic peoples[5,6].

Climate change is altering biodiversity patterns on Earth[7,8]. Elevated rates of species extinctions have led to biodiversity loss at the global scale[9,10]. At regional scales, biotic homogenization has been observed[11,12], whereas at local scales, studies have shown increased species turnover, but often no net changes in species richness[13,14]. The effects of climate change on biodiversity have been observed across temperate and tropical biomes[8,15]. However, little is known about changes in species diversity at northern latitudes, despite Arctic ecosystems experiencing four times faster warming than the global average[1]. Although rapid warming is expected to alter Arctic vascular plant communities, the direction of local diversity changes remains uncertain[2,16], particularly because local changes in species richness do not necessarily translate into large-scale biodiversity trends[17]. Plants

are the foundation of Arctic terrestrial food webs, the carbon cycle and the livelihoods of Arctic people. Therefore, to understand the effects of climate change on Arctic ecosystems, we must first quantify how climate change is affecting terrestrial plant communities.

Changes in Arctic plant diversity could be shaped by interacting processes following four pathways. (1) If species migrate northward to track climate warming, we would expect a net increase in overall Arctic plant species richness[2,18]. (2) Richness increases could also result from short-distance dispersal and colonization events by species that are already present in neighbouring local species pools, as growing conditions improve and communities are potentially able to support more species[19]. (3) Conversely, reduced Arctic floral diversity could result from losses of cold-adapted species[20] that cannot cope with increasing

temperatures[21]. (4) These declines could be exacerbated by increased competition with colonizing species originating from Low Arctic and boreal latitudes[22,23] or by local species becoming better competitors under warmer conditions[4]. Because these pathways may be acting in concert, it is possible, and indeed likely, that richness increases and decreases could occur simultaneously, resulting in no net changes in richness. Yet, the effects of these different pathways on current and future Arctic plant diversity trends remain poorly understood. We address this knowledge gap by quantifying the direction and magnitude of Arctic vascular plant diversity change over time at the local level (α diversity) and temporal turnover in species composition (β diversity). We also investigate which geographical, climatic and biotic drivers are related to different aspects of diversity change to understand trends across the Arctic.

Apart from evolutionary history and biogeography, species richness patterns at large scales are broadly driven by climatic gradients[24]. Many taxa have a latitudinal gradient in diversity, in which species richness is greater at lower latitudes, which are generally warmer[25,26]. Therefore, Arctic vascular plant richness is expected to increase over time as rapid warming[1,27] leads to new, warmer thermal niches for warm-adapted species at northern latitudes. This expectation is further supported by observed increases in vascular plant species richness with warming across European mountain tops[28,29], where elevational gradients mirror Arctic latitudinal climatic and richness gradients[30]. Spatially, we would expect plant richness to increase at warmer, lower Arctic latitudes because of the potential influx from the species-rich boreal forest (borealization)[31,32] and because the dissimilarity between Low Arctic and boreal flora is more pronounced than the dissimilarity between High and Low Arctic flora[33]. Overall, we expect richness increases where more warming has occurred and at lower latitudes closer to the boreal zone.

Warming-driven shifts in biotic interactions are another key driver of changes in species distributions and community composition[3]. Changes in dominance of different functional groups (for example, graminoids, forbs and shrubs) can affect the plant diversity and abundance of the entire plant community[5]. For instance, shrub expansion has been associated with decreases in lichen, bryophyte and bare ground cover[2,16]. Traits such as higher and denser canopies allow tall shrubs to outcompete shorter species for light by shading[3,34] and deciduousness that results in greater litter fall can smother shorter plants[35,36]. An increase in nitrogen-fixing tall shrubs (for example, alder) may also lead to increased soil nitrogen and result in suppression and competitive exclusion of non-nitrogen-fixing vegetation[22,37]. Tundra species with high light and specific nutrient requirements, or those specialized in cold environments, might be particularly vulnerable to changing competitive interactions, with rare species at greater risk of local extinction[38], as has been observed in Arctic-alpine ecosystems[20]. Overall, a decline in species richness may be expected in areas where shrub cover has increased over time.

Shifts in species composition owing to warming are likely to lead to temporal changes in the spatial dissimilarity (that is, spatial β-diversity changes over time) of plant communities across the Arctic. Climate change might lead to ecological communities experiencing biotic homogenization, as observed in other biomes, such as tropical[39] and temperate forests[12]. Arctic vegetation might become spatially more homogeneous (that is, lower β diversity) owing to the expansion of dominant and widespread species[40], such as dwarf shrubs across the High Arctic, as a result of reduced winter mortality and increased recruitment with warming[41,42]. At the forest–tundra ecotone, shrub expansion could lead to biotic homogenization as shrubs become more dominant[43]. However, habitat heterogenization could also occur[40]. For example, permafrost thaw and hydrology changes with warming could lead to the development of new wetland plant communities[44]. Moreover, the borealization of Arctic ecosystems close to the treeline could further differentiate Low and High Arctic plant communities[45]. In summary, whether Arctic plant communities will become more or less similar to each other with climate change remains uncertain.

Here we quantify multiple dimensions of local Arctic vascular plant diversity: richness, richness change, evenness (Pielou), evenness change, temporal turnover on the basis of presence–absence and abundance (Jaccard and Bray–Curtis) and species trajectories (proportions of species gains, losses and persistence) over time (Supplementary Tables 1–3). We also evaluate changes in subsite-level composition over time using principal coordinate analyses (PCoAs). We used 42,234 records from 2,174 plots in 45 study areas (Fig. 1a), encompassing 490 vascular plant species (Extended Data Figs. 1 and 2). First, we quantified spatial patterns in Arctic diversity across latitudinal and climatic gradients, to inform our expectations of diversity changes in response to warming. Second, we identified the specific geographical (latitude and biogeographical region), climatic (moisture, warmest quarter temperature, precipitation and their change over time), biotic (functional group cover and its change over time) and sampling variables (plot size, plot-level species richness and monitoring duration) associated with Arctic diversity change. Third, we investigated whether vascular plant communities across the Arctic are becoming more similar (that is, declining β diversity) over time. Our monitoring dataset from the International Tundra Experiment Plus (ITEX+) database consists of marked plots with plant species composition surveyed at different intervals between 1981 and 2022 (Fig. 1d and Extended Data Fig. 3). ITEX+ sites have a hierarchical structure, with species composition data recorded at the plot level. There are multiple plots in a subsite and often multiple subsites in a study area (Extended Data Fig. 1b–d). The 45 long-term monitoring study areas capture most of the variation in temperature and precipitation across the Arctic tundra (Fig. 1b and Extended Data Fig. 4) and represent diverse assemblages of tundra functional groups (Fig. 1c and Extended Data Fig. 5a–c). We address three main research questions.

First, we investigated how and why the Arctic vascular plant richness has changed over time. We hypothesize that there has been an overall increase in plot-level richness (α diversity) over recent decades across the Arctic[18]. We expect greater richness increases in warmer sites and at lower latitudes, which are closer to boreal forest species pools[31], paralleling the latitudinal biodiversity gradient[46]. Despite the presence of some shade-tolerant species, we also hypothesize that plant species richness will decline where shrub cover increases over time, because sun-loving plants could be outcompeted by shading and increased litter production from taller and denser shrub canopies, as per spatial analyses[22]. Therefore, tundra plant communities close to the treeline could follow different trajectories in shrub-dominated versus open tundra plant communities.

Second, we investigated how and why the temporal plant species turnover has changed. We hypothesize that there is an increase in plot-level turnover and species replacement with warming[37] and increasing shrub cover[2]. We expect proportionally greater species gains with warming as a result of increases in thermophilous species[47]. Where shrubs are increasing in dominance, we expect greater species losses owing to shading and litter production[22].

Third, we investigated whether vascular plant communities across the Arctic are becoming more similar in composition over time. Despite uncertainty, we hypothesize that there is biotic homogenization of plant communities (declining spatial β diversity through time)[42]. This homogenization could be caused by an infilling of warmer thermal niches[32,41,42] by the same increasingly dominant, high-occupancy species with higher growth rates, good dispersal and colonization capacities[47]. These plants will outweigh proportional gains of low-occupancy species.

## Richness patterns and trends over time

We found support for the extension of the latitudinal species richness gradient across the Arctic, with higher spatial plot-level richness at lower latitudes (slope = −0.03 log[species] per °C, corresponding to

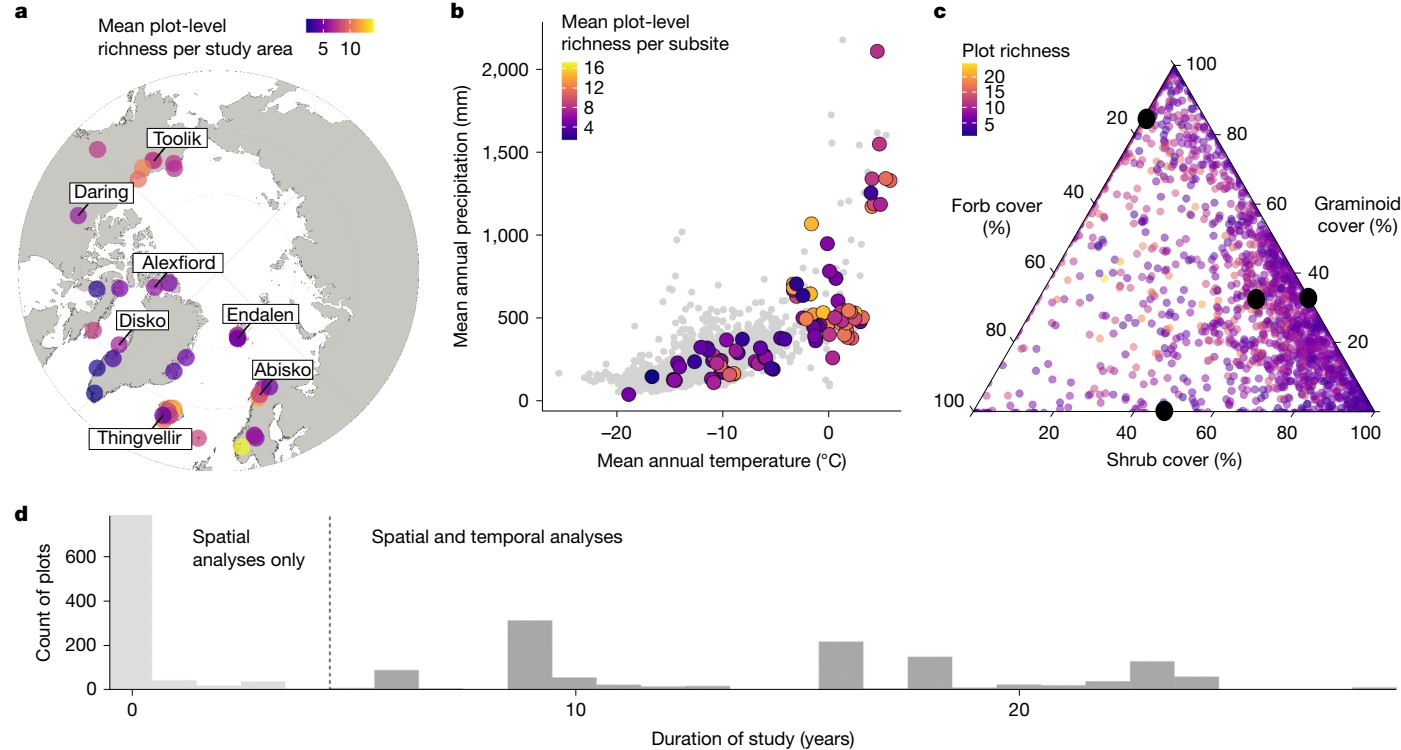

**Fig. 1 | Representation of our dataset in geographical, climatic and biotic space and its temporal resolution. a**, Distribution of study areas, coloured according to mean plot-level plant richness per study area (*n* = 45). This mean calculation is for visualization purposes only, with all analyses and estimates presented elsewhere using individual plot-level richness, unless stated otherwise. A few study areas are labelled for reference. Polar projection with a southern limit of 57° N. Map created in R with the ggOceanMapsData[69] package v.1.4, which uses base layers from Natural Earth (https://www.naturalearthdata.com/). **b**, Subsites included in this study as a function of their climatic space, coloured according to their mean plot-level richness (*n* = 115). Background grey points represent a selection of 1,189 randomly extracted geographical coordinates from the Circumpolar Arctic Vegetation Map[33]. Subsites included in our study cover an extensive gradient of Arctic climatic conditions (Extended Data Fig. 4).

**c**, Relationship between mean cover (calculated as average cover over the entire monitoring period) of the different functional groups per plot (*n* = 2,174). Species-rich plots had greater forb cover, whereas greater graminoid cover was associated with species-poor plots. Cover of all three functional groups were negatively correlated. Points represent plots and are coloured according to mean plot richness. Black points indicate mean plot cover for each functional group on each axis and the black point inside the ternary plot indicates the mean cover overall. **d**, Duration of monitoring for all plots in our dataset (*n* = 2,174). Only plots that were monitored for more than 5 years (in dark grey) were included in temporal analyses (*n* = 1,266 plots), while those monitored shorter than 5 years (in light grey) were included only in the spatial analyses (*n* = 908 plots). The dotted line indicates the 5-year duration boundary. For a survey timeline, see Extended Data Fig. 3.

a decrease of around one species per every 5° increase at mid-range Arctic latitudes, 97.5% credible interval (CI) = −0.05 to −0.01, conditional $R^2$ = 0.67, marginal $R^2$ = 0.1; Fig. 1a, Extended Data Fig. 6 and Supplementary Table 3, model 1). Richness was also greater at warmer sites, with approximately one species gained on average for every 2 °C increase in warmest quarter temperature (slope = 0.06 log[species] per °C, 97.5% CI = 0.03–0.1, conditional $R^2$ = 0.63, marginal $R^2$ = 0.16; Supplementary Table 3, model 2) and in plots with greater forb cover and lower graminoid cover (Fig. 1c and Supplementary Tables 2 and 3, models 4 and 5).

Despite greater plant richness at lower latitudes and warmer sites, Arctic plant richness did not change directionally over time, on average (slope = 0.0021 log[species] per year, 95% CI = −0.0002 to 0.0043, equating to 0.01 species gained per year, conditional $R^2$ = 0.63, marginal $R^2$ = 0.003; Fig. 2b,c and Supplementary Table 1). Species richness change was not related to latitude (Fig. 2a and Supplementary Table 3, model 51) or to long-term warming trends (Fig. 2d and Supplementary Table 4). There was no interactive effect between temperature and temperature change on richness change (slope = 0.07, 95% CI = −0.65 to 0.78, conditional $R^2$ = 0.13, marginal $R^2$ = 0.03). Declines in richness occurred with increasing shrub cover and particularly where erect shrubs, but not dwarf shrubs, increased over time (conditional $R^2$ = 0.16 and marginal $R^2$ = 0.05 for model without shrub categories; conditional $R^2$ = 0.08 and marginal $R^2$ = 0.007 for model with shrub

categories; Fig. 2e and Supplementary Table 3, models 52 and 52b). Richness change was not dependent on initial shrub cover (Extended Data Fig. 7a and Supplementary Table 5). Richness increased over time with increasing forb cover (conditional $R^2$ = 0.18, marginal $R^2$ = 0.07; Fig. 2f and Supplementary Table 3, model 53). The effects of shrub and forb change on richness change remained even when extreme values of change were removed from analyses (Extended Data Fig. 7b–d). Spatial richness and evenness were correlated (Supplementary Table 2 and Supplementary Discussion). Overall, plots that were more diverse and/or had more evenly distributed species abundance experienced fewer plot-level species gains and losses as a proportion of total species richness (Extended Data Fig. 8a–f).

## Changes in species composition

Nearly all (99%) of the plots experienced changes in species composition owing to altered relative abundances (Bray–Curtis) and 59% of plots showed compositional changes owing to species gains and losses (Jaccard; Fig. 3a–c). Arctic communities experienced a mean temporal turnover of 0.22 (Jaccard) and 0.36 (Bray–Curtis; data bounded between 0 and 1), representing presence–absence (Jaccard) and both presence–absence and abundance-related turnover at the plot level (hereafter referred to as abundance-related turnover). Greater presence–absence temporal turnover (Jaccard) occurred in colder and

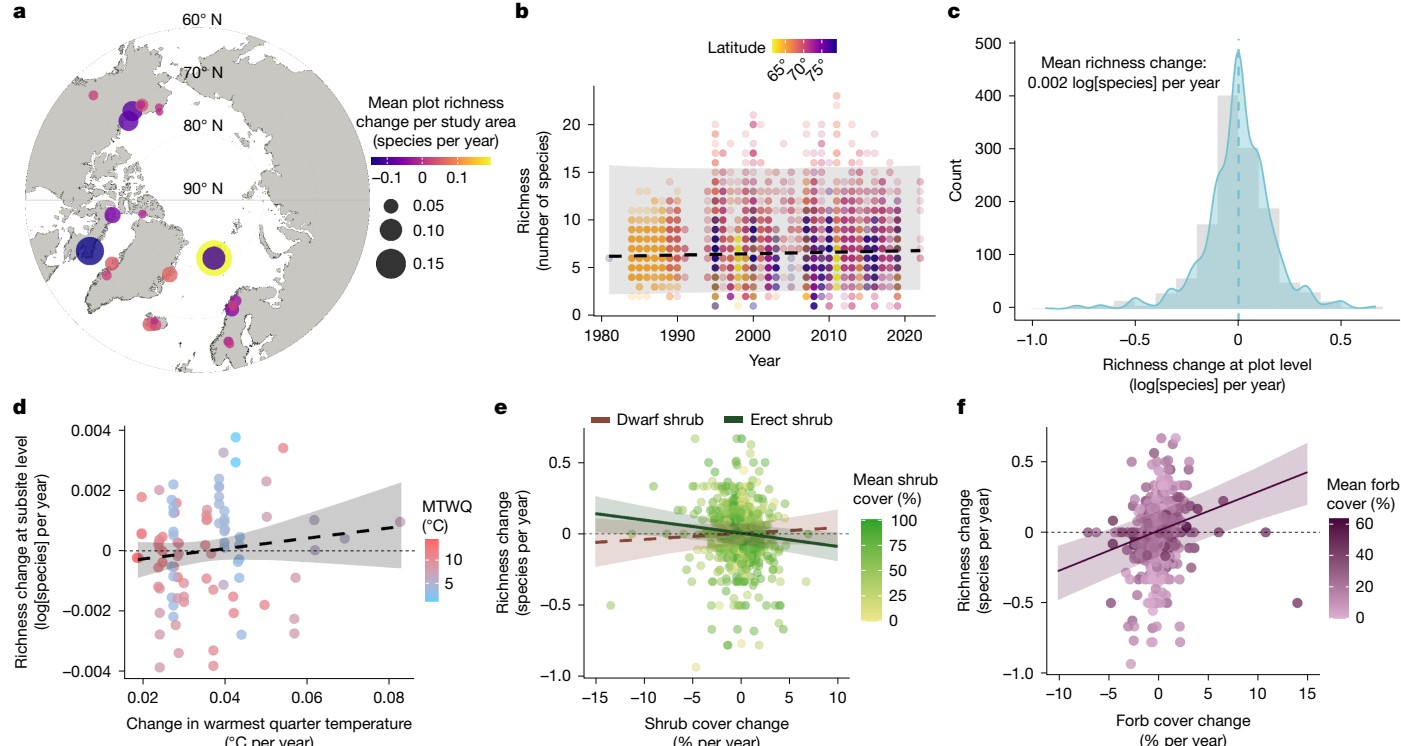

**Fig. 2 | There was no directional change in Arctic species richness on average. a**, There was no clear relationship between species richness change and latitude (Supplementary Table 3, model 51). Richness change values were calculated as the slope estimate of the linear models of richness change over time per plot and then averaged to the study area level (*n* = 25) for visualization purposes. Points are coloured and sized according to their richness change value. Polar projection map created in R with the ggOceanMapsData[69] package v.1.4, which uses base layers from Natural Earth (https://www.naturalearthdata.com/). **b**, Richness did not change directionally over time. Points represent richness per plot and per year, coloured according to latitude. The dashed line and grey band represent the output from the high-level model in Supplementary Table 1. **c**, Mean richness change (*n* = 1,266 plots) as the slope of richness over time per plot. The dashed blue line represents mean richness change. Histogram bin width is 0.1. Model structure and output are from the high-level model in Supplementary Table 1. **d**, Richness did not increase at subsites with stronger long-term warming trends. Points represent richness change as slope subsite-level estimates (*n* = 90), extracted from the high-level model in Supplementary Table 1 and coloured according to climatology. MTWQ, mean temperature of the warmest quarter. **e**, Richness decreased where erect shrubs (but not dwarf shrubs) increased over time (Supplementary Table 3, models 52 and 52b). Points are coloured according to mean shrub cover. **f**, Richness increased where forbs increased over time (Supplementary Table 3, model 53). Points are coloured according to mean forb cover per plot. Richness change estimates per plot in **e** and **f** are extracted from the richness-over-time linear model. Dashed lines indicate a model in which the CIs on the slope overlapped with zero, solid lines indicate CIs that did not overlap with zero and bands show the 95% CIs of the models.

wetter sites, regions with stronger warming trends and species-poor plots (Fig. 3a,b and Supplementary Table 3, models 12–18). Conversely, greater abundance-related temporal turnover (Bray–Curtis) occurred in warmer sites, regions with weaker warming trends, species-rich plots (Fig. 3a,b and Supplementary Tables 3 (models 19–26) and 4) and plots monitored over longer periods of time (Extended Data Fig. 8h). Shrub cover change was not directly related to turnover (Fig. 3c). Plots experienced substantially more species persisting over time (mean = 5.49 species per plot, 64%) than species gained (1.84, 19%) or lost (1.67, 17%) as a proportion of the plot-level species trajectories (Extended Data Fig. 5d). Proportions of species gained, persisting and lost were similar across functional groups and to the overall dataset composition (*P* > 0.05 for all groups in two-proportion *z* test; Extended Data Fig. 5e–h and see Supplementary Table 6 for top species per trajectory). Species that were more frequently lost across plots were generally rarer (that is, were found at fewer study areas, slope = −0.13, 95% CI = −0.17 to −0.09, conditional and marginal *R*² = 0.18).

## Drivers of species gains and losses

Species persistence was positively related to mean summer temperature, with colder sites experiencing proportionally more gains and losses than warmer sites (Fig. 3d and Supplementary Table 3, models 28, 36 and 44). Stronger warming trends were associated

with lower proportions of plot-level species persistence and higher proportions of losses and gains over time (Fig. 3e and Supplementary Table 3, models 32–34, 40–42 and 48–50). There were proportionally more species losses in plots where shrubs had increased (Fig. 3f; this relationship also held up when removing the most extreme values of change) and graminoids had decreased and proportionally more species gained in plots where forbs had increased (Supplementary Table 3, models 40–42 and 49). There were proportionally fewer species gains in plots where shrubs had increased, but the effect was not significant (Fig. 3f and Supplementary Table 3, model 48). See Supplementary Discussion for the effects of geographical and sampling design variables, additional turnover and evenness results, overall functional group composition and climate change context.

Warming (Fig. 3b,e) and shrubification (Figs. 2e and 3f) emerged as two main drivers of Arctic plant diversity change. We therefore conducted additional analyses to better understand how and where these drivers interact (Supplementary Table 7). Overall, shrub cover did not increase significantly over time in our dataset (Supplementary Tables 8 and 9). Shrub cover change was not associated with latitude (Extended Data Fig. 9a) and the rate of long-term warming was not related to the rate of shrub cover change over time (Extended Data Fig. 9c). However, interannual variation in shrub cover sensitivity to temperature was different between shrub categories, indicating that dwarf shrubs

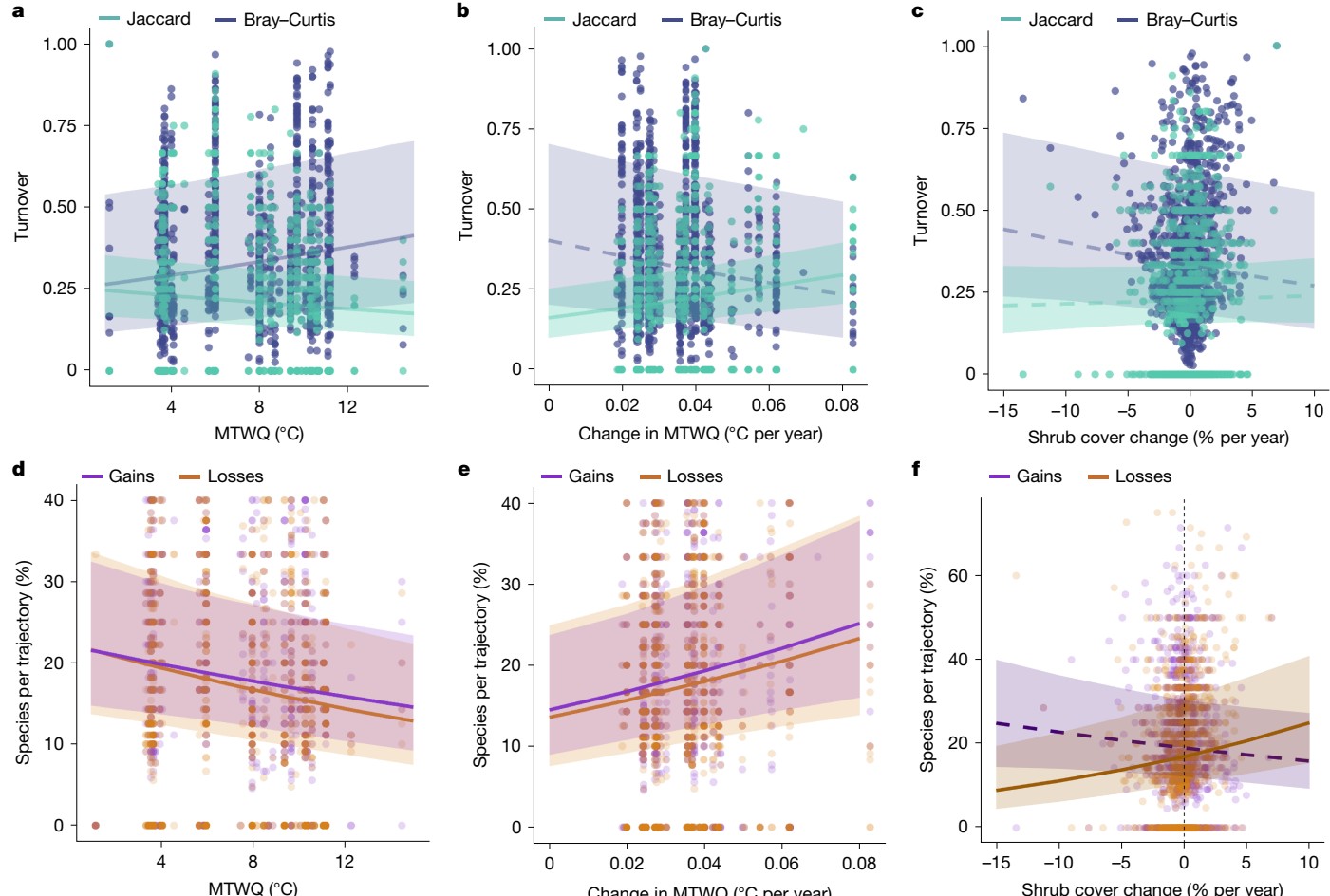

**Fig. 3 | Local climate, climate change and shrubification influenced temporal turnover and species trajectories. a**, Relationships between MTWQ and two temporal turnover metrics: Jaccard (presence–absence turnover) and Bray–Curtis (presence–absence and abundance turnover). Model outputs are in Supplementary Table 3, models 12 and 20; note that the significance of the Bray–Curtis models differed between the univariate and multivariate models (Supplementary Table 4). **b**, Relationships between temperature change over time (slopes from linear models) and the two turnover metrics ($n = 1,266$). Model outputs are in Supplementary Table 3, models 16–18 and 24–26; note that the significance of the Bray–Curtis models differed between the univariate and multivariate models (Supplementary Table 4). The univariate model is presented here for visualization purposes. Nearly half of the plots (526 plots, 41.5%) did not change in terms of presence–absence turnover (Jaccard) whereas only six (0.4%) plots did not change when considering both presence–absence and abundance turnover (Bray–Curtis); these are indicated by a turnover value of 0 in **a**–**c**. **c**, Turnover metrics were not directly associated with shrub cover change over time (Supplementary Table 3, models 16 and 21). **d**, Relationships between MTWQ and the proportion of species lost or gained for each trajectory. Model outputs are in Supplementary Table 3, models 36 and 44. **e**, Relationships between MTWQ and the proportion of species lost and gained. Model outputs are in Supplementary Table 3, models 40–42 and 48–50. **f**, Increases in shrub cover over time were associated with decreased species gains (although this effect was not significant) and increased species losses (Supplementary Tables 2, 3 (models 40 and 48) and 4). Lines and bands represent predicted model fits and the 95% CIs, respectively. Dashed lines indicate CIs that overlapped with zero and solid lines indicate CIs that did not overlap with zero. All analyses are Bayesian hierarchical models.

responded negatively whereas erect shrubs responded positively to warmer temperatures (Extended Data Fig. 9b).

## No indication of Arctic biotic homogenization

Our ordination analyses did not find any signs of Arctic-wide biotic homogenization or differentiation (Fig. 4). Subsites did not become more or less similar to each other over time. Their composition shifted in all possible directions and their location in the ordination space was broadly associated with latitude (Fig. 4a,b). There were similar distances to centroid between start (that is, baseline) and end (that is, final) time points per subsite both for Jaccard (mean ± s.d.; start: 0.66 ± 0.03; end: 0.66 ± 0.03) and Bray–Curtis (start: 0.65 ± 0.04; end: 0.64 ± 0.04) ($P > 0.05$ in analysis of variance (ANOVA) for all β-diversity metrics; Fig. 4c,d and Extended Data Fig. 10). Mean shifts in distance between time points per subsite (as Cartesian coordinates, reflecting change in

community composition relative to starting point) were 0.035 ± 0.03 (Jaccard) and 0.04 ± 0.03 (Bray–Curtis; Fig. 4e).

## Discussion

Contrary to our hypotheses, there is so far no directional trend in plant richness change on average (α diversity; Fig. 2b,c), despite the Arctic experiencing the greatest rates of warming on Earth over the past decades[1] (Extended Data Fig. 4b). This result, based on the local scales, ran counter to literature predictions[18] and experimental observations of plant diversity declines at the landscape scale[48] and modelling studies predicting a regional decline of 15–47% in Arctic-alpine plant species richness[20]. We found that Arctic plant composition and richness changes are decoupled, with no net richness change on average despite widespread changes in composition over time (Figs. 2 and 3). Consistent with our hypotheses, in plots where diversity changes do

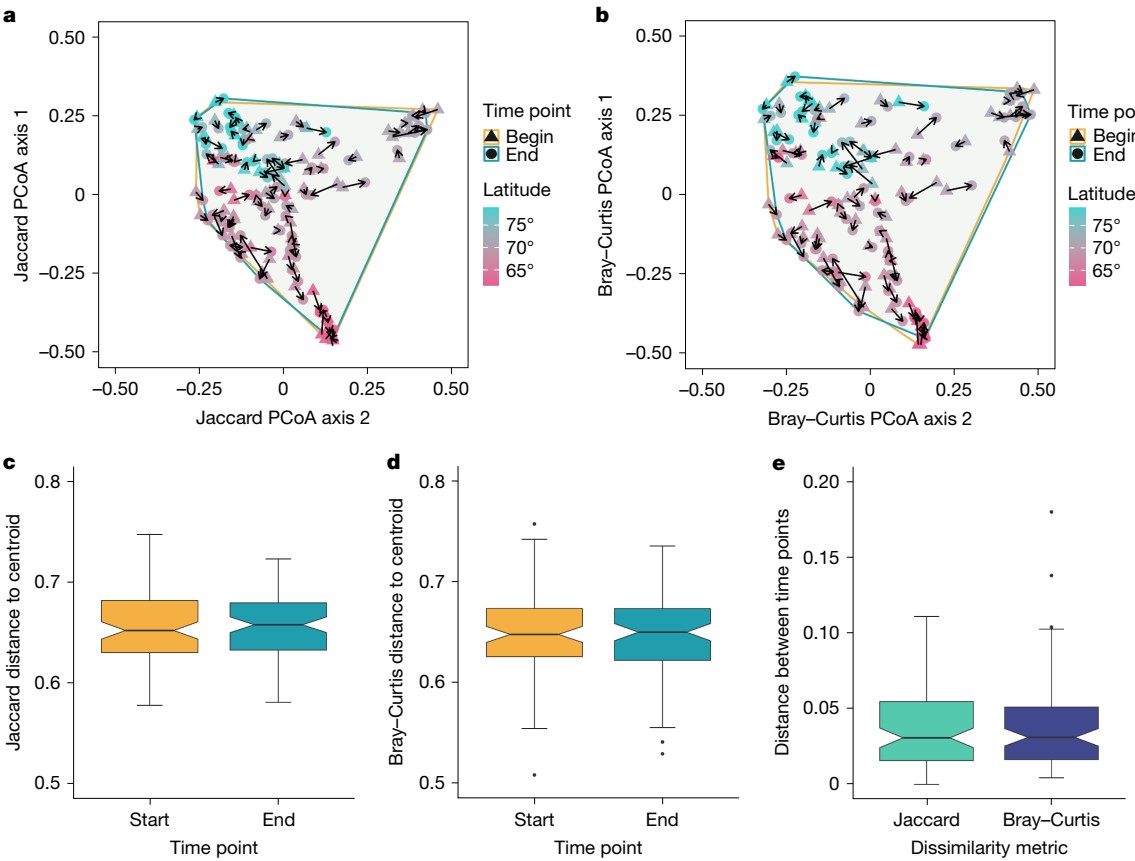

**Fig. 4 | Subsites showed no homogenization or differentiation over time across the Arctic. a,b**, Jaccard and Bray–Curtis β-diversity metrics. We calculated temporal change in spatial turnover (β diversity) between the start (baseline) and end (final) time period for all subsites. PCoAs are shown with the Jaccard (**a**) and Bray–Curtis (**b**) β-diversity metrics. Triangles represent the start time point and circles represent the end time points for all subsites, joined by an arrow for each subsite, indicating the direction of change over time. Points are coloured according to latitude. Enclosing convex hulls are drawn around subsites. **c,d**, Jaccard and Bray–Curtis scores derived from PCoAs. Box plots show the mean distance to centroid for all subsites at the start versus the end for Jaccard (**c**) and Bray–Curtis (**d**) scores derived from PCoAs (*n* = 90 for each time point). **e**, Mean distances in ordination space between time points (start versus end) for all subsites, calculated as Cartesian coordinates (*n* = 90 for each metric). These values show how much plant communities have changed in composition and abundance. Additional β-diversity metrics are presented in Extended Data Fig. 10. In **c**–**e** data are represented as box plots in which the middle line is the median, the lower and upper hinges are the first and third quartiles, the upper whisker extends from the hinge to the largest value within 1.5 × the interquartile range (IQR) from the hinge and the lower whisker extends from the hinge to the lowest value within 1.5 × IQR of the hinge. Data beyond the end of the whiskers are outliers and plotted as points.

occur, these are driven mainly by the combined effects of warming and plant–plant competition, especially increases in erect shrubs[2,22]. Despite the lack of a strong relationship between warming and richness change, both proportional species gains and losses were greater in plots where temperatures increased the most (Figs. 2 and 3). We found a more consistent influence of shrub increases over time, with relatively greater species losses, therefore leading to decreased species richness in plots where shrub cover (particularly of erect shrubs) increased the most over time (Figs. 2 and 3). We did not find evidence of homogenization of Arctic vascular plant communities over time, with no directional temporal changes in spatial dissimilarity of species composition (Fig. 4 and Extended Data Fig. 10), indicating that plant communities changed in their composition in a variety of ways. Overall, we found that Arctic plant community composition changed to different assemblages on the basis of local context and both climate warming and shrubification emerged as key factors influencing the magnitude of species turnover.

## Climate influences on diversity change

Despite spatial species richness being greater at lower latitudes and warmer sites (Extended Data Fig. 6 and Supplementary Table 2) and rapid Arctic warming over time (Extended Data Fig. 4b), species richness did not change directionally (Fig. 2b). Richness change was not greater towards southern Arctic edges (Fig. 2a), where we hypothesized that northward species migration from the boreal forest (that is, borealization) might be a major driver of change. This lack of latitudinal change might indicate that, where diversity is changing, one of the main sources is colonizations by species present in local species pools that have not yet been recorded in long-term monitoring plots (that is, 'landscape' or 'dark' diversity)[49]. Species richness increases were not greater at sites with greater rates of warming over time (Fig. 2d), but warming was associated with proportionally greater species gains and losses (Fig. 3e). Given the importance of biotic interactions at species' warm edges[50], gains could represent expansions of warm-adapted species, which could outcompete cold-adapted species[47,51]. This could be generating species losses, together with cold-adapted species being less able to cope physiologically with warming. This suggests that plant community composition is being influenced by warming (Fig. 3b), but that species gains and losses in plant communities balance each other on average (Fig. 3e and Extended Data Fig. 5d), therefore resulting in the observed overall non-directional richness change (Fig. 2b). This is consistent with some predictions of equilibrium theory[52]. With 99% of plots experiencing composition changes through altered relative species abundance (Bray–Curtis > 0) and 66% of plots gaining and/ or losing species (Jaccard > 0), composition change could begin to influence richness change over time. Overall, these compositional

changes could result in further species reshuffling owing to altered biotic interactions, potentially leading to losses of rare and ecologically important species and associated changes in ecosystem function.

## Shrubification drove diversity change

We found that shrubification was associated with richness and compositional changes. Increases in shrub cover over time were accompanied by decreases in richness and evenness and greater proportional species losses relative to sites with decreasing shrub cover (Figs. 2e and 3f and Supplementary Table 2). Shrub expansion has been widely reported[2,4], although we found only a marginal increase (that is, the CIs overlapped with zero) in Arctic shrub cover at the plot scale in the ITEX+ dataset (Supplementary Table 9). Shrub cover change has been frequently linked to warming in previous site-level studies[2–4]. However, we did not find clear evidence for greater shrub change with greater rates of warming in this dataset (Extended Data Fig. 9c), in agreement with previous pan-Arctic studies[4]. Instead, we found that shrub cover sensitivity to temperature differed between shrub categories, with erect shrub cover increasing and dwarf shrub cover decreasing with warmer temperatures (Extended Data Fig. 9b).

Across space, lower species richness has been observed with greater shrub cover, with shading and litter production leading to decreases in sun-loving plants under shrub canopies[22,23]. Using space-for-time approaches, studies have assumed a similar pattern to occur over time, without necessarily testing it. Here we found and confirmed this pattern over time: at sites where shrub cover increased over time, community evenness decreased and greater species losses occurred, leading to reduced species richness (Figs. 2e and 3f and Supplementary Table 3). Our Arctic-wide results corroborate site-level reports that increasing shrub cover over time may lead to less diverse plant communities and the displacement of rare and/or less competitive species[37,53]. Therefore, Arctic diversity might be more at risk at sites with increasing shrub cover, particularly from erect shrubs (Fig. 2e). Conversely, both increasing graminoid and forb cover were associated with increased richness over time and increasing graminoid cover was related to lower species losses (Fig. 2f and Supplementary Table 2). Graminoids were more likely to persist than forbs (Supplementary Table 2), perhaps because graminoids are good competitors that can displace shallow-rooted forbs where they both occur, due to their deeper root networks, faster nutrient uptake, greater height and better resistance to herbivory[54,55]. Overall, our findings suggest that species may be more at risk in areas where taller shrubs are expected to increase due to aboveground competition for light[55,56].

## Multi-directional plant diversity change

Our findings demonstrate that Arctic plant richness changed in different directions (Fig. 2b,c) amid continued warming. We found that, on average, plots had a majority of species persisting over time (64%; Extended Data Fig. 5d). Plots with high species richness and more even communities showed the least amount of change, with a lower proportion of species losses and gains (Extended Data Fig. 8a–f). This pattern could be a statistical artefact owing to smaller species pool sizes leading to proportionally greater gains and losses or be a result of greater community resistance owing to the reduced extinction risk derived from greater richness and lack of species dominance[38], as per the diversity–stability relationship[57]. We found that persistence was more common in locally warm and dry environments relative to colder and wetter environments, whereas there were proportionally more species losses at cold sites relative to warm sites (Fig. 3c). Homogenization has been predicted for High Arctic vegetation[41,42], but we found no evidence of either biotic homogenization or differentiation (calculated as temporal changes in spatial dissimilarity) in Arctic plant communities so far, in line with global syntheses[40], with no particular directionality of subsite-level change (Fig. 4). Our findings support the observed global decoupling of compositional and richness change[13,14], as we observed more temporal turnover than directional Arctic richness change. One consequence of temporal turnover is the increase in tundra plant community height over time owing to the immigration of taller species[56]. Continued compositional changes are likely to lead to additional shifts in plant traits and the functioning of Arctic ecosystems[5,56].

A better understanding of the underlying mechanisms that drive local biodiversity changes will be key to identifying future rates and hotspots of change under accelerating warming[58]. Further research is required to determine whether Arctic plant communities are showing resistance to warming[59], as additional processes could contribute to a lack of detected richness change on average. For example, the same species could be both lost and gained across plots over time owing to stochastic dynamics or sampling effects (Supplementary Table 6). Future changes in species richness and composition may not yet be detected owing to extinction lags[60] and slow colonization rates in communities of long-lived perennial species. Furthermore, priority effects could cause heterogeneity in species responses to warming[61]. Variation in topography, microclimate and nutrient availability could mediate ecological responses and buffer against climate change effects by providing microhabitats with suitable conditions[19,62,63]. Rising temperatures are projected to be accompanied by increasing precipitation, leading to a warmer and wetter Arctic. This could ameliorate warming-derived drought effects on plants[44]. Moreover, herbivory may mitigate warming-driven shrub expansion in certain regions[48]. Therefore, the integration of extinction lags, priority effects, local context and both microclimate and macroclimate is an essential next step to better identify the mechanisms behind Arctic plant dynamics.

It was not possible to include non-vascular plants (bryophytes and lichens) in our analyses owing to inconsistent recording across plots[64], but their influence on vascular plant dynamics cannot be discounted. Bryophytes can suppress vascular plant regeneration[65], whereas both lichens and mosses have a strong buffering effect on microclimate extremes, and can therefore mitigate further shrubification[66]. Therefore, plots that were initially more dominated by non-vascular plants might be more resistant to vascular plant colonizations, which could explain temporal lags in richness change. Furthermore, the presence or absence of certain bryophytes reflect subtle differences in changing surface hydrology (for example, drying and paludification), soil chemistry and disturbance[67,68], which can in turn affect species composition. A future priority will be to expand non-vascular plant surveys to obtain a comprehensive view of plant biodiversity changes and biotic interactions among functional groups.

Overall, we found that changes in Arctic plant diversity and community composition depend on the local context, with both warming and shrubification emerging as key factors that influence the magnitude of species turnover. Probable mechanisms underlying the observed diversity changes include colonization from local species pools[49], gain of thermophilous species[47], loss of less competitive and/or rare species[51] and increased competition with canopy-forming shrubs[22]. Our results indicate that we should not necessarily expect an overall loss or gain of vascular plant biodiversity with warming in the Arctic. Instead, directional changes in plant communities will depend on the combination of changing environmental conditions and available species pools, with warming leading to greater plant community composition changes and shrubification, resulting in decreasing species richness over time. This research demonstrates the value of long-term in situ monitoring at local scales for the detection of biodiversity change and improving our understanding of biome-wide responses or resistance to climate warming. The extensive reshuffling of Arctic vascular plant composition in recent decades observed in this study underscores the urgent need to explore the effects of these shifts on ecosystem function, wildlife habitats and the livelihoods of Arctic peoples[5,6].

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

[1]School of GeoSciences, University of Edinburgh, Edinburgh, UK. [2]Department of Forest & Conservation Sciences, Faculty of Forestry, University of British Columbia, Vancouver, British Columbia, Canada. [3]Department of Biological and Environmental Sciences, University of Gothenburg, Gothenburg, Sweden. [4]Gothenburg Global Biodiversity Centre, Gothenburg, Sweden. [5]Institute of Arctic and Alpine Research, University of Colorado Boulder, Boulder, CO, USA. [6]Department of Biology, Aarhus University, Aarhus, Denmark. [7]Department of Ecoscience, Aarhus University, Roskilde, Denmark. [8]Arctic Research Center, Aarhus University, Aarhus, Denmark. [9]Amsterdam Institute for Life and Environment (A-LIFE), Amsterdam, The Netherlands. [10]Environmental Science Center, Qatar University, Doha, Qatar. [11]Forest & Nature Lab, Department of Environment, Ghent University, Melle, Belgium. [12]Department of Earth Sciences, University of Gothenburg, Gothenburg, Sweden. [13]Department of Geography, University of Victoria, Victoria, British Columbia, Canada. [14]Department of Forest Resources, University of Minnesota, St Paul, MN, USA. [15]Department of Arctic and Marine Biology, Faculty of Biosciences, Fisheries and Economics, UiT – The Arctic University of Norway, Tromsø, Norway. [16]Biodiversity, Ecology and Conservation Group, International Institute for Applied Systems Analysis, Laxenburg, Austria. [17]Ecology and Global Change, School of Geography, University of Leeds, Leeds, UK. [18]Department of Geography and Planning, School of Environmental Sciences, University of Liverpool, Liverpool, UK. [19]Arctic Centre, University of Lapland, Rovaniemi, Finland. [20]Department of Geography, University of British Columbia, Vancouver, British Columbia, Canada. [21]Biology Department, Grand Valley State University, Allendale, MI, USA. [22]Department of Ecoscience, Aarhus University, Aarhus, Denmark. [23]Department of Environment and Mineral Resources, Greenland Institute of Natural Resources, Nuuk, Greenland. [24]Department of Electronics, Mathematics and Natural Sciences, Faculty of Engineering and Sustainable Development, University of Gävle, Gävle, Sweden. [25]Life and Environmental Sciences, University of Iceland, Reykjavík, Iceland. [26]Organismal and Evolutionary Biology Research Programme, Faculty of Biological and Environmental Sciences, University of Helsinki, Helsinki, Finland. [27]Faculty of Environmental Sciences and Natural Resource Management, Norwegian University of Life Sciences, Aas, Norway. [28]Department of Environmental and Biological Sciences, University of Eastern Finland, Joensuu, Finland. [29]Centre de recherche sur la dynamique du système Terre (Geotop), Université du Québec à Montréal, Montreal, Quebec, Canada. [30]Department of Arctic Biology, University Centre in Svalbard, Longyearbyen, Norway. [31]Centre d'Études Nordiques, Department of Biology, University of Moncton, Moncton, New Brunswick, Canada. [32]UMR CNRS 7058, Ecologie et Dynamique des Systèmes Anthropisés (EDYSAN), Université de Picardie Jules Verne, Amiens, France. [33]Institute of Hydrobiology, Biology Centre of the Czech Academy of Sciences, Ceske Budejovice, Czech Republic. [34]Chair of Biodiversity and Nature Tourism, Institute of Agricultural and Environmental Sciences, Estonian University of Life Sciences, Tartu, Estonia. [35]Department of Biology, University of Waterloo, Waterloo, Ontario, Canada. [36]Department of Biology, University of Copenhagen, Copenhagen, Denmark. [37]Plant Ecology and Evolution, Evolutionary Biology Center, Uppsala University, Uppsala, Sweden. [38]Department of Wildland Resources, Quinney College of Natural Resources and Ecology Center, Utah State University, Logan, UT, USA. [39]Department of Wildlife, Fish and Conservation Biology, University of California Davis, Davis, CA, USA. [40]WSL Institute for Snow and Avalanche Research SLF, Davos, Switzerland. [41]Climate Change, Extremes and Natural Hazards in Alpine Regions Research Centre (CERC), Davos, Switzerland. [42]Botanical Institute of Barcelona (CSIC-CMCNB), Barcelona, Spain. [43]Université de Lorraine, AgroParisTech, INRAE, Silva, Nancy, France. [44]Department of Evolutionary Biology and Environmental Studies, University of Zurich, Zurich, Switzerland. [45]School of Geography, University of Nottingham, Nottingham, UK. [46]School of Biosciences, University of Nottingham, Loughborough, UK. [47]Bayreuth Center of Sport Science (BaySpo), University of Bayreuth, Bayreuth, Germany. [48]Bayreuth Center of Ecology and Environmental Research (BayCEER), University of Bayreuth, Bayreuth, Germany. [49]Department of Natural Sciences and Environmental Health, University of South-Eastern Norway, Bø, Norway. [50]Natural Resources Institute Finland, Oulu, Finland. [51]Department of Biological Sciences, University of Texas at El Paso, El Paso, TX, USA. [52]Département de Biologie, Université de Sherbrooke, Sherbrooke, Quebec, Canada. [53]Deceased: Ulf Molau. ✉e-mail: mariana.garcia.criado@gmail.com

## Methods

### Plant composition data

We extracted composition and abundance data from the ITEX+ dataset[70]. Our dataset was composed of 42,234 unique records from 2,174 plots in 155 subsites distributed across 45 study areas encompassing 490 vascular plant species, recorded during different intervals over the past four decades (1981–2022) across the Arctic (Extended Data Fig. 3). We kept only control (ambient) plots and did not include experimental data. All ITEX+ sites have a hierarchical structure, with species abundance and composition data recorded at the plot level, multiple plots in a subsite and generally multiple subsites in a study area. Study areas are general regions ranging in size from several hundred square metres up to tens of kilometres. Subsites are smaller regions, or clusters of plots, in larger study areas, either located in different habitat types or created as blocks of plots in study areas. Plots are the smallest spatial units, nested in subsites and study areas. We refer to these terms throughout to indicate specific levels of this hierarchy and we use the terms plant communities or sites when referring more generally to groups of Arctic species at any scale or resolution. Our analyses were carried out with plot as the replication unit, unless specified otherwise.

Our dataset contained 2,174 plots and they were all retained for spatial analyses. For temporal analyses, we retained the 1,266 plots (58.2%) that had been surveyed at least twice over a minimum of 5 years, since shorter timeseries tend to overrepresent real change in Arctic communities[17,56]. The remaining 908 plots (41.7%) were used only in the spatial analyses (Fig. 1d and Extended Data Fig. 3). Of all the plots that were surveyed more than once, 35.3% were surveyed twice, 21.5% were surveyed three times, 19.7% were surveyed four times, 23.3% were surveyed five or more times and 0.5% were surveyed ten or more times (Extended Data Fig. 3).

Plots range in size (that is, surveyed area) on the basis of the plant species community of interest and landscape characteristics[71] (mean plot size = 0.57 m$^2$, range = 0.048–1 m$^2$). There is an average of 48 plots per study area (range = 5–276), 14 plots per subsite (range = 1–87) and three subsites per study area (range = 1–11). The total surveyed area per subsite (calculated as plot size × number of plots per subsite) is generally constrained under 20 m$^2$ (Extended Data Fig. 1b–d). Plots were monitored over different periods during four decades (Fig. 1d and Extended Data Fig. 3), with a mean study duration of 8 years (range = 1–28), a mean of three monitoring time points per plot (range = 1–11) and a mean time between surveys of 5 years (range = 1–26).

For data cleaning (taxonomic verification, input errors), we followed a previously published protocol[56]. Furthermore, we retained only Arctic and sub-Arctic plots in the Northern Hemisphere (>60° latitude). We kept plots that had consistent sampling methods and plot sizes over time. We retained data for only vascular plants (shrubs, graminoids and forbs) because non-vascular plants were not recorded consistently across study areas. We defined biogeographical regions as Eurasia, Greenland–Iceland, eastern North America and western North America according to glaciation history[72–74]. We kept only plots for which the surveyed area was ≤1 m$^2$ to ensure comparable richness values across plots, given that plant species richness tends to increase with plot size[75]. Because Arctic plants are relatively small individuals, a plot size of 1 m$^2$ is appropriate to reflect ecological assembly processes at the local scale[76]. We included the natural log transformation of plot size in all models (except for the evenness model) to account for variability between plot sizes and to most closely resemble species–area relationship theory[75,77]. We did not include the plot size term as a fixed effect in evenness models because the evidence of a relationship between plot size and evenness is mixed, with studies finding positive, negative and no relationships[78]. Therefore, there are no clear theoretical reasons to expect such a relationship. We tested an additional plot-size sensitivity analysis by re-running models behind some of the main outcomes (Supplementary Table 3, models 45 and 52) but only with plots for which the size was 1 m$^2$ ($n$ = 631 and 597 for the main analysis and the sensitivity analysis, respectively). Both estimates of temperature change and shrub cover change had the same direction and significance as their original model counterparts.

Because plots in the ITEX+ dataset were surveyed by different methods, we retained only plots that were surveyed using percentage cover as an abundance metric and/or another metric that was convertible to percentage cover, including point-framing and cover-class methods (for example, Braun–Blanquet). We kept all types of point-framing information (top hit, top–bottom hits and all hits), because values of overall richness were similar across methods (Extended Data Fig. 1a). We compared data with hit order information and found that top, top–bottom and all values were very similar as were point-framing data with and without coordinates (Extended Data Fig. 1). We converted all values to relative cover (0–100%) to ensure consistency between survey methods (Extended Data Fig. 2). See Supplementary Methods for a detailed account of data cleaning and cover conversion.

We calculated functional group proportion in each plot-by-year by adding up the total cover of species in a functional group (shrubs, graminoids and forbs), so that the total vascular plant cover was 100% in each plot-by-year. We also calculated the proportion of functional group cover per plot by averaging the proportion of functional group cover across all years in a plot. We use this metric as an indication of the extent to which a functional group covers a plot and refer to it as 'greater' or 'lower' cover. Finally, we calculated functional group change over time by adding up cover values of all species per functional group and year and fitting linear models of cover over time per plot and per functional group separately. These slopes (mean annual values of functional group change) were used as fixed effects in subsequent models (as shrub percentage change, graminoid percentage change and forb percentage change; Supplementary Table 2). We use this metric to indicate the degree to which functional group cover had changed over time in each plot and refer to it as 'increasing' or 'decreasing' cover over time. When models featured functional group cover or functional group change as covariates (functional-group-composition or plot-change-over-time models, see 'Multivariate models' below and Supplementary Table 2), we fitted three models, each including change in one functional group, to achieve convergence given that functional group proportions were inherently negatively correlated. These three models included all the same covariates except for the functional group in question and are all represented in the same row under functional-group-composition and plot-change-over-time models in Supplementary Table 2.

### Climate data

We extracted, at the subsite level, data from long-term climatologies at CHELSA (v.1.2.1)[79], including mean annual temperature, MTWQ per year, mean temperature of the coldest quarter per year and mean annual precipitation (hereafter precipitation) for the period 1979–2013. After examining correlations between the three temperature variables, we found that they were correlated with each other. Therefore, for our temperature variable, we chose MTWQ (hereafter temperature) as it best represents the growing-season conditions and has previously been linked to plant biomass, growth and reproductive rates[80–82], which are in turn relevant variables driving diversity change. Furthermore, we extracted time series of the daily mean air MTWQ per year and annual precipitation amount during the period 1979–2013.

We calculated change over time in temperature and precipitation by fitting linear models of yearly climatic values over this time period and used the slopes of change per plot as fixed effects in the multivariate models described below (as temperature and precipitation change; Supplementary Table 2). Because geographical coordinates are available only at the subsite level, multiple plots in the same subsite had the same climatic change values. This was accounted for with the inclusion of a random effect for subsite in the models (Supplementary

Table 2). We chose CHELSA as the source of our climate data because, as a quasi-mechanistic statistical downscaling product, it has a very fine grain size ($1 \times 1$ km) and has been shown to outperform other interpolation-based climate products, particularly for precipitation metrics[79,83,84].

### Biodiversity metrics

We chose to analyse common biodiversity metrics that capture species diversity, dominance and composition change, rather than composite indices, to examine the specific elements of biodiversity in isolation from each other. Richness was defined as the total number of species co-occurring in a plot. We acknowledge that some authors refer to this term as 'species density' when it is based on an area metric[85], but hereafter we refer to richness as a more common term used in the literature. We refer to richness change as changes in richness over time, including increases, decreases and no change trends. Temporal turnover was defined as the replacement rate, in terms of species composition, in a focal plot and between the start (baseline survey) and the end (last resurvey) year of the time period covered by the focal plot. We computed the Jaccard (on the basis of presence–absence only) and Bray–Curtis (which considers both presence–absence and abundance change) indices. Both metrics were calculated with the betapart package v.1.5.6 in R[86]. Evenness defines the relative abundance of different species, with high evenness indicating similar abundances of species and low evenness indicating varying abundances. It is based on Pielou's $J$, calculated as $H$/log[$S$], where $H$ is Shannon's diversity index and $S$ the total number of species[77].

We considered species to be locally 'lost' if they were originally surveyed in a plot but were not present in the last resurvey. Similarly, local 'persisting' species are those that were present at both the start and end year of the monitoring period. Species 'gained' are those absent during the baseline survey but occurring in the last resurvey. These species trajectories were originally calculated as counts and then transformed to proportions to account for the inherent variability in species richness across plots. Species proportions were calculated by dividing the number of species per trajectory in a plot by the total number of species in each plot at both time points combined (that is, total number of unique species present at each plot in both time points, including losses, gains and persisting species). This approach allows for an overview of species trajectories per plot and for comparability across plots.

### Statistical analyses

We used a Bayesian framework for all analyses. We used the software and programming language R v.4.1.0[87]. Bayesian models were fitted using the brms package v.2.17[88] and ran for as many iterations as necessary to achieve convergence (2,000–3,000 iterations over four chains), which was assessed through examination of the Rhat term and trace plots.

### Data families

We fitted hierarchical models with different family distributions depending on the structure of the response variable (Supplementary Tables 1 and 2). These included Gaussian family with an identity link function (for continuous response variables with a normal distribution), negative binomial family with a log link function (for count data for which the variance is greater than the mean), beta family with a logit link function (for values ranging between 0 and 1, but excluding 0 and 1), zero-inflated beta family with a logit link function (for values ranging between 0 and 0.99) and zero–one-inflated beta family with a logit link function (for values between 0 and 1, including 0 and 1). For the beta family, we included in our models 'zi - 1' (where zi is the probability of being 0), 'zoi - 1' (where zoi is the probability of being 0 or 1) and 'coi - 1' (where coi is the conditional probability of being 1, given that an observation is 0 or 1). In the case of the spatial richness models (Supplementary Table 2, models 1–5), the log link function with a negative binomial distribution assumes the relationship between richness

and plot size to be log–log: log[richness] ~ log[plot size]. We specified weakly informative priors for beta and negative binomial families. Data families for each model are specified in Supplementary Tables 1 and 2.

### High-level models

To obtain the mean-richness and evenness-change estimates across the tundra, we fitted hierarchical models of richness and evenness per year over time and included nested random slopes per plot in the subsite (Supplementary Table 1). In these two models, the year covariate was centred as needed to achieve model convergence. Plot-level estimates were extracted from the richness change over time model to visualize overall richness change over time (Fig. 2b,c) and subsite-level estimates were extracted to fit the richness-change - temperature-change model (Fig. 2d and Supplementary Table 4).

### Multivariate models

We fitted three main types of multivariate models: spatial, two time point and temporal (Supplementary Table 2). Spatial models refer to current diversity metrics across space, with one unique value of the response variable (richness, evenness) measured at the last surveyed time point. These models identify the main drivers behind spatial patterns of plant diversity. Two time point models consider a response variable that has been derived from two points in time, with a single value providing the measure of change (temporal turnover through Jaccard and Bray–Curtis models and proportions of species lost, gained and persisting). Temporal models reflect metrics in which the response variable had multiple values over time and had a minimum of two time points over 5 years (richness change, evenness change, models derived from the spatial homogenization-over-time analyses). For these temporal models (richness change and evenness change), we followed a two-step modelling approach to examine diversity metrics over time. First, we calculated change over time by fitting linear models of richness and evenness per plot with sampling year as the fixed effect (one linear model per plot); these are referred to as change-over-time models. Then, we extracted the slopes of change over time per plot and used them as a response variable in a second set of models to test the relationships between putative drivers of temporal diversity change, which were measured at the plot or subsite level (SUBS in Supplementary Table 2). Both the two time-point model and temporal model identified the main drivers behind temporal patterns of changes in plant diversity (that is, research questions 1 and 2).

Across all three model types (spatial, two time point and temporal) and for each response variable, we fitted several multivariate models (that is, geographical, climatic, functional group composition, change over time, plot change over time and subsite) depending on the scale at which the covariates affected the response variable, to avoid collinearity and obscuring patterns between fixed effects (Supplementary Tables 2 and 3). We used a hierarchical modelling approach by including subsite as a random effect (a random intercept) to account for the non-independence of plots in subsites. For key results, we also fitted univariate models to understand whether the relationships were consistent with the multivariate model results without the influence of other covariates (Supplementary Table 4).

### Sampling design covariates

All multivariate models (Supplementary Table 2) included a set of relevant sampling design variables to account for different surveying methods ('plot size'), survey timing ('duration') and local context ('mean richness'). We included the natural log transformation of plot size in all models to account for variability between plot sizes and for the fact that different plot sizes may lead to different chances to detect changes over time[17,77]. Mean richness was calculated as the mean values of richness across all years to reflect the most common conditions in a plot over time (Supplementary Table 2). Duration was calculated as the difference between the first and the last years of surveying per plot.

See Supplementary Discussion for an overview of the effects of the sampling design variables on biodiversity metrics.

## Post hoc analyses

To understand the relationship between two of the main drivers of diversity change, shrub cover change and warming over time, we performed extra analyses (Extended Data Fig. 9 and Supplementary Table 7), given that previous literature has suggested a positive relationship between them[4,89]. First, we modelled shrub increases as a function of latitude, with subsite as a random effect (Extended Data Fig. 9a). To identify whether shrubs were sensitive to temperature, we calculated the mean temperature of the past 5 years for each monitoring time point (Extended Data Fig. 9b). We centred temperatures in the subsite before analyses to standardize magnitudes across regions and to enable model convergence. We modelled shrub cover at each time point as a function of mean temperature of the past 5 years, with a nested random effect structure of plot within subsite and an interaction term for shrub type (dwarf versus erect). Furthermore, we modelled shrub cover change per plot as a function of long-term temperature change (over the 1978–2013 period), with a random effect of subsite and an interactive term of shrub type (Extended Data Fig. 9c). To assign shrub categories, we followed a previously published methodology[90] and categorized shrubs as dwarf and erect (including low and tall shrubs), because we were interested in the ecological effects of species with a sprawling versus an erect physiognomy.

## Additional models

A number of models were fitted outside the context of the already described high-level models, multivariate models and post hoc analyses described above. To understand the effects of increasing shrub cover on richness, we modelled richness change as a function of shrub cover change and its interaction with starting shrub cover (Supplementary Table 5). To understand whether species losses were related to rarity, we modelled the proportional losses per species (as a percentage of losses relative to all trajectories across plots) as a function of the number of study areas where the species was present in our dataset. To understand whether our temporal turnover versus richness models reflected a priori relationships or whether there was a meaningful biological relationship, we compared them with null models. To fit null models, we randomly removed 20% of species per plot (to simulate species losses) and randomly included 20% of species (to simulate species gains). We used this simulated dataset to calculate turnover values (Jaccard and Bray–Curtis). We fitted intercept-only null models with each metric and modelled Jaccard and Bray–Curtis turnover as a function of species richness.

Snow is another important driver of tundra plant composition. However, analyses of satellite remote-sensing products providing snow cover variables[91] showed that gridded layers of snow-related variables contained too many spatial and temporal gaps to generate a reliable time series of snow cover duration at our sites. Instead, we extracted data on temporal trends, over the period 1950–2021, for three snow-related variables: snow season length, onset of snow season and end of snow season. These three variables were downloaded from the Bioclimatic atlas of the terrestrial Arctic database (ARCLIM)[92], at a spatial resolution of approximately 9 km by 9 km. We fitted a selection of mixed-effects models to analyse temporal changes for a series of biodiversity variables (richness change, Jaccard turnover, Bray–Curtis turnover, persisters, gains, losses and evenness change) with these three snow-related variables as fixed effects, together with sampling design variables (plot size, duration and mean richness). None of the snow variables was significant in either of these models. This might be owing to a non-significant ecological effect of snow season length on diversity trends or instead be the result of a scale mismatch. The spatial resolution at which diversity metrics were calculated is 1 m² or smaller, whereas the spatial resolution at which snow data are available

is 9 km. Therefore, this scale mismatch precludes us from making any ecological inferences on the effect of temporal trends in snow season length on plant diversity change.

## Ordination analyses

We performed ordination analyses to understand whether community homogenization or differentiation had taken place at the subsite level (that is, research question 3). To assess temporal changes in spatial turnover, we calculated spatial dissimilarity in species composition for all subsites separately at the start time point and at the end time point. To aggregate plot-level data into subsite-level data, we calculated the mean cover per species across all plots in a subsite, both for the start time point and for the end time point. PCoAs were carried out with the vegan[93] v.2.6-2 and ape[94] v.5.6-2 R packages. We calculated multiple β diversity dissimilarity metrics (Jaccard, Sørensen, Bray–Curtis, modified Gower, Manhattan and Euclidian) for both the start and end time point of all 90 subsites (Extended Data Fig. 10). These dissimilarity metrics had varying degrees of emphasis on presence–absence versus abundance turnover[95].

Subsequently, we calculated homogeneity of variance between the mean distance to centroid for start and end subsites, using the methodology outlined previously[96] and assessed the difference in mean distance to centroid between start and end time subsites through ANOVAs. Here, centroids indicate the average community composition across subsites. We then calculated the distance between start and end time points per subsite in the PCoA space for two β-diversity metrics (Jaccard and Bray–Curtis) through Cartesian coordinates (equation (1)), where $x_2$ and $y_2$ refer to the final time point per subsite and $x_1$ and $y_1$ refer to the start time point per subsite. These values reflected the change in community composition and abundance relative to the start time point of each subsite. Next, we modelled the distances between PCoA coordinates as response variables against the set of fixed effects in Supplementary Table 2.

Finally, we calculated the difference in the distance to centroid between start and end time for each subsite and modelled those values as response variables against the set of fixed effects (Supplementary Table 2). These values reflected the difference in each subsite relative to the overall mean composition of subsites across the Arctic. An overall decrease in this distance across all subsites would indicate compositional homogenization.

$$\text{Distance between PCoA coordinates} = \sqrt{(x_2 - x_1)^2 + (y_2 - y_1)^2} \quad (1)$$

## Reporting summary

Further information on research design is available in the Nature Portfolio Reporting Summary linked to this article.

## Data availability

Plant composition data are available at Zenodo (https://doi.org/10.5281/zenodo.14884498)[97]. Climate data from CHELSA can be accessed at https://chelsa-climate.org/ and snow data are available at Figshare (https://doi.org/10.6084/m9.figshare.c.6216368.v2)[98].

## Code availability

The R code used to generate the figures and analyses of this manuscript is available at Zenodo (https://doi.org/10.5281/zenodo.14884498)[97].

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

**Acknowledgements** We dedicate this article to the memory of Ulf Molau, one of the key founders of the International Tundra Experiment. We thank all local and Indigenous communities for the opportunity to work with data collected on their lands, T. Finderup Nielsen for sharing the starting code for the homogenization analysis, J. Kerby for extracting the random selection of Arctic locations, R. Essery for his help with snow data, C. Colesie and V. Vandvik for their comments on the first version of the manuscript, those who contributed to the preparation of the ITEX+ database, including Z. Panchen, A. Becker-Scarpitta, J. Prevéy, J. Von Oppen, K. Johansson, J. J. Henn and J. Everest. M.G.C. was funded by the British–Spanish Society Award and the Principal's Career Development Scholarship from the University of Edinburgh. M.G.C., I.H.M.-S., S.N., B.C.F., C.R. and G.S.-S. were funded by the EU Horizon 2020 Research and Innovations Programme through the CHARTER project (grant number 869471). M.G.C. and I.H.M.-S. were funded by the NERC Tundra Time project (NE/W006448/1). I.H.M.-S. was funded by the ERC Synergy project RESILIENCE (GA 101071417) and Canada Excellence Research Chairs Program. A.D.B. was funded by the Knut and Alice Wallenberg Foundation (WAF KAW 2019.0202), the Swedish Foundation for Strategic Research (FFL21-0194) and the Swedish Research Council (2019-05264). S.N. and A.D.B. were funded by The Danish Council for Independent Research: Natural Sciences (DFF 4181-00565 to S.N.). S.C.E. and R.D.H. acknowledge funding from NSF-OPP 1836839. M. P. Bjorkman was funded by the research and development projects to future research leaders at FORMAS—the Swedish Research Council for Sustainable Development (grant agreement 2016-01187) and the Latnjajaure Field Station monitoring programme by the strategic research environment of BECC—Biodiversity and Ecosystem services in a Changing Climate. E.E.B. was funded by Biological Integration Institutes Grant NSF-DBI-2021898. G.N.D. was funded by a Schmidt Science Fellowship. B.F. was supported by the EU Marie Curie-IF 892383 (RESCATA). G.H.R.H. was funded by the Natural Science and Engineering Council of Canada, ArcticNet and the Canadian International Polar Year Program, with logistical support from Polar Continental Shelf Program and the Royal Canadian Mounted Police. E.K. was funded by the Academy of Finland (347188). N.L. was funded by the Canada Research Chair Program and the Natural Sciences and Engineering Council of Canada. P.M. was funded by the Estonian Academy of Sciences (research professorship for Arctic studies). M. Petit Bon was supported by the Governor of Svalbard (Svalbard Environmental Protection Fund, grant project number 15/128), the Research Council of Norway (Arctic Field Grant, project number 269957) and the National Science Foundation (grant ANS-2113641). E.P. was funded by the US National Science Foundation and the National Geographic Society. R.R. was supported by Danish National Research Foundation (DNRF100 and DNRF168). N.L is Canada Research Chair at the Centre d'Études Nordiques and P.M. is Chair of Biodiversity and Nature Tourism at the Estonian University of Life Sciences.

**Author contributions** M.G.C. conceived the study together with A.D.B. (who initiated the study), I.H.M.-S., S.C.E. and S.N. I.H.M.-S. obtained funding for the data-synthesis research. M.G.C., A.D.B. and S.C.E. prepared and cleaned the plant composition data from ITEX+. M.G.C. conducted the analyses and wrote the manuscript, with contributions from I.H.M.-S., A.D.B., S.C.E., S.N., P.A., R.A., J.M.A., L.B., R.G.B., M. P. Bjorkman, N.B.-L., E.E.B., E.J.C., J.H.C.C., G.N.D., B.F., B.C.F., G.H.R.H., R.D.H., T.T.H., I.B.D.J., A.K.J., I.S.J., E.K., O.K., K.K., T.H.M.K., S.I.L., N.L., J.L., P.M., J.M., A.M., U.M., R.M., M.-L.N., M. Petit Bon, E.P., K.R., R.R., C.R., I.R., J.M.S.-D., G.S.-S., N.M.S., F.S., S.S., M.J.S., L.S., B.S., A.T., C.E.T. and M.V. ITEX+ data contributors (I.H.M.-S., S.N., P.A., R.A., J.M.A., R.G.B., M. P. Bjorkman, N.B.-L., E.J.C., J.H.C.C., G.H.R.H., R.D.H., T.T.H., I.B.D.J., A.K.J., I.S.J., E.K., O.K., K.K., T.H.M.K., S.I.L., N.L., P.M., J.M., A.M., U.M., M.-L.N., M. Petit Bon, E.P., K.R., R.R., C.R., I.R., N.M.S., S.S., L.S., B.S., A.T., C.E.T.) provided plant composition data and ArcFunc participants (I.H.M.-S., A.D.B., S.C.E., S.N., L.B., E.E.B., J.L., J.M., R.M., C.R., J.M.S.-D., G.S.-S., F.S., M.J.S., M.V.) contributed to the initial study framework.

**Competing interests** The authors declare no competing interests.

**Additional information**
**Correspondence and requests for materials** should be addressed to Mariana García Criado.

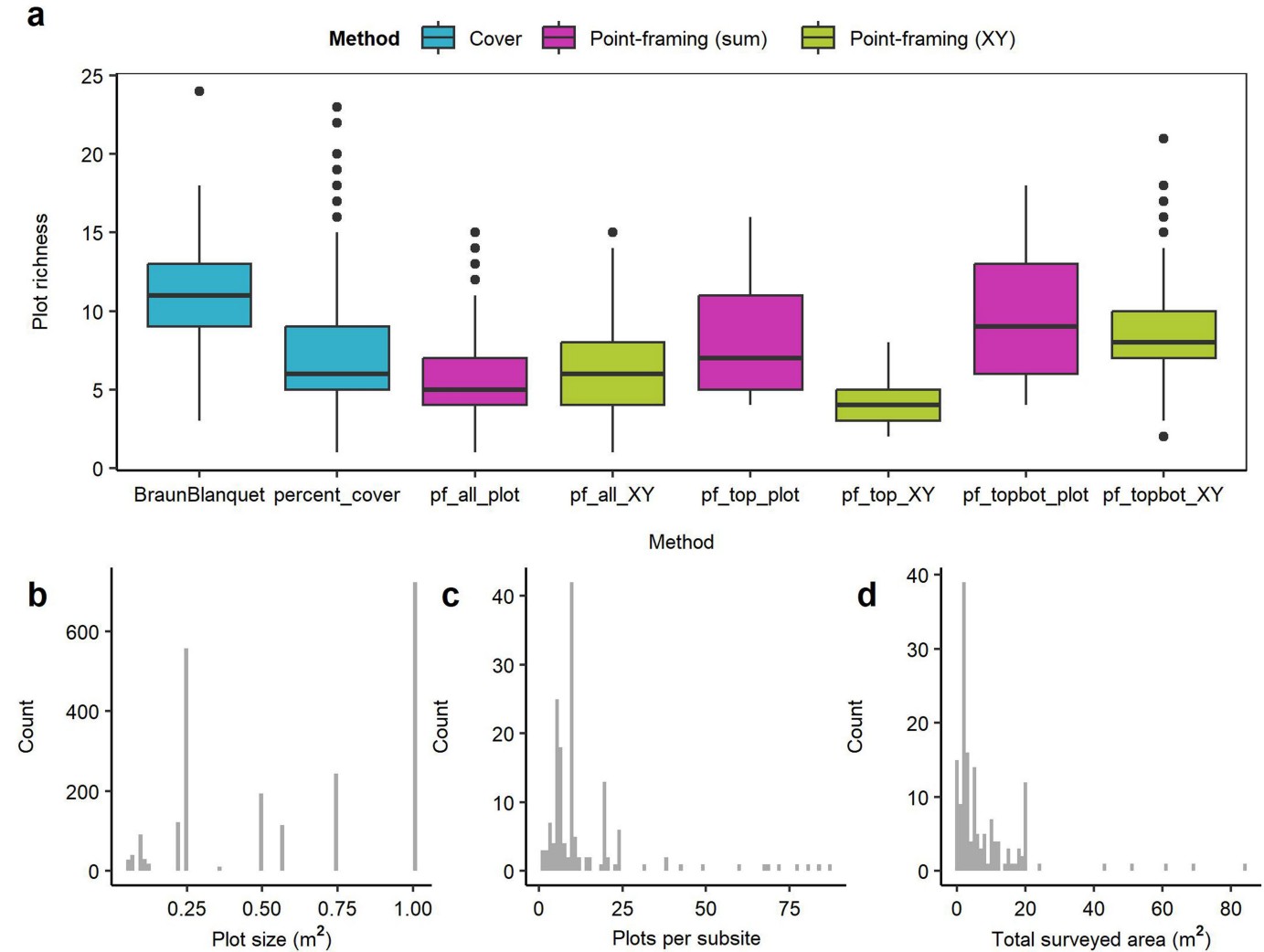

**Extended Data Fig. 1 | Comparison of data collection methods and overview of surveyed plot area across our dataset. a)** Current species richness at the plot level obtained by different field surveying methods. Mean values of richness are similar across point-framing data with and without coordinate values, with slightly lower values for top-only data as would be expected. Boxes are coloured according to the main survey method (*n* = 2,174 plots). Central boxplot lines represent medians and vertical whiskers represent the 25% and 75% percentiles. pf_plot = pointframing with no coordinates (sum of hits), pf_XY = pointframing with XY coordinates, top = top hits only, topbot = top and bottom hits only, all = all hits (including middle hits). Variability in **b)** plot size, **c)** number of plots per subsite, and **d)** total surveyed area, calculated as plot size * plots per subsite.

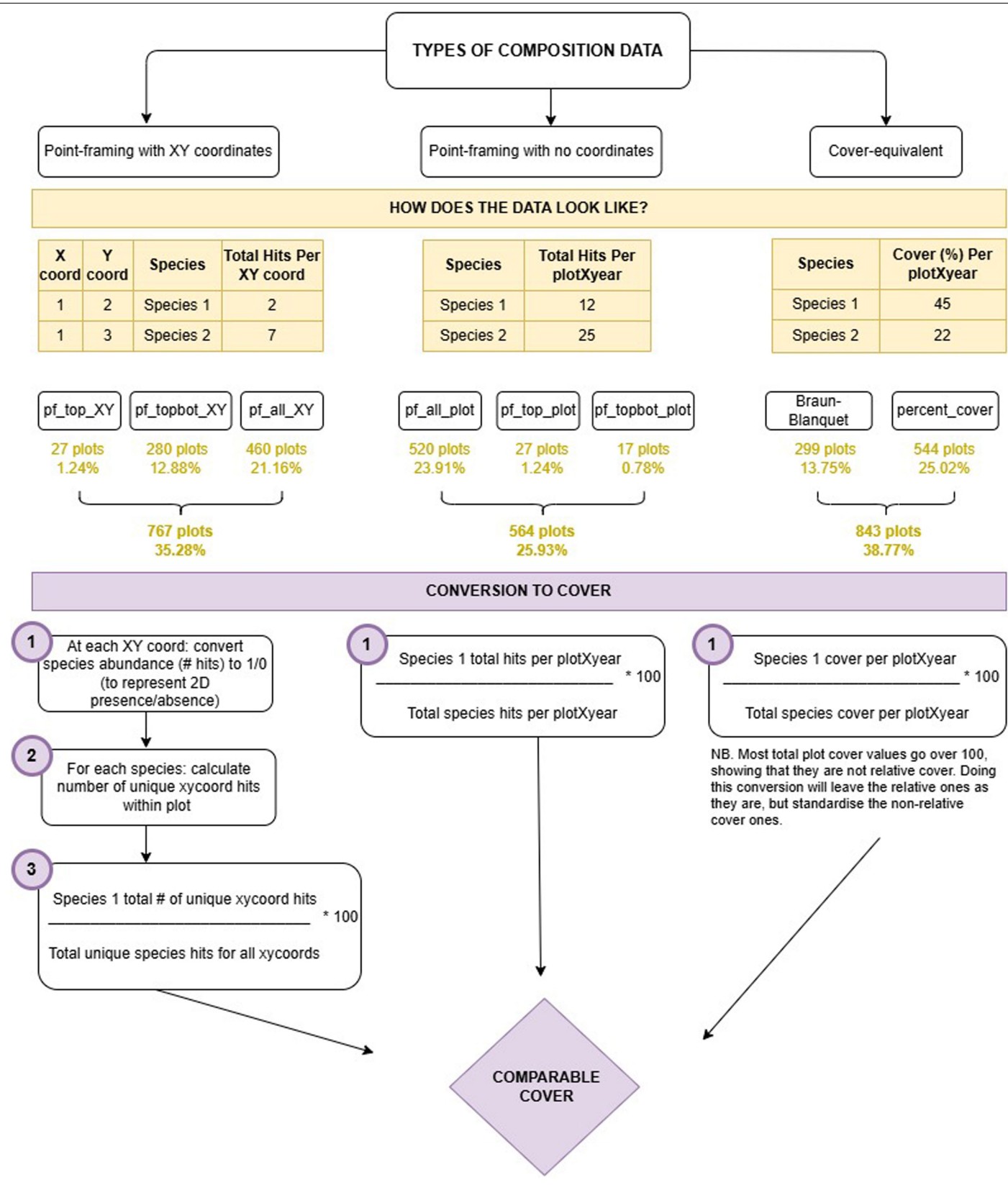

**Extended Data Fig. 2 | Overview of plant composition data types and their conversion to comparable cover.** Conceptual diagram showing the different types of data compiled within the ITEX+ dataset and the process to convert them to comparable cover values. The total number of plots in the dataset is 2,174.

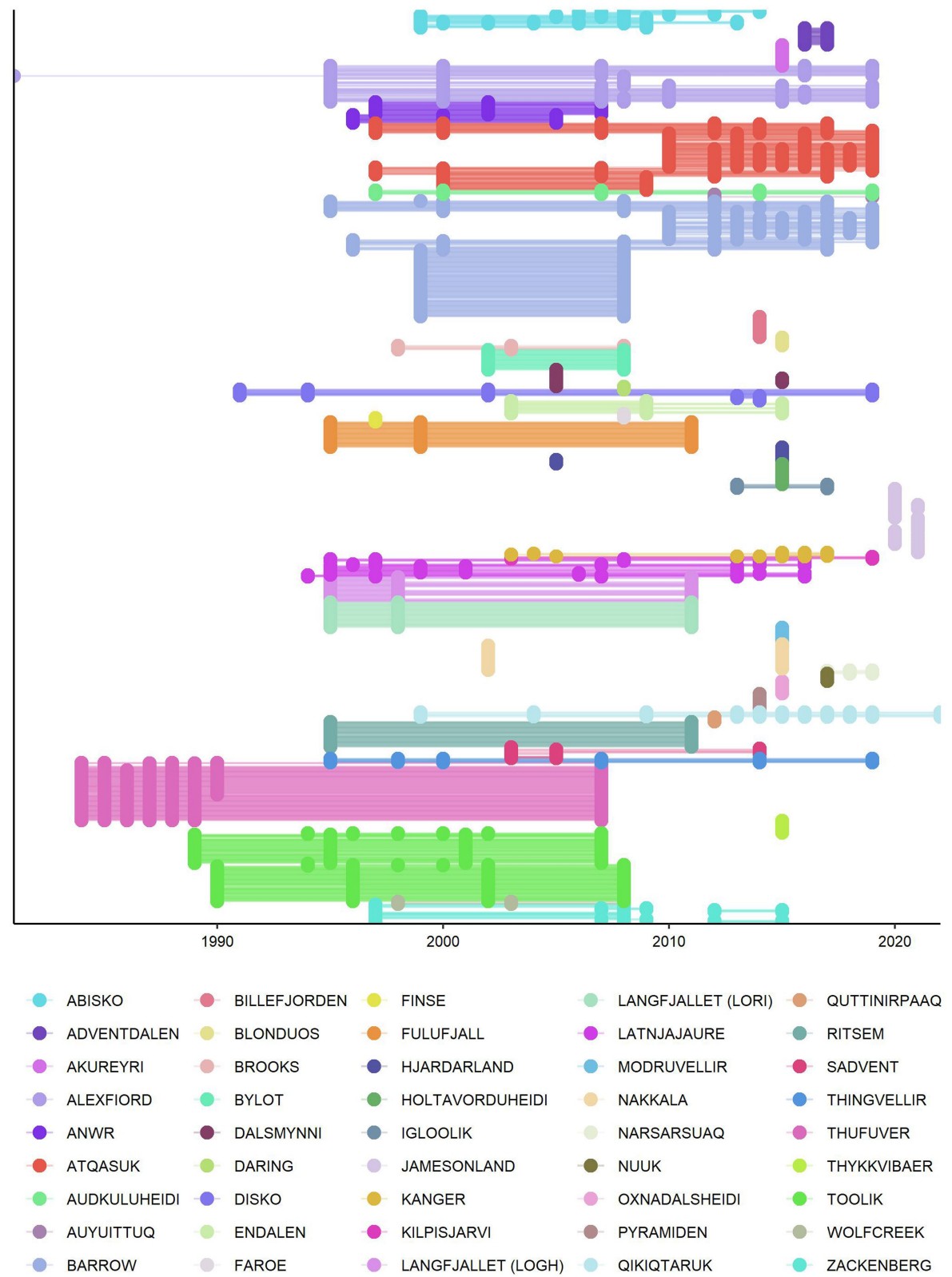

**Extended Data Fig. 3 | Timeline of the surveying and monitoring period for each plot included in our dataset.** Each colour represents a study area, with lines showing the duration of the monitoring period and points representing survey years per plot. Lines and points are coloured by study area and ordered alphabetically from top to bottom. Plots monitored for shorter than five years were only included in the spatial analyses.

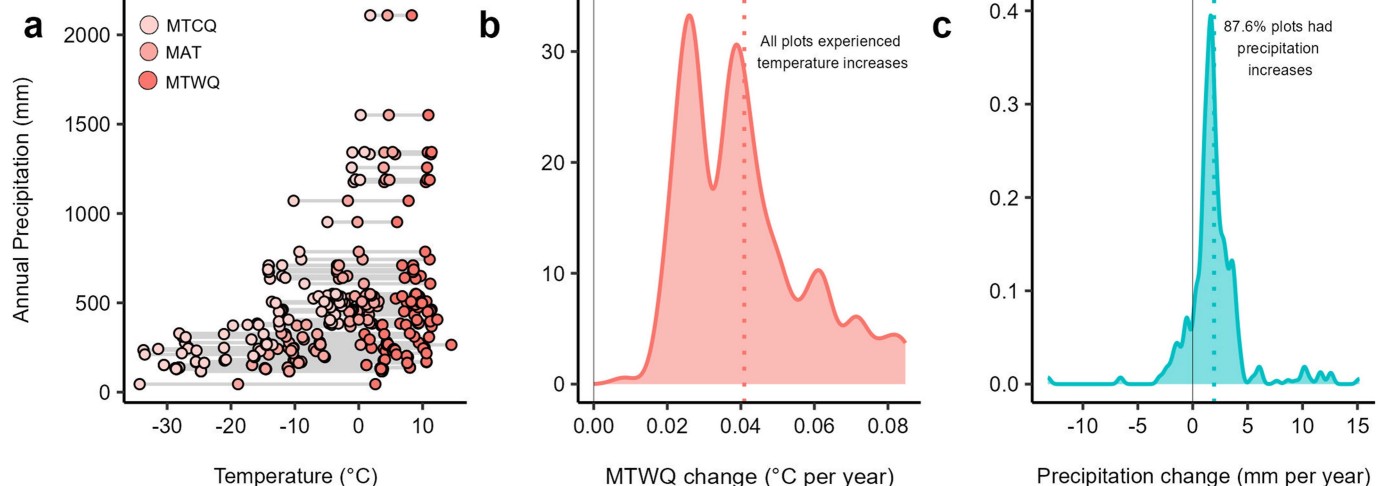

**Extended Data Fig. 4 | Overview of the climatic space of our plots and their climate change over time.** All plots experienced warming and the majority of plots experienced increasing precipitation. **a**) Climatologies of all subsites across the tundra according to their temperature and precipitation variables over the 1978–2013 time period. Each line represents a subsite, and each coloured point a temperature variable. MTCQ = mean temperature of the coldest quarter, MAT = mean annual temperature, MTWQ = mean temperature of the warmest quarter. Change over time in **b**) temperature, and **c**) precipitation in our Arctic plots, calculated as the slopes of annual climate change over time. Dotted colour lines in **b**) and **c**) represent the mean slope of climatic change across plots. Black lines positioned at zero are included for reference.

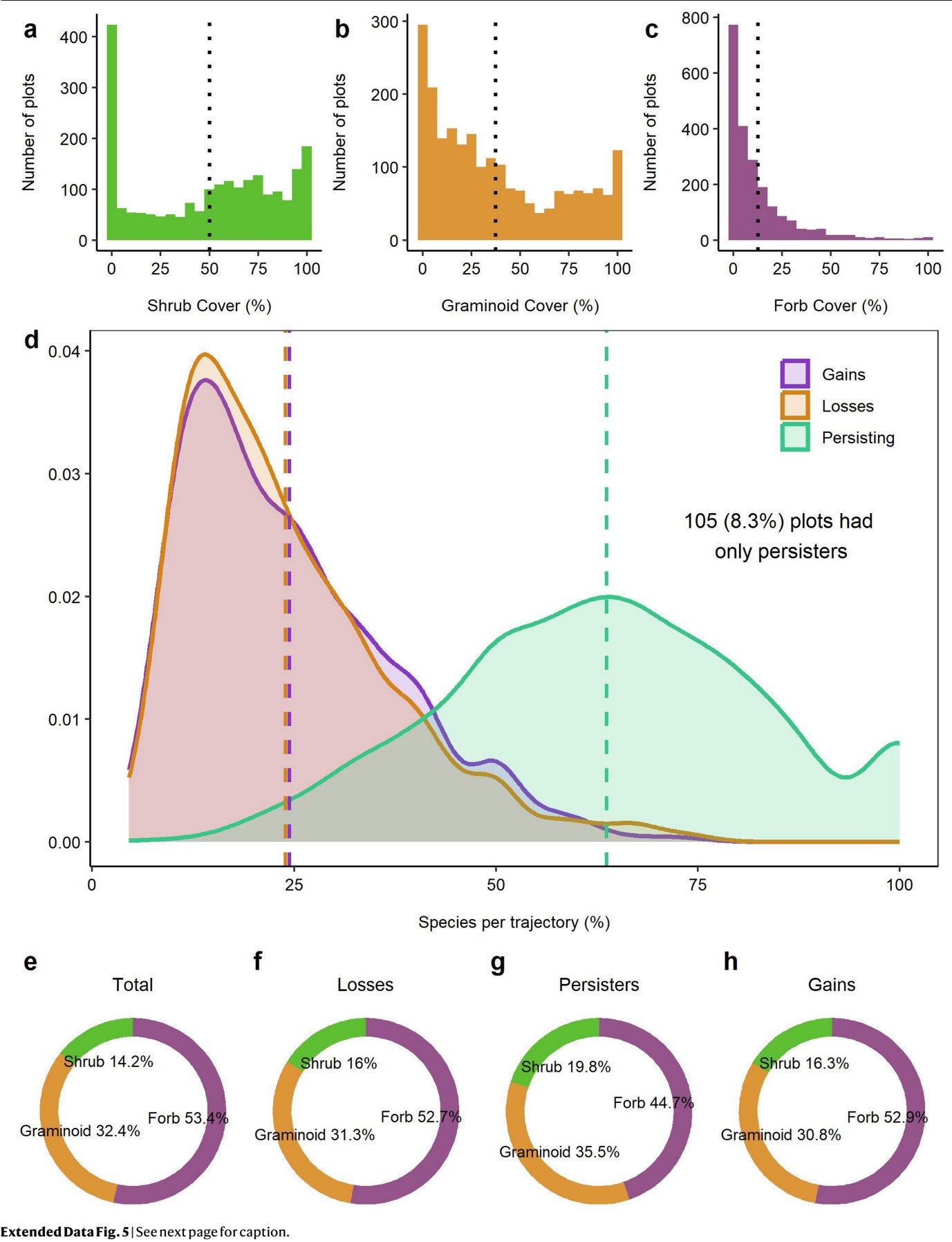

**Extended Data Fig. 5** | See next page for caption.

**Extended Data Fig. 5 | Functional group composition and species trajectories. a-c)** Mean cover of the different functional groups across all ITEX+ plots. **a)** Shrub and **b)** graminoid proportions are similar, while proportion of **c)** forb cover is much smaller across plots. High frequency of shrubs was relatively common across plots, and several plots were fully dominated by shrubs and by graminoids. Dotted lines indicate overall mean cover per functional group. **d)** There were substantially more species persisting in plots over time (64%) than species gained (19%) or lost (17%) species across plots. Proportion of species per trajectory across plots (gains, losses, persisting). Each plot is represented in each density curve via the proportion of species belonging to each trajectory. Dashed lines represent the mean proportion of species per trajectory and per plot. **e-h)** Proportions of species becoming lost, persisting or gained were similar across functional groups, and to overall dataset composition. Doughnut charts show the relative abundance of each functional group within a given trajectory: **e)** represents functional group composition proportion within the dataset for comparison with **f)** species losses, **g)** persisting species, and **h)** species gains.

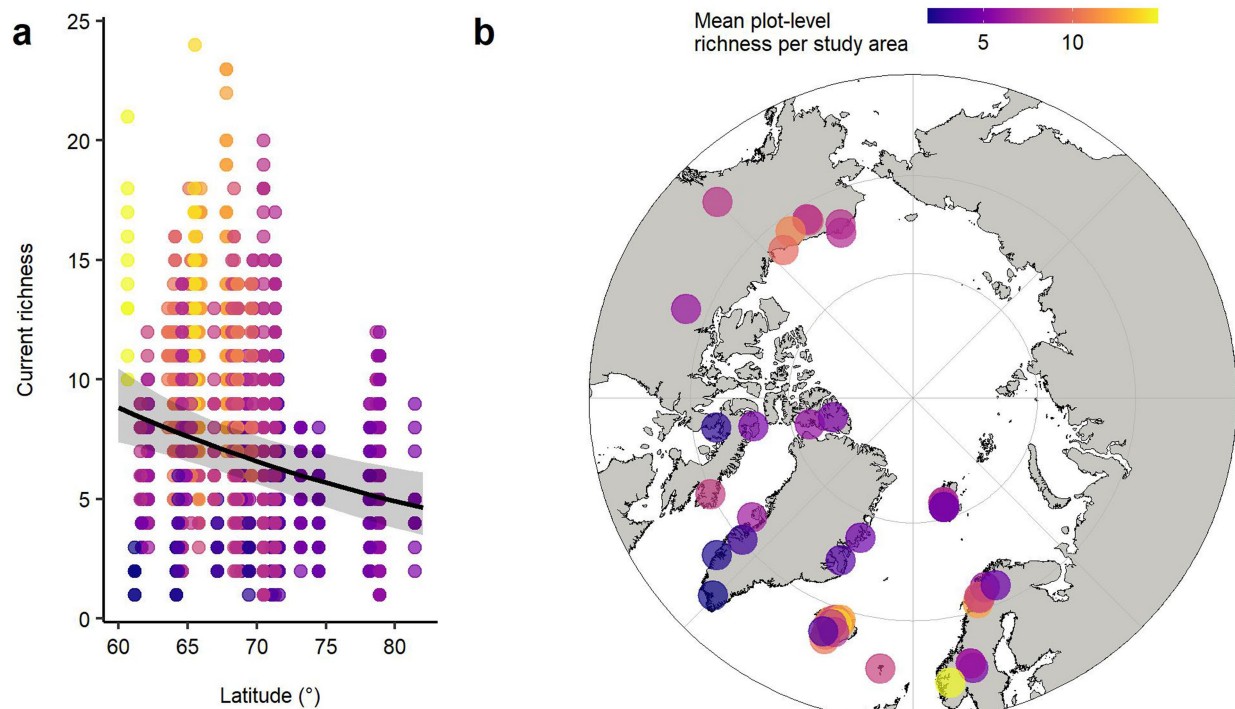

**Extended Data Fig. 6 | Relationship between latitude and species richness.**
Species richness is greater at lower latitudes across the Arctic. **a)** Richness per plot at the last monitoring year across our latitudinal gradient of 20.78°. Each point represents a plot, coloured by the mean plot richness per study area, and darker shades indicate overlap of multiple plots ($n$ = 2,174). The black line represents the predicted model fit and bands show the 95% credible intervals.

**b)** Mean plot-level richness per study area, coloured according to the richness gradient. This mean calculation is done for visualisation purposes only, with all analyses and estimates elsewhere using individual plot-level richness, unless directly indicated. A few sites are labelled for reference. Polar projection with a southern limit of 57° latitude. Map created in R with the ggOceanMapsData[69] package v.1.4.

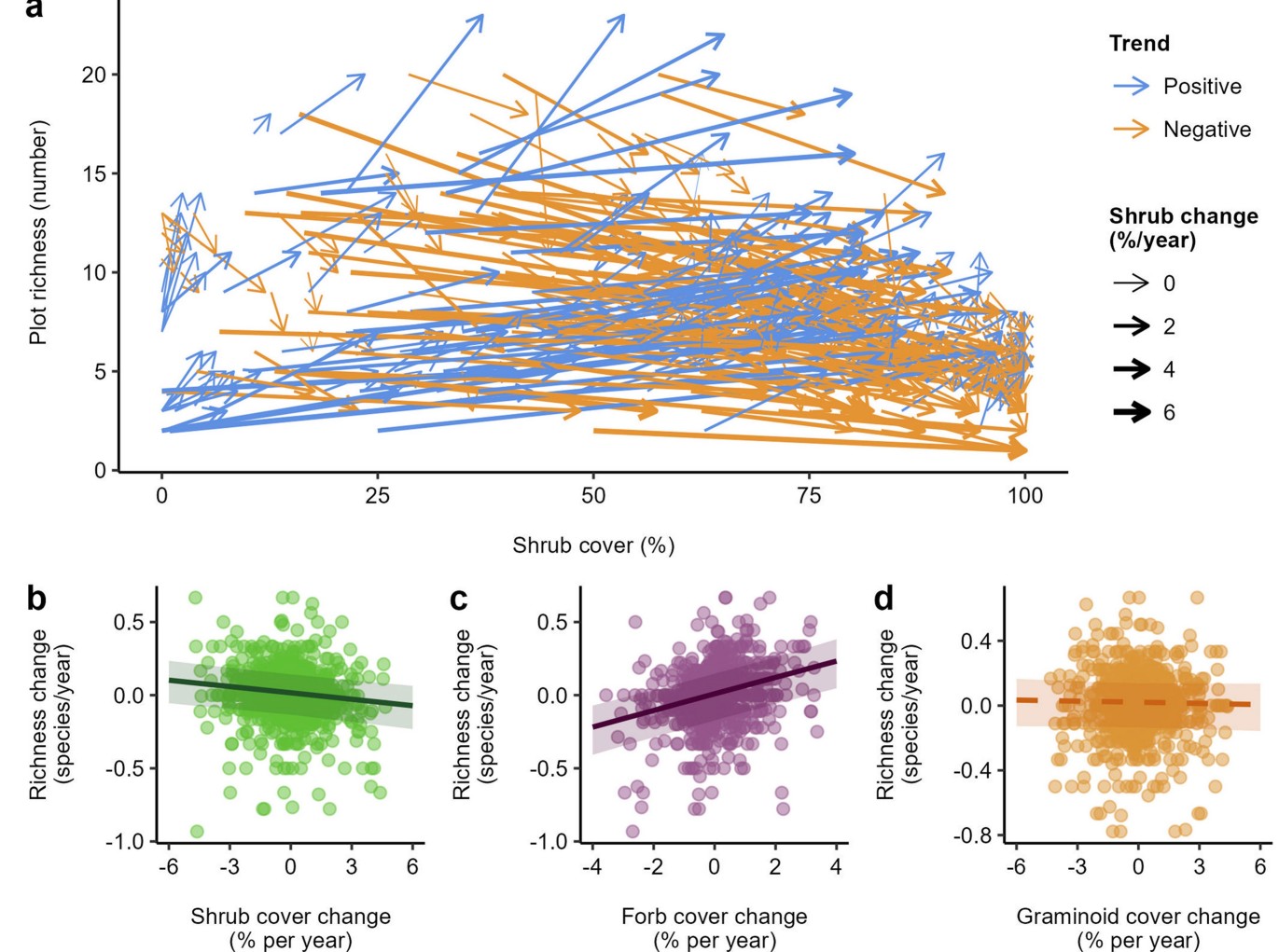

**Extended Data Fig. 7 | Relationships between richness and its change with functional group cover and its change. a)** Plot richness change was related to shrub cover increases over time, but it was not dependent on starting shrub cove (Supplementary Table 5). Each arrow connects the first and last monitoring point for each plot, with the arrow head pointing at the end time point. Arrow colours indicate the relationship between shrub cover increase and plot richness. 'Positive' indicates that plot richness increased as shrub cover increased. 'Negative' indicates that plot richness decreased as shrub cover increased. Arrow thickness indicates the magnitude of shrub change over time. Only plots where shrub cover increased over time are displayed (*n* = 432). **b-d)** Models of richness change as a function of functional group change (without extreme values of cover change). Values were removed when the slopes of functional group change were greater than three times the standard deviation. We found that the relationships hold up for shrub cover change (slope = −0.03, 95%CI = −0.04 to −0.02, conditional $R^2$ = 0.15, marginal $R^2$ = 0.06) and for forb cover change (slope = 0.06, 95%CI = 0.05 to 0.07, conditional $R^2$ = 0.21, marginal $R^2$ = 0.1). Graminoid change remains non-significant (slope = 0.002, 95%CI = −0.007 to 0.01, conditional $R^2$ = 0.14, marginal = 0.04). **b)** Richness decreased as shrub cover increased over time, but increased when **c)** forb cover increased. **d)** There was no relationship between richness change and graminoid cover change. Scatterplots represent richness change over time as a function of changes in cover of shrubs, forbs and graminoids. Points represent slopes of linear models of change in richness and in functional group change per plot over time. Lines represent predicted model fits and bands show the 95% credible intervals. Dashed lines indicate models whose credible intervals overlapped zero, and solid lines show models whose credible intervals did not overlap zero.

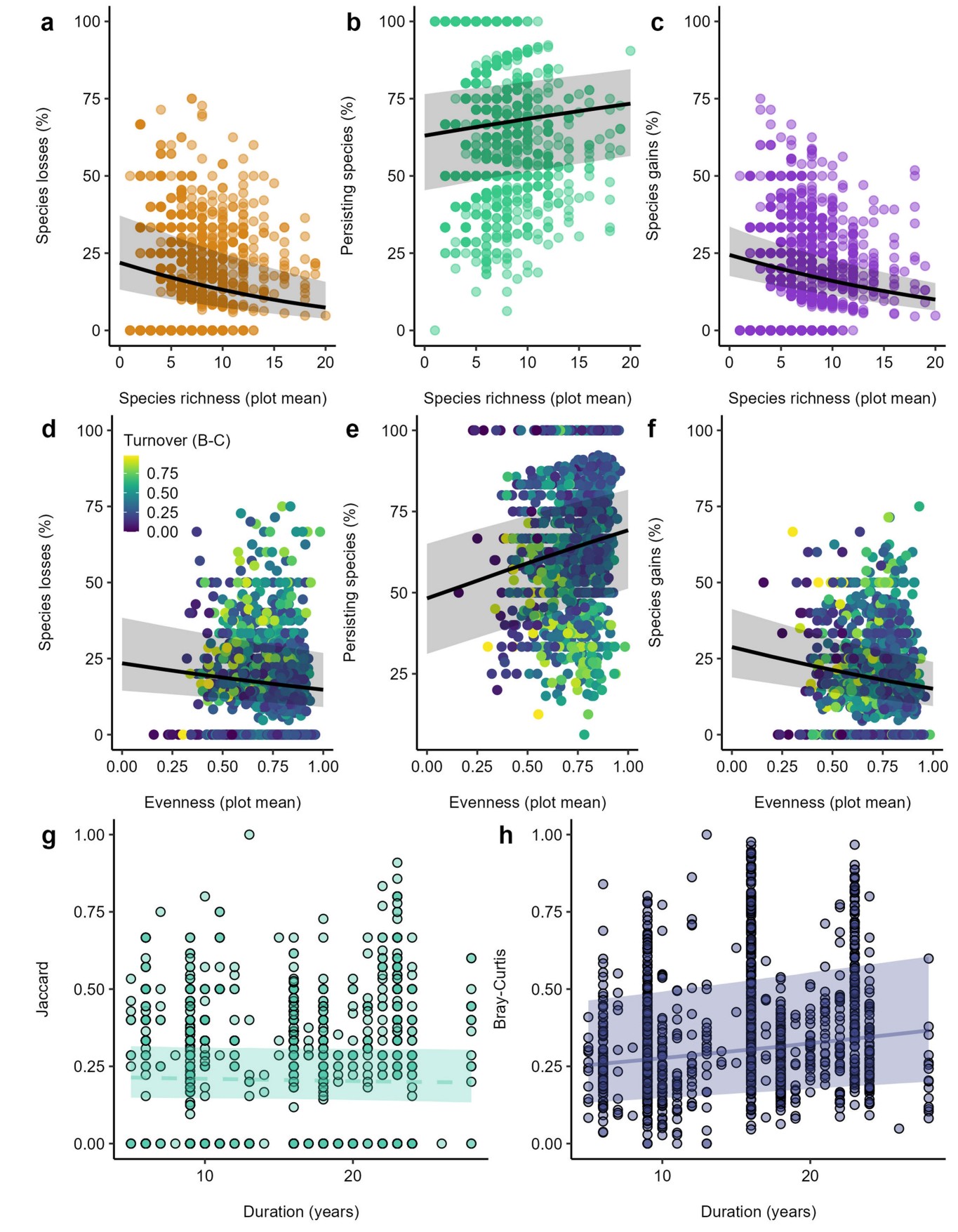

**Extended Data Fig. 8 |** See next page for caption.

**Extended Data Fig. 8 | Relationships between species trajectories and species richness and evenness, and between turnover and duration.**
a-f) More species-rich and/or even plots had a greater proportion of persisting species, and fewer local species losses and gains over time. a-c) Proportion of species per trajectory as a function of mean plot richness over time (as number of species) for a) species losses, b) persisting species, and c) species gains ($n = 1,266$). d-f) Proportion of species per trajectory as a function of plot mean evenness over time for d) species losses, e) persisting species and f) species gains ($n = 1,263$). Points are coloured by turnover (measured as Bray-Curtis). g-h) Relationship between turnover metrics and study duration. g) There is a non-significant relationship between Jaccard turnover and study duration and h) a positive relationship between Bray-Curtis turnover and study duration ($n = 1,266$ for each metric). Each point represents a plot. Solid lines represent predicted model fits (whose credible intervals do not overlap zero) and a dashed line represents a model estimate whose credible intervals overlap zero. The bands show the 95% credible intervals.

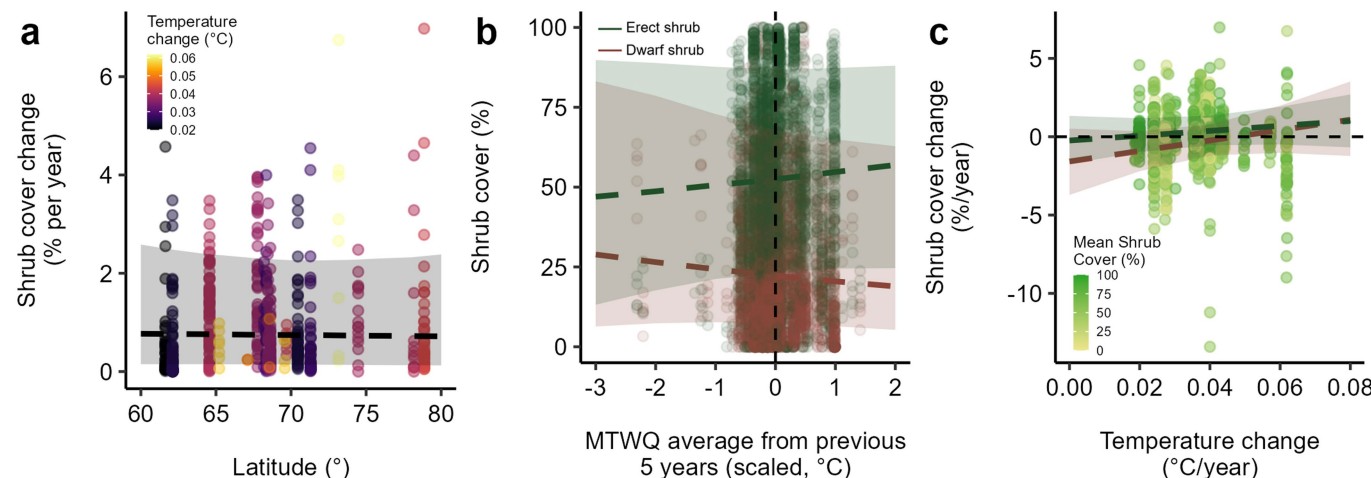

**Extended Data Fig. 9 | Relationships between main drivers of diversity change (temperature increases and shrub expansion).** These reflect the *post hoc* analyses, with model outputs in Supplementary Table 7. **a**) Shrub cover change was not related to latitude (*n* = 503). **b**) Shrub cover sensitivity to the mean MTWQ of the previous five years differed between shrub categories: erect shrub cover was greater at warmer temperatures, and dwarf shrub cover was lower at warmer temperatures. Mean temperatures are centred per subsite to account for variability and enable model convergence (*n* = 6,715). **c**) Shrub cover change rates per plot were not related to temperature change rates over the 1987–2013 period (*n* = 665). Lines represent predicted model fits and bands show the 95% credible intervals. Dashed lines indicate models whose credible intervals overlapped zero.

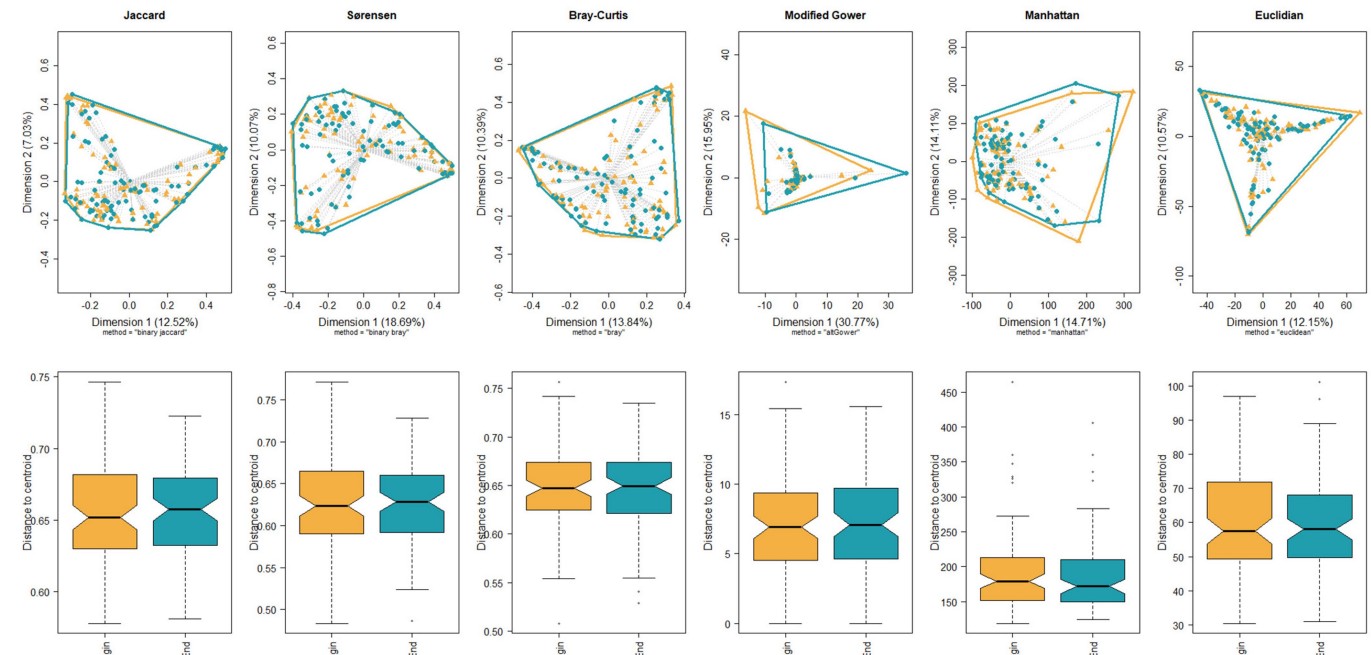

**Extended Data Fig. 10 | Ordination analyses with multiple metrics.** Subsites showed no homogenisation or differentiation over time across the Arctic. The panel shows Principal Coordinate Analyses with six β-diversity metrics. Yellow triangles and blue circles represent the start (i.e., baseline) and the end (i.e., final) time points for all subsites, respectively. Convex hulls are drawn around them following the same colour scheme. The boxplots show the mean distance to centroid for all start versus end subsites. In boxplots, the middle line is the median, the lower and upper hinges are the first and third quartiles, the upper whisker extends from the hinge to the largest value within 1.5 * IQR from the hinge (where IQR is the inter-quartile range) and the lower whisker extends from the hinge to the lowest value within 1.5 * IQR of the hinge. Data beyond the end of the whiskers are outliers and plotted as points.

# Reporting Summary

## Statistics

For all statistical analyses, confirm that the following items are present in the figure legend, table legend, main text, or Methods section.

| n/a | Confirmed | |
|---|---|---|
| ☐ | ☒ | The exact sample size (*n*) for each experimental group/condition, given as a discrete number and unit of measurement |
| ☐ | ☒ | A statement on whether measurements were taken from distinct samples or whether the same sample was measured repeatedly |
| ☐ | ☒ | The statistical test(s) used AND whether they are one- or two-sided<br>*Only common tests should be described solely by name; describe more complex techniques in the Methods section.* |
| ☐ | ☒ | A description of all covariates tested |
| ☐ | ☒ | A description of any assumptions or corrections, such as tests of normality and adjustment for multiple comparisons |
| ☐ | ☒ | A full description of the statistical parameters including central tendency (e.g. means) or other basic estimates (e.g. regression coefficient) AND variation (e.g. standard deviation) or associated estimates of uncertainty (e.g. confidence intervals) |
| ☐ | ☒ | For null hypothesis testing, the test statistic (e.g. *F*, *t*, *r*) with confidence intervals, effect sizes, degrees of freedom and *P* value noted<br>*Give P values as exact values whenever suitable.* |
| ☐ | ☒ | For Bayesian analysis, information on the choice of priors and Markov chain Monte Carlo settings |
| ☐ | ☒ | For hierarchical and complex designs, identification of the appropriate level for tests and full reporting of outcomes |
| ☒ | ☐ | Estimates of effect sizes (e.g. Cohen's *d*, Pearson's *r*), indicating how they were calculated |

*Our web collection on statistics for biologists contains articles on many of the points above.*

## Software and code

Policy information about availability of computer code

| | |
|---|---|
| Data collection | No specific software was used to collect data. We used the software and programming language R version 4.1.0 (R Core Team, 2022) to produce the input data file of plant composition. |
| Data analysis | We used the software and programming language R version 4.1.0 (R Core Team, 2022). Bayesian hierarchical models were fitted using the 'brms' package v2.17 in R (Bürkner, 2017). Principal Coordinate Analyses were carried out with the 'vegan' v2.6-2 (Oksanen et al. 2020) and 'ape' v5.6-2 (Paradis & Schliep, 2018) packages in R. Polar projection maps were created with the 'ggOceansMapsData' package v1.454 (Vihtakari 2024). The R code to generate the figures and analyses of this manuscript is accessible at https://doi.org/10.5281/zenodo.14884498 |

For manuscripts utilizing custom algorithms or software that are central to the research but not yet described in published literature, software must be made available to editors and reviewers. We strongly encourage code deposition in a community repository (e.g. GitHub). See the Nature Portfolio guidelines for submitting code & software for further information.

# Data

Policy information about availability of data

All manuscripts must include a data availability statement. This statement should provide the following information, where applicable:
- Accession codes, unique identifiers, or web links for publicly available datasets
- A description of any restrictions on data availability
- For clinical datasets or third party data, please ensure that the statement adheres to our policy

Plant composition data is available at https://doi.org/10.5281/zenodo.14884498. Climate data from CHELSA can be accessed at https://chelsa-climate.org/ and snow data is available at https://springernature.figshare.com/collections/ARCLIM_bioclimatic_indices_for_the_terrestrial_Arctic/6216368

# Research involving human participants, their data, or biological material

Policy information about studies with human participants or human data. See also policy information about sex, gender (identity/presentation), and sexual orientation and race, ethnicity and racism.

| | |
|---|---|
| Reporting on sex and gender | Not applicable. |
| Reporting on race, ethnicity, or other socially relevant groupings | Not applicable. |
| Population characteristics | Not applicable. |
| Recruitment | Not applicable. |
| Ethics oversight | Not applicable. |

Note that full information on the approval of the study protocol must also be provided in the manuscript.

# Field-specific reporting

Please select the one below that is the best fit for your research. If you are not sure, read the appropriate sections before making your selection.

☐ Life sciences    ☐ Behavioural & social sciences    ☒ Ecological, evolutionary & environmental sciences

For a reference copy of the document with all sections, see nature.com/documents/nr-reporting-summary-flat.pdf

# Ecological, evolutionary & environmental sciences study design

All studies must disclose on these points even when the disclosure is negative.

| | |
|---|---|
| Study description | This study quantifies local-scale plant species richness and composition and its change over time across plots in the Arctic, and identifies the geographic, climatic and biotic drivers behind these changes. The majority of models are Bayesian hierarchical models with a subsite random effect. Data families were chosen depending on the structure of the response variable and include Gaussian, negative binomial, beta, zero-inflated-beta and zero-one-inflated beta. Sample size was dependent on the response variable, and it is specified throughout the manuscript. |
| Research sample | We used a compilation of 42,234 records of 490 vascular plant species from 2,174 plots at 155 subsites within 45 study areas across the Arctic. Out of the 2,174 plots, 787 plots (36.2%) had only been surveyed once (and thus were only included in spatial analyses) and 1,266 (58.2%) plots were surveyed more than once and over a minimum period of five years (and thus were used for both spatial and temporal analyses). |
| Sampling strategy | We retained records that complied with our criteria (i.e., plots equal or smaller to 1m2, north of 60 degrees latitude, consistent surveying methods over time, only vascular plants, plots with <10% morphospecies). We used all available records per plot in order to appropriately capture local-scale plant diversity patterns and trends over time. |
| Data collection | Anne Bjorkman, Mariana García Criado and Sarah Elmendorf cleaned and curated the ITEX+ database. |
| Timing and spatial scale | The data in this manuscript span the period 1981-2022. Plots were monitored over different periods, with a mean study duration of 8 years (range = 1 to 28), a mean of 3 monitoring time points per plot (range = 1 to 11) and a mean time between surveys of 5 years (range = 1 to 26). Our dataset contains records from Europe, Greenland and North America north of 60 degrees latitude. Our database comprises 2,174 plots in 155 subsites within 45 study areas across the Arctic. Plots are equal or smaller to 1m2. |
| Data exclusions | Plant records from the ITEX+ database were excluded in the following instances: 1) records were obvious duplicates, 2) plots that were under 60 degrees latitude, 3) plots that had inconsistent surveying methods and/or plot sizes over time, 4) plots that were greater than 1m2, 5) plots that had >10% of morphospecies, and 6) records that were not vascular plants (i.e., non-vascular plants |

and non-biotic records). For temporal analyses, plots were excluded when they had been surveyed less than twice and over a shorter time period than five years.

Reproducibility | All code and data are publicly available at https://doi.org/10.5281/zenodo.14884498

Randomization | Randomization was not applicable to this study as we required all available relevant records in order to capture local-scale plant diversity patterns and trends over time.

Blinding | Blinding was not applicable to this study as no participants were involved and no comparison was made between control and experimental studies.

Did the study involve field work? ☐ Yes ☒ No

# Reporting for specific materials, systems and methods

We require information from authors about some types of materials, experimental systems and methods used in many studies. Here, indicate whether each material, system or method listed is relevant to your study. If you are not sure if a list item applies to your research, read the appropriate section before selecting a response.

## Materials & experimental systems

| n/a | Involved in the study |
|---|---|
| ☒ | ☐ Antibodies |
| ☒ | ☐ Eukaryotic cell lines |
| ☒ | ☐ Palaeontology and archaeology |
| ☒ | ☐ Animals and other organisms |
| ☒ | ☐ Clinical data |
| ☒ | ☐ Dual use research of concern |
| ☒ | ☐ Plants |

## Methods

| n/a | Involved in the study |
|---|---|
| ☒ | ☐ ChIP-seq |
| ☒ | ☐ Flow cytometry |
| ☒ | ☐ MRI-based neuroimaging |

