## [Peer Review File · Nature]

Plant diversity dynamics over space and time in a warming Arctic

Corresponding Author: Dr Mariana García Criado

This file contains all reviewer reports in order by version, followed by all author rebuttals in order by version. Parts of this Peer Review File have been redacted as indicated to remove third-party material.

Version 0:

Reviewer comments:

Referee #1

(Remarks to the Author)

The manuscript entitled “Plant diversity dynamics over space and time in a warming Arctic” by García Criado et al. describes how vascular plant species richness and composition have changed over time, and it aims to identify the geographic, climatic, and biotic drivers behind such changes. I was quite pleased to read this manuscript, which is concise, straightforward, and very well written.

At first sight, the results obtained in this study with such a large sampling effort, seem very unspectacular. For example: no directional trend in changes in species richness across the Arctic. Or: no biome-wide homogenization across the Arctic (the plots shifted in their composition in all possible directions), etc. Among such results, the detection that shrubification out-compete other groups stands out for a reader with finally a clear statement (line 496-498).

However, in my opinion, this is the strength of this study. It confirms the complexity of the Arctic ecosystems, and the impressive resistance of Arctic plant communities to climatic and global changes. This is also highlighted by authors, by enumerating (from line 527) other processes that could contribute to a lack of detected diversity change.

The study is very solid, based on more than 42,000 records, covering nearly 500 vascular plant species, more than 2,000 plots in 45 study areas across the Arctic (obtained from the International Tundra Experiment (ITEX+ database).

However, several statements, such as the impressive numbers listed above, in the Abstract, could be misinterpreted. For example, “only” 64% of study plots were monitored at least twice, and thus only those could be use both for spatial and temporal analyses. And the time of monitoring was never 40 years (at average 8 years period). Thus, the statement of “four decades” is somewhat misleading and should be adapted/explained accordingly (e.g. line 136). Similarly, the authors are often using the term “across the Arctic”... although large part of the Arctic is not covered by the study (notably Russian Arctic, see Fig. 1, Fig. S2, etc.).

The manuscript references previous literature appropriately. Eventually, two papers published very recently could be included:

Körner C. 2023. Concepts in Alpine plant ecology. *Plants* 12: 2666.

Zhang et al. 2023. Evolutionary history of the Arctic flora. *Nature Communications* 14: 4021.

Concluding, the manuscript is very well written, all analyses and graphics, tables resulting from them, as well as the supplementary materials are of excellent quality. I find that the conclusions and data interpretation are robust, valid and reliable. The results are of large interest to many researchers from several disciplines.

Referee #2

(Remarks to the Author)

In their manuscript “Plant diversity dynamics over space and time in a warming Arctic” the authors address the question how plant species composition and diversity changed over time and why in a warming arctic using resurveyed plots ranging from Europe to Canada (excl Russia). They found no directional changes of plant species composition in response to climate. However, species richness at plot level increased most in plots with higher temperature and shrub encroachment leading to higher species losses and shifts in species abundances.

The manuscript touches an interesting and important topic; that is how vegetation is expected to change with progressing climate change. Also, the data is interesting allowing for such kind of analyses. The results were mixed but in general do mirror those obtained from other, similar studies from different regions. While this is interesting in itself, I found the manuscript lacked novelty in the sense to provide new and mechanistic insights into changes of vegetation over time. While

the authors incorporated many different diversity indices into their analyses it is not clear what they expect from this and why (please also see my comments on the introduction). More explanation and insights into their choice of indices could have helped to strengthen the manuscript. Other studies for instance (e.g., Jandt et al Nature 2022; not cited) showed clear trends in winners and losers at plot scale over time and hypothesised that this may be linked to habitat types. Such could have been incorporated here. It seemed to me that the very generalistic approach chosen here (e.g., a lot of indices were correlated with many factors without formulating clear hypotheses why they matter and how) prevented insights into the real drivers of biodiversity change in the arctic.

More specific comments:

The Introduction was hard to follow. Many different concepts and ideas were mixed preventing a logical flow leading to the aims of this study. For instance, mechanistic explanation how and why species diversity in the arctic may change over time is only loosely touched or missing. The authors mention several processes but do not explain these in detail. This is problematic as it does not equip the reader with the knowledge needed to understand the difficult and complex results. For instance, in lines 161 to 172 the authors mention that species can be gained as well as lost in the arctic due to northward migration and extinction of cold-adapted species. However, this is poorly explained how. Why do species go extinct? Is it due to increased competition or to eradication of their psychological climate niche? Also which species can be expected to migrate northwards? Those with good dispersal ability or those being highly competitive (maybe shrubs)? Further, I think that this is highly contingent on scale – which is also mixed in the intro (often it was not clear whether a specific pattern was expected at local, regional or whole-arctic scale). Can we expect different patterns at plot, regional and whole-arctic scale? This all leads to poorly introduced research aims and questions.

The results looked interesting but the choice of figures shown in the main text was difficult to apprehend. Why mean temp of the warmest quarter as explanatory variable? In the intro the authors mentioned that warming winters may lead to pronounced changes (line 209). This needs better justification. Summary statistics in the regression plots would be helpful.

Further minor comments:

- Line 131: Referenced abstract needed – please provide. Please also note that max of 50 refs are allowed in the main text.
- Line 158: unprecise: there are arctic ecosystems that are not structured by plants. Do you mean those that are ice-free and terrestrial?
- Line 167: How? Some more mechanistic explanations on effects of poleward species movements could help to understand better the rationale of this study.
- Line 174: This is, of course, greatly scale dependent. It is often unclear which spatial scale is considered and discussed in the manuscript.
- Line 176: Also here some reference to the LDG could be helpful. With climate warming, what species richness could we expect where and why (for instance)?
- Line 180: To what extent? This is a very simplistic assumption and may need more nuanced explanation.
- Line 183: Unclear. Why would one assume that boreal species colonise low arctic habitats if they are more dissimilar than those of the high arctic?
- Lines 174 to 189: Can we expect a delay of species migrating northwards with climate? Not all species are equally well adapted to travel northwards mainly due to different dispersal strategies and biotic dependencies. There is broad literature on species colonising recently deglaciated areas, which show that some species are much faster in colonising new terrain than others linked to their traits. This is briefly mentioned in the paragraph below but again depth is missing.
- Line 205 to 217: Can specific hypotheses/likely scenarios be included here? For instance, what may happen to those permafrost soils? Do they behave in a similar way as all other habitat types? As for now it sounds that everything can happen everywhere in the arctic completely neglecting previous studies that have made an effort understanding changes in the arctic.
- Lines 241 to 244: And what about all the other indices calculated? How do these change? In general, formulating clear hypotheses (and not research questions) that are introduced above would help to reader to better understand the rationale of this manuscript.
- Line 254: This is the first-time dispersal is mentioned. It would be helpful to explain the effects of different dispersal capacities further up already.
- Line 596: Would be good to know the species richness of the final dataset used after cleaning.
- Line 624: Was climate (temp and precip) from the year of the survey used (I gather from the results that you did; but this needs to be explained better in the methods)? And why did you decide on temp of the warmest quarter for the main results?
- Figures
 - o Please add summary statistics to all regressions in the figures.
 - o Fig 3: Can you please add the data to the figures? There seems to be a wide variance in the data, which would be interesting to see.
 - o Fig 4: In all regression plots is huge variation especially in the centre of the plots. What is happening here? The regression line seems mainly to be driven by the outliers to the left and right respectively. Could this have affected the overall results and conclusions?
 - o Fig 4 e and f: Same here – the trend seems very weak (significant?) to justify the strong conclusions drawn from this.

Referee #3

(Remarks to the Author)

Plant diversity dynamics over space and time in a warming Arctic

In this paper the authors used an impressive data compilation to examine plant diversity dynamics in the Arctic. With high latitudes experiencing among the fastest rates of climate change, these results represent an important advance for our understanding of climate impacts on biodiversity. The authors find lots of variation in diversity change, though no changes in

local species richness on average. Richness increases were strongest in places that warmed the most, and shrub expansion was associated with the greatest richness declines. Turnover (i.e., species composition changes) and proportions of species gained and lost were related to temperature in the warmest quarter and temperature change. The authors fully embraced the complexity of biodiversity change, and I appreciated their efforts to combine multiple descriptions of the various aspects of diversity (richness, relative abundance, and composition). That said, I did not find it easy to follow all of the results, and perhaps some simplification is in order. In particular, the spatial analyses (the latitudinal and temperature gradients) were not part of the aims set out at the end of the introduction, which all focused on diversity change. Moreover, the spatial analyses made some additional assumptions to deal with the heterogeneity in plot size that were not well explained. Indeed, how variation in plot size was handled for all results where plots of different sizes were compared (i.e., the spatial analyses and changes in spatial beta-diversity) needs better explanation, and may be sufficiently problematic in some cases to warrant reanalysis or removal.

The editor directed me to focus my attention on the statistics, which were frequently overly difficult to parse. Table S1 represents an aesthetically pleasing attempt to summarise all the many models fit, but it was difficult to read (e.g., new acronyms not featured in the methods section, coloured symbols, and missing link functions). Greater structure in the methods, e.g., subheadings and separate descriptions linking analyses to specific questions would greatly increase clarity. At the moment, the details of a single analysis emerge across multiple paragraphs in the methods, and some important details (e.g., link functions) are omitted entirely. Critically, the analytical decisions made to deal with the variation in plot size are not completely described or justified. For example, what are the assumptions that accompany including $\log(\text{plot size})$ as a covariate in the different spatial models? Moreover, there was insufficient clarity in the description of the spatial beta-diversity analyses to understand exactly what was done.

With $\log(\text{plot size})$ included in regression models, I think questions (and inferences) change from species richness to species density (Gotelli & Colwell 2001). Whilst you have not simply divided richness by area, including this control or adjustment covariate is doing something similar. For example, to generate a predicted richness value from a model with $\log(\text{plot size})$ as a covariate, you need to set plot size to some value, meaning you are predicting richness for a given area, which is a density. Gotelli & Colwell (2001) are not at all dogmatic about one or the other (richness or density) being right, but they do show that they are fundamentally different things. What are the assumptions made when including the $\log(\text{plot size})$ term in the statistical model? Currently this cannot be fully discerned because the link functions are unknown. Assuming default link functions for the negative binomial (log link) and beta (logit link) error distributions for the richness, and evenness (and gains, losses and persistent proportion) models, respectively, result in fundamentally different assumptions about the scaling of diversity and plot size for the different response variables. The $\log(\text{richness}) \sim f(\log[\text{plot size}])$ is possibly defensible as a model of species density, though the assumptions entailed need to be fully explicated in the methods. However, I'm not convinced that $\text{logit}(\text{evenness}) \sim f(\log[\text{plot size}])$ makes sense (with the same argument applying for the models of the proportions of species gained, lost and persisting).

Unfortunately, I don't understand how dissimilarity was calculated for the spatial beta-diversity analyses. And, given the complete absence of any mention of plot size, strongly suspect these calculations are confounded by the variation in sample effort (i.e., plot size). Either sample effort needs to be standardised, or compositional (dis)similarity can only be calculated between plots of the same size. Similarly, calculating the distance to a single centroid across plots of varying size seems likely to confound compositional variation with variation in sampling effort.

So, that leaves the temporal analyses. And I was reassured to read that only plots with consistent sampling methods and sizes over time were retained for analyses (L583-584). Technically, these are also models of species density, though I acknowledge that referring to this as species richness is very common in the ecological literature (though you might consider referring to something other than richness for accuracy). I did get a bit confused between the "two-time point" and "temporal" models (terms that were introduced, but then not further described individually in detail in a single place). Table S3 says the model fit to calculate rates of species richness change assumed a negative binomial error distribution, though there is no mention of the link function. Assuming the default (log link), then the units of change are $\log(\text{richness})/\text{year}$; this needs to be checked and clarified throughout (e.g., Figure 2, Figure 4a-c, L440). Note also that changes in evenness calculated assuming a beta error distribution are not linear in time (due to the default logit link). Regardless of the scale of the modelled responses (natural, logarithmic, logit-transformed), I think these temporal analyses are the strongest component of the paper, and the ones that I would focus the presentation on (due to their documenting changes in plots of consistent size).

My final query regarding the temporal analyses, and in particular the link to temperature change, is whether the authors looked at an interaction between temperature change and a "baseline" temperature for given subsite? Earlier work, e.g., Antao et al 2020 NEE, suggests at least the potential for the effect of temperature change to depend on average temperatures at a site, and the authors also speculate about greater species losses in plots that warmed the most being due to cold plots experiencing the greatest warming (L462). Is this something that they can test directly with these data?

Other, line referenced comments:

L180: elevational gradients mirror latitudinal gradients in what sense? Temperature? Diversity? Small edit for clarity.

L242: I'm not sure what "reflecting" means in this sentence.

L267: random with respect to what?

L279: distracting (bolded) topic sentence. This bolded text might work as a sub-heading; but the only sentence in this

paragraph related to changes in species richness is the last one. The majority of this paragraph reports results not related to changes, and which were not part of the questions introduced at the end of the introduction.

L283-284: I'm likely missing something simple, but how does a slope of 0.06 species/°C equate to one species gained for an increase of 2°C?

L286: average (?) plot-level richness change was not different from zero?

Figure 2: do the estimates of richness change (i.e., slope estimates) on Fig. 2b and Fig. 2c come from the same model?

L 321: difficult to see these mean values on Figure 3a, b. Are they the (unlabelled) density plots on the righthand side? Can the density plots be labelled (i.e., axes and a panel tag) so as a reader can see this as the distribution of turnover values easily. Alternately, remove this pointer to the figure.

Similarly, are the relationships between turnover and species richness (L323, L324) visible on the plot? I got confused looking, and couldn't find them.

L403-405: does this refer to dissimilarity distance? It is worth making this explicit here, I was confused after the PCoA results (and distances to centroids).

L453: I'd call this incomplete sampling (i.e., present but not captured due to rarity and the finite nature of biodiversity samples).

L474: check language. "...fewer species losses and gains..." implies a count. However, the figures suggest that you modelled (and only report) proportional (or percentages).

L476-478: the logic underpinning how the species pool size can influence greater abundance turnover is not clear to me. If larger species pools have more even SADs, then perhaps stronger changes in relative abundance are possible?

L482: I cannot see shrubification on Figure 5.

L491: does the tense change for the first half of this sentence?

L523: here (and other places in this paragraph) you refer to a lack of diversity change (i.e., a statement about diversity in general). But haven't you found considerable changes, e.g., compositional. Does the "lack" refer to average richness change? If yes, be more precise.

L528: what is a slow lag?

L614-617: sometimes cover or proportions were analysed with response distributions that respect the bounded nature of the data (i.e., 0-1 interval), other times not? This is not necessarily a problem, but you need to state the assumptions and give reasoning for why different choices were made.

L638: if I understand correctly, you do not calculate a rate (i.e., changes per unit time). Turnover is species compositional dissimilarity between the baseline sample and the last sample (where duration between samples are highly heterogeneous [Figure S1]). It'd be good to see a plot of turnover ~ f(duration). Indeed, some examination of scale-dependencies in model results (see suggestions e.g., Spake et al. 2020 Ecology Letters) would be a great addition.

L648-653: this paragraph suggests that your losses and gains are counts of species. But the figures report percentages and Table S1 shows beta regression models. Needs clarification, i.e., what was the number used as the denominator to convert counts to proportions? Was it the same for gains and losses?

L662-664: this suggests that the two time point models are differences, i.e., they are not rates per unit time. Unless I am missing something, you need to check throughout that you have not described these as rates. Or are only the turnover results reported with these two-time point models? In general, the methods describe a lot of different models in a piecemeal fashion. Linking each question to the descriptions of their respective analyses in a single spot would increase clarity.

L669: what is the difference between a change over time model, and a plot change over time model? TableS1, whilst aesthetically appealing, is very difficult to read and doesn't help me to answer this question.

L675: are these two-time point models, or the temporal models (i.e., > 2 time points) as described above?

L699-700: the question underlying the null model is not clear to me. And, though I possibly missed it, I couldn't find a description of the null model either, just a terse statement stating no relationship between turnover and richness in the supplementary results.

L703-704: This is key. I was wondering what you did with the variation in the plot size. I think models that include log(plot size) effectively mean results describe species density, not species richness (Gotelli & Colwell 2001). Different link functions for the different response distributions are going to result in different scaling relationships, some of which might be hard to

justify.

L712-719: link functions? Do you include covariates for the zero or one components of the zero-inflated or zero-one-inflated beta regression models?

L726: I'd like to see all models fully described in the methods. I found Table S1 very difficult to read. e.g., the methods introduced the idea of spatial, two-time point models, and temporal models, but the table does not use this terminology, and introduces new acronyms for a different classification of models fit (GEO, CLIM, FG, CHG). There is no mention of link functions in the table or methods. Models with multiple likelihoods (e.g., zero-inflated or zero-one-inflated beta regression models) are not fully described (i.e., were covariates described in Table S1 used to predict the mean only?).

L731: how was species composition (and relative abundance) determined for subsites? I understand the lowest resolution of the data to be plots. How was the aggregation to subsites done? And, again for emphasis, I think the spatial variation in plot size is going to confound these calculations of composition dissimilarity. Without some standardisation, I think you can only calculate (dis)similarity between plots of the same size. Similarly, calculating the distance to a single centroid using plots of varying size is confounded by the variation in plot size to me.

Literature cited

L. H. Antão, A. E. Bates, S. A. Blowes, C. Waldock, S. R. Supp, A. E. Magurran, M. Dornelas, A. M. Schipper, Temperature-related biodiversity change across temperate marine and terrestrial systems. *Nature ecology & evolution*. 4, 927–933 (2020).

N. J. Gotelli, R. K. Colwell, Quantifying biodiversity: procedures and pitfalls in the measurement and comparison of species richness. *Ecol Lett*. 4, 379–391 (2001).

R. Spake, A. S. Mori, M. Beckmann, P. A. Martin, A. P. Christie, M. C. Duguid, C. P. Doncaster, Implications of scale dependence for cross-study syntheses of biodiversity differences. *Ecology Letters*. 24, 374–390 (2021).

Version 1:

Reviewer comments:

Referee #1

(Remarks to the Author)

I read the revised 2nd version of the manuscript with interest, and I am very satisfied with the new version. All my comments and suggestions for improvement are now integrated (with new analyses and illustrations). The more so, I would find this new version very suitable for publication in *Nature*.

Referee #2

(Remarks to the Author)

In their revised version of the manuscript "Plant diversity dynamics over space and time in a warming Arctic" the authors largely rewrote the text and reanalysed the data – this enormous effort needs to be congratulated.

Overall, I think the manuscript is much improved with many previously critical or unclear sections addressed, changed or deleted. However, the revision couldn't resolve all my main criticisms, largely centred around the dataset and interpretation. Specifically, I still wonder what the key message of this manuscript is. At some point (line 547) the authors say that the arctic communities are resistant against warming, but elsewhere is stated that communities lost species over time, mostly due to increasing cover of shrubs. Also it was mentioned that plot diversity is governed by idiosyncratic processes (I guess site-specific is a synonym here). This is not new at all and has been shown countless times before in nearly all ecosystems. Only when zooming out at larger scales more predictable patterns may emerge – which the authors couldn't detect because of the nature of the dataset. That said, I strongly suspect that plots <1m² are too small to address the specific question of vegetation change as response to warming across the whole arctic due to stronger local-scale specific effects (note that local scale richness is also greatly depended on pool size richness, which likely differed a lot depending on geographic location; also some islands were included with probably smaller pool sizes than the plots on the mainlands). May all this taken together add up leading to the weak ecological effects observed? In sum, what can the manuscript tell us new that hasn't been detected before (also by other studies led by the first author - García Criado et al *GEB* 2020; García Criado et al *Nat Com* 2023)? What is the major advancement and where do we need to go from here? All these questions still largely remain unanswered.

I am sorry not to be more optimistic at this point. But I do hope that the authors can use the comments to further refine their manuscript.

Specific comments:

Introduction: I very much like how the authors have restructured the introduction making it much stronger now in my opinion. Results: In general, it would be interesting to have the R² of the regressions mentioned in the results section as well. Knowing how much variance is explained by the predictors would help to better understand the results.

Line 146: This study doesn't span the whole arctic. Writing it here misleads on scale of the dataset.

Line 224: But what about other herbaceous species traveling polewards as well? Wouldn't these lead to an increase of species richness? Or do the authors argue that shrubby environments are in general poorer in species richness due to their superior competition over herbs? This part is still not clear to me.

Line 273: Again, what about shade-tolerant species travelling polewards? Could these compensate the losses?

Fig 2b: what does this plot show? It says it is richness per plot per year. But to assess directional change in richness over time wouldn't the gains and losses between consecutive sampling events of the same plot needed to be compared? As presented here it seems meaningless as different plots (also of different sizes with larger plots per se having higher richness) have been analysed together? If the model mentioned is accounting for this, it needs to be stated in the figure legend.

Fig 2e,f: Thanks for looking specifically at the large variation in the middle of the plots in the rebuttal letter. However, may it be possible to add R^2 to the figure? Given so many data points a significant relationship can be expected but how much the relationship actually explains would be more informative to know in my opinion. It still looks like to me that the large variation in the middle is preventing any meaningful interpretation of trends.

Lines 468-471: This is an interesting finding/statement. Does it mean that changes in species composition are highly site-specific? Do the authors have any means for exploring this further? Do perhaps models with a nested design (random effects) pick up stronger signals with climate accounting for region specific idiosyncrasies? What may these local-scale specific factors be? Herbivory? Management practices? Fire? More discussion on this would be useful to add.

Line 485: That is an interesting statement. In general, it is believed that the warmer-edge boundary of a species is due to competition and not to physiological constraints. Can the authors reach any more specific conclusions in this direction? This paper is perhaps an interesting read in this context: Paquette, A. and Hargreaves, A. L. 2021. Biotic interactions are more often important at species' warm versus cool range edges. – *Ecol. Lett.* 24: 2427–2438.

Line 493: This would be an important finding. Would it be possible to test if especially rare species are those constituting the losses the authors found? Given that the dataset would allow testing this it feels a bit shallow just speculating here.

Line 507: well, if species richness is decreasing species must be lost. Maybe delete this last statement?

Line 508: What does "Lower species richness has been observed with greater shrub cover spatially" mean?

Line 511: This is a strong conclusion but unsupported. So far, the authors didn't test if less competitive species are being lost.

Line 519: Or better resistance against herbivory?

Line 542: Could you support this statement with data or at least a reference?

Line 547: This statement comes as a surprise. Was the main conclusion of this piece that the arctic is resistant to warming?

Line 576: Note that this is the key prediction of MacArthur and Wilson's Equilibrium Theory of Island Biogeography. There is a huge body of literature linked to that hypothesis and why there is no change in richness despite assumed community turnover. Maybe there are some further clues and ideas in the island literature why arctic communities seemingly didn't respond to warming possibly also useful for this study?

Line 620 and line 637: Did the authors test for additional "plot size correction methods"? Comparing richness of differently sized plots, even at such small spatial differences, can introduce huge variation in itself. Rarefaction could be a solution or testing if same results were obtained if only including 1m² plots for instance.

Referee #3

(Remarks to the Author)

Plant diversity dynamics over space and time in a warming Arctic

I was Referee #3 in the previous round.

For the most part I think the authors have responded strongly to comments and issues identified by all reviewers. Unfortunately, I found the methods remained very hard to parse. And, if I understand the analysis correctly, I remain puzzled by some of the choices made, and find others to be without justification (and approaching untenable). Due to my continued uncertainty of the analyses, I will step through each in turn, before making some suggestions on other aspects of the presentation that the authors might consider.

Spatial models: in the rebuttal the authors show two relationships between richness and plot size: 1) $S \sim \text{plot size}$; 2) $S \sim \log(\text{plot size})$. However, neither of these relationships reflect the one assumed by the richness models fit by the authors, which with the log link, mean you are assuming $\log(S) \sim \log(\text{plot size})$. As I stated in my initial review, I think this is a reasonable assumption, but it does need to be stated clearly in the methods. The other spatial models fit to the evenness metric assumed beta error, and my comment regarding the assumption underlying the inclusion of $\log(\text{plot size})$ in my initial review remained unaddressed. The models with beta error and logit links amount to a logistic (S-shaped) functional form for the relationship between evenness and log plot size (i.e., $\text{logit}(\text{evenness}) \sim \log(\text{plot size})$). Again, this assumption is not stated clearly in the ms anywhere that I can see. And without further persuasion, I remain unconvinced that it is justifiable. The evenness results from the spatial models receive very little emphasis in the main text, and short of strong justification for either including them in their current form or modifying them somehow, I would recommend removing them. Unfortunately, for this bounded evenness metric no alternative solutions to including sample effort into a regression model come to mind.

Two time point models: Because plot size is consistent through time, I am puzzled as to why the authors chose to include it in these models. Again, it entails some unexplained assumptions regarding the relationship between metrics and sample effort (i.e., $\text{logit}(\text{dissimilarity}) \sim \log(\text{plot size})$, and $\text{logit}(\text{proportion}) \sim \log(\text{plot size})$). Including plot size in these regressions impacts the estimates of the other parameters, and theoretical reasons for its inclusion are not made clear (nor can I think of

any). In contrast, there are good reasons to think that richness and duration will be related to these measures of diversity change, and I like that the authors have included these in their analyses. A strong presentation of these analyses would make the reasoning for all inclusions (or omissions) clear in the methods, and report any sampling effects when describing the patterns found.

Temporal models: these were the hardest to understand, and some reconsideration of how the models are presented in methods and supplement are needed if readers are to easily grasp the workflow. Critically, it was not clear to me whether plot size was included in the models to estimate rates of change or not. Table S2 shows the 'Temporal models' all included plot size. But Table S3 shows the 'change over time models' referred to in the methods (L730-731), and does not include plot size. If Table S2 is showing the second-stage analysis, then the inclusion of plot size is less problematic, though I would still question whether the authors want to include it.

If I am understanding correctly, Table S3 shows the first stage models and should precede the temporal models currently detailed in Table S2; relatedly, greater clarity is needed in the methods to clearly delineate the first and second stage analyses. Any inclusion of plot size in models used to estimate the rate of change will impact the estimated rate of change, and I don't believe that is what the authors want. A model to estimate the rate of change without plot size estimates the rate of change assuming that the plot size is constant (which is a fair assumption for these data - it is constant through time). But when plot size is included, the estimated rate of change is adjusted for the plot size (which didn't change, so seems to be an unwanted, unnecessary adjustment). Similarly, when plot size is included in the second stage analysis it will impact the other coefficient estimates, and I am unaware of any theoretical reason to think multiplicative (proportional or log-scale) rates of richness change should be grain size dependent. The highly non-linear model for temporal changes in evenness (due to the logit-link) is even harder to form an expectation for any theoretical relationship with grain size.

Spatial beta-diversity: I was reassured to read that the analysis of spatial beta-diversity through time used plots of the same size (>99% of subsites). The authors also mentioned the variation among the spatial extent of the subsites (i.e., # of plots * size of plots), and they should confirm that the spatial extent of the plots within subsites (and the number of plots within subsites) was the same for each time point in the beta-diversity analysis.

Stepping out of the weeds of the analyses, which with some further clarifications, adjustments, and a greatly improved presentation, should prove to serve the authors aims well, I remain convinced that this work has the potential to make an important contribution.

L152: do you mean no directional local richness changes *on average*? Looks to me that some locations go up, others go down, resulting in no directional trend on average. Indeed, you report greater proportional species losses and richness declines in locations where tall shrubs have increased in cover. And conversely, show a relationship between changes in forb cover and richness changes that implies species richness increases in locations where forb cover increased.

Aside from my desire to see the authors clarify that they find no directional local richness trend *on average*, I think the authors do a good job of visualising and reporting uncertainty on their figures and throughout the manuscript.

L197: if only it were so simple! What about history and biogeography? E.g., Ricklefs & He 2016 PNAS.

L320: richness did not change *on average*. Figure 2c suggests that tens, possibly hundreds, of plots had changes $\geq 20\%$ of their species (i.e., slopes of $|0.2|$ or greater) per year.

L362: not sure how to parse changes in species abundance in relation to figure 3. Do you mean changes in species relative abundance? Or in light of figure 3, would referring to changes in species composition be most accurate?

L402-410: If I am reading Figures 4d-f correctly, they show the percentage of species being gained or lost. All results referring to these panels should describe proportions not numbers. All descriptions of these results should be carefully checked throughout.

L410-413: what is important to see here? For example, I expected negative relationships between duration and dissimilarity. But S11b shows the opposite for Bray-Curtis! Help readers by unpacking some interesting relationships (or non-relationships) here.

L460-461: is it accurate to refer to your turnover estimate as a rate? And again, I think you mean higher proportions of species lost or gained, not rates of species losses and gains.

L484: proportions of species gained or lost. This needs to be remedied throughout.

L490: this second reference to 'changes in species abundance' is no more illuminating than the first (L363). Do you mean changing composition by altered species relative abundances (as revealed by the Bray-Curtis analyses)?

L523-530: I don't see any evidence for resistance on Figure 2b, c. I see that plot-scale richness did not change on average (Fig. 2b), because the number of plots where richness went up were approximately balanced by other plots where it went down (Fig. 2c).

I appreciated the acknowledgement that smaller proportions of species gained and lost in more species rich plots is

expected by chance alone, However, it didn't silence my questioning the strength of the evidence for the resistance assertion. Indeed, I found the assertion of resistance weak and distracting.

Any link between spatial patterns (i.e., homogenisation and differentiation) and resistance implied by the subheading were not clear to this reader. I think the homogenisation/differentiation result is interesting, but if you want to link it to resistance, much more is needed. Why not simply discuss it on its own terms? Finally, the inclusion of homogenisation/differentiation here mixes scales unnecessarily.

L544: I'd assert you detected a lot of changes, just no change in richness *on average*.

L572: Here and throughout, I think you'd be well served to say you detected no directional trend in plot-level vascular plant species richness *on average*.

L582-585: as stated earlier, I don't see any development of a strong conceptual link between no change in spatial beta-diversity and resistance. Resistance is first mentioned in the discussion (L464), remains vaguely defined, and in addition to the undefined link with no change in spatial beta-diversity, appears to be additionally inferred by an empirical pattern expected solely by chance (i.e., proportionally fewer species gained and lost in more diverse plots).

L683-685: This seems like sound reasoning, though it would be unwise to think (or imply) that these metrics are independent of each other, e.g., I expect richness and the chosen evenness metric are (strongly?) correlated for these data.

L706-708: First, why convert to proportions? And were all proportions calculated using the same total number of species in both time points combined. It could be argued that the proportion of colonising species (i.e., species gained) should be a proportion of the species in the second time point only (i.e., divided by the total number of species in the second sample). And conversely, the proportion of species lost would be a proportion of species in the initial sample (i.e., the total number of species in the first sample). More rationale and detail is required here.

L719: you don't report any evenness results from the spatial models in the main text. And given the (from my perspective) problematic relationship assumed with plot size, I recommend omitting this analysis.

L724-734: I found this description of the temporal models (in particular), and all the analyses more generally, very difficult to follow. And the lack of clarity means I still don't know whether plot size was in the models used to estimate the rate of change. If I understand correctly, the temporal models shown in Table S2 do not have "multiple values over time" (L725), and instead show the second stage analysis of the rates of change previously estimated. This paragraph appears to mix single- and two-step analyses all together, and caused considerable confusion and frustration for this reader. Are the first stage models fit to estimate rates of change described under the next sub-heading "additional models"?

The structure of methods needs an overhaul. Currently there are multiple "model" sections. And the multi-section description of (the many) models is separate from the specification of their response distributions and link functions. This means that the reader has to identify which of the numerous models or response metrics previously (or yet to be [ordination and posthoc analyses]) described would have normally distributed response, etc. For example, the general statement of what the negative binomial is used for (count data with variance > mean) doesn't much help the reader link it to your analyses of species richness. Most importantly, the logical flow of your analyses is broken; for example, "I think" the clearest description of first stage analyses quantifying rates of change in richness and evenness come after the second stage models are described.

Fig. 2a: this is very hard to read, and the caption does not help much. It appears that these results are unrelated to the analyses, and it is hard to know exactly what is being shown. E.g., what is magnitude of change? Proportional change or something else? You state in the caption that the average changes reported are for visualisation only, but the citations to this figure report and infer patterns. Moreover, without latitudinal contours on the map, it was hard for me to see the patterns the authors allude to.

Fig. 2d: do these subsite level estimates come from the same model shown on Fig. 2b, c?

Fig. 2e, f: A reader has no way of knowing where these estimates of richness change come from at this point. Are these the plot-level estimates (i.e., individual slopes aggregated on 2c), or subsite level estimates (as per 2d)? If yes, why did the units change? If no, where did they come from?

Referee #4

(Remarks to the Author)

The authors report results from a large study of plant diversity observations, undertaken across the North American and European Arctic. Many of these observations represent a time series at specific locations, while others are from single points in time. The authors perform an impressive set of statistical analyses to assess if changes in plant diversity can be identified, if any changes have a particular direction and they relate these changes to climate in the Arctic. As plant biodiversity is not my specialist subject area, this review confines itself to assessing the climate part of the paper.

Overall, and as a non-biology specialist, I found this a very interesting and informative study to read, particularly as the observations are widely spread spatially (the lack of Russian observations, for presumably obvious reasons is to be regretted, but not a reason to reject this study) and with a temporal component. Given the widespread interest in climate impacts in the Arctic, I find this a helpful contribution with the focus on ecosystems.

Regarding the climate analysis, the CHELSA dataset could be considered a slightly unusual choice in the climate community, but it is well established in the climate impacts community. Given the importance of well-resolved topography, and the lack of detailed observation networks in the Arctic, it is also a good compromise to use as a homogenous dataset especially as the main basis for CHELSA in this region is the ERA-5 reanalysis, which is well documented in the Arctic, though it does also include a well-known but small warm bias. The coming CARRA-2 high resolution reanalysis will be an even better choice in the future but is not yet available. I have two further points that I think should be included.

1) The authors assess temperature and precipitation changes in the paper, but one aspect missing that I consider may be important is the impact of the snow season – that is both any systematic changes in snow depth as well as how long snow lies on the ground at each site. There are well-documented trends in snow season length and variability, specifically a shortening of the season due to earlier melt onset but also a later start in some regions. Given the importance of snow cover on ground temperatures, I would be surprised if this was not also significant for vegetation dynamics. As snow depth on the ground and ground temperatures are often subject to systematic biases in model products, I would suggest the authors look at the satellite-derived datasets as well as in-situ snow depth observations, rather than relying on model products for this data. There are a number of high resolution data products that would be relevant here including the ESA CCI snow products. Local temperature change gradients should also include this effect, but I am uncertain if it is included in CHELSA and the documentation does not mention it.

2) Much of the recorded warming in the Arctic occurs during the winter season. In the summer, increasing temperatures have been partly dampened by enhanced ice melt of both sea ice and glaciers. It is probably worth mentioning this effect in the discussion when considering how changes in plant diversity will develop in the future as well as why there is not as strong signal at present as might be expected when considering annual rates of warming compared to warmest quarter warming rates. I am unsure if the plant data has sufficient granularity to be able to compare sites close to melting ice and sites further away, but the existing analysis does also include different rates of warming at observation sites. I think it likely these differential warming rates are partly related to proximity to local ice sources, so I do not consider it necessary to do extra analysis on this latter point.

Version 2:

Reviewer comments:

Referee #2

(Remarks to the Author)

I would like to thank the authors for their thorough revision that addressed all my comments.

Referee #3

(Remarks to the Author)

Plant diversity dynamics over space and time in a warming Arctic

I think the authors have done a comprehensive job responding to the reviewer comments, and find the manuscript much improved. With a bit of effort, an interested reader should now be able to follow, and glean the rationale, for the analytical decisions and methods.

I've only minor comments and queries that the authors might consider as they prepare this work for publication. Congratulations to all involved in this important contribution to the biodiversity change literature.

Line numbers refer to the clean (i.e., without tracked changes) version.

L272: do you really need an acronym for Research Questions?

L281: does this treeline idea come up again? I'm a bit confused by the exact meaning of treeline in this context, and if it comes up again, I missed it if it did. Consider removing.

L295: I think you need something extra here for this to make sense, e.g., proportional gains of low occupancy species. Note also that dominant here needs to refer to high occupancy (as opposed to local relative abundance or competitive superiority).

L382: would this be more accurate if you said nearly all plots experienced changes in species composition due to altered relative abundances (Bray-Curtis), and 59% showed compositional changes due to species gains and losses (Jaccard)? I'm not sure the reference to Jaccard makes sense as phrased currently.

L398-400: this loss of low occupancy species is a little inconsistent with the finding of no biotic homogenisation reported below.

L450: “associated with” is probably more appropriate here than “driven by”

L481-482: I'd urge a little bit of caution regarding richness changes at larger scales here. If I understand correctly, you've only looked at local (indeed, quadrat) scale changes (with the exception of the spatial beta-diversity, which does not consider richness). Too many biodiversity scientists continue to assume that changes at one (smaller) scale will be consistent at another (larger) scale.

L495: not sure about despite in this sentence, nor of the word choice “assemblies” - perhaps assemblages, or just delete: “... composition changed based on local context, and both climate...”. More generally, the presentation would benefit from an edit with a focus on making the writing more direct.

L511-514: this sentence reads awkwardly to me, and I'm not sure of the intended meaning. Can you rephrase to more directly say what you are trying to say?

L520-522: some logic is needed to underpin this assertion that composition changes might eventually drive richness changes (e.g., altered biotic interactions resulting greater competitive exclusion). At the moment it reads as contradictory (and confusing for this reader at least) given the preceding discussion of your results (gains and losses balanced, consistent with equilibrium theory). Similarly, (L522-523), why might compositional changes result in further reshuffling?

L560, 562: word choice: is amidst better than among here?

Fig. 2b: Change Years to Year on the x-axis.

Referee #4

(Remarks to the Author)

2nd Review of Criado et al., Plant Diversity Dynamics over space and time in a warming Arctic

This is my second review of the Criado et al paper and although again I will confine my remarks to my area of expertise, namely Arctic climate, I would like to note that I found the revised paper again interesting and I think better written than the first version. As a non-biologist with a keen interest in the Arctic, I found it a very enlightening read.

The additional analyses done by the authors on the role of snow is very convincing, it is obviously disappointing that the EO datasets are not yet up to scratch for this kind of analysis and I will take this as feedback to the relevant CCI group.

Nevertheless, the ARCLIM dataset is likely a good substitute and the results of this analysis seem to show fairly convincingly that the effects of climate change on snow cover and duration are still small. This may also perhaps play into the extinction lag that the authors mention elsewhere. I do also think the observation of scale effects is important and I am therefore happy that the additional paragraph on this analysis is sufficient to show that snow has been taken into account.

Plant diversity dynamics over space and time in a warming Arctic

Response to Reviewers

Response: Thank you for assessing our manuscript for publication at Nature and giving us the opportunity to revise our work. We have implemented the reviewers' feedback, addressed the novelty concerns, and better explained the drivers of plant diversity change.

As a summary, we have undertaken the following main revisions:

1. We have re-made the main manuscript figures and paired down from 5 to 4 to focus on the main messages (**Figure 1-4**). We have re-organised the subplots to ensure that they all thematically align better with each other, and included a few additional figures and tables to support our results (**Figure 1d, 2b, 2e, S4, S8, S9, S11, S14, Table S2, S7**). We have also updated our figures to reflect the variability inherent to random effects in model predictions, and have differentiated 'significant effects' (i.e., those whose credible intervals don't overlap zero) from 'non-significant effects' (i.e., those whose credible intervals overlap zero) by adding solid and dashed lines in figures, respectively.
2. We have re-run all temporal analyses by including a filter of 5 years duration for the plots in our dataset (see response **3.27** for a detailed description of this decision), in order to ensure that extreme stochastic values of change did not skew our results.
3. We have investigated the relationship between the two main drivers of diversity change: temperature increases and shrub cover change, in order to help untangle the complex processes undergone by these Arctic plant communities (**Figure S14, Table S7**).
4. We have provided more detail on the mechanistic pathways of diversity change, and explicitly address how our results of observed patterns of Arctic diversity change are linked to these pathways.
5. We have emphasised the novelty of our study throughout the Introduction and Discussion sections. We have also clarified our hypotheses and the reasoning behind our predictions.
6. We have clarified our Methods throughout, included additional detail on our model and data link choices, and included **Table S2** with full model structures and summary statistics, as a more detailed complement to **Table S1**.
7. We have implemented the concept of spatial and temporal scale in a more clear manner throughout the manuscript, and have ensured that it's always clear at which scale we are referring to.
8. We have performed additional analyses as requested by the reviewers (e.g., interaction between temperature and temperature change, effects of functional group change on richness excluding extreme values of change, etc.)

Our new findings indicate:

1. Richness change was not directly related to warming temperatures, though species gains and losses were greater where temperatures had increased the most. Since species gains and losses were similar across plots, this can result in the lack of direction in richness change over time that we observed.
2. Richness decreased over time where shrub cover increased over time, a relationship that was particularly driven by an increase in tall shrubs and their competitive effects over other species groups.
3. Shrubs cover change rates were not directly related to rates of temperature increase, but shrub cover was broadly sensitive to climate, with dwarf shrubs responding negatively while non-dwarf shrubs responded positively to warming

We believe that the manuscript is greatly improved as a result and can robustly show the unique trajectory of plant biodiversity change at a time of heightened warming. Our responses to the reviewers are below in blue, and the revisions in the manuscript and in the

Supplementary Information file are in red. Line numbers below refer to the version of the manuscript without track changes.

Editor comments

I should note that we were in the process of assigning a referee to assess your climate analysis (as you rely on a single climate dataset, and there may be non-trivial differences among datasets), but were unable to do so prior to obtaining the three reports enclosed here.

For any potential future review of the climate data/analyses, we have used the CHELSA dataset (version 1.2.1) which is composed of interpolated meteorological station data. This is similar to the Climate Research Unit dataset, but with a higher resolution pixel size of 1 km x 1 km instead of 0.5 degrees x 0.5 degrees. All interpolated gridded climate data are well correlated with each other as they are based on mostly the same underlying data, so switching datasets should not produce any particular differences among the climate timeseries or climatology data, or in any analyses using those data. The limitations of Arctic climate data are the records themselves and not the interpolation methods; however, the CHELSA algorithm captures more topographical influences on temperature than the CRU and other interpolation algorithms.

Karger, D. N. *et al.* Climatologies at high resolution for the earth's land surface areas. *Sci. Data* 4:170122 doi: 10.1038/sdata.2017.122 (2017)

<https://www.nature.com/articles/sdata2017122>

Karger et al. 2017 summarises their interpolation algorithm in the abstract and explains in detail in the paper:

“The temperature algorithm is based on statistical downscaling of atmospheric temperatures. The precipitation algorithm incorporates orographic predictors including wind fields, valley exposition, and boundary layer height, with a subsequent bias correction. The resulting data consist of a monthly temperature and precipitation climatology for the years 1979–2013.”

In Karger et al. 2017, a comparison is made between precipitation among gridded climate datasets including the CHELSA, CRU, WorldClim, ERA and GPCC datasets (see Fig. 2 from the paper below). They summarise this global comparison:

“Among the models which use GHCN in their algorithm, CHELSA shows the highest fit between stations and predicted precipitation, with WorldClim, GPCC and CRU showing smaller, but still high fits with the station data (Fig. 2).”

However, they go on to say:

“A statistical comparison with different datasets is complicated by the fact that most gridded temperature and precipitation datasets are parameterized using similar observational data, leading to generally high correlations between climatic reanalyses. To validate the results of the CHELSA algorithm, we identified several independent datasets of various size and temporal extents.”

Karger et al. 2017 include a comparison with met station data for Nordic countries including area within the Arctic. Karger et al. 2017 summarise the results with the following sentences:

“CHELSA, WorldClim, and CRU climatologies fit the Nordklim data well, with CHELSA performing better than the other models, despite the fact that Nordklim data are included in WorldClim but not in CHELSA (Table 3). ERA-Interim and CHPclim do not perform well in this region, which is probably due to the larger errors of remote sensing data in arctic regions, on which both models depend.”

The paper summarises the differences among gridded climate datasets with the following paragraph:

“The validation results in general show that including orographic effects can improve existing climatologies and reanalysis to a degree that the derived analysis (here the SDMs) show increasing accuracies. While CHELSA is an improvement over existing very high-resolution climatologies, it still exhibits errors which we quantified in several ways. The validation of main correction step in the algorithm that includes the orographic wind effects and boundary layer shows that the precipitation at the stations is better captured after the correction than before the downscaling to 30 arc sec resolution. The improvement varies by region and month with the majority of months showing an improvement. Most importantly, the better prediction with regard to SDMs in which precipitation and temperature data are used already indicates that CHELSA might be a substantial improvement over existing products which are currently being employed for such purposes.”

Finally, we have also included mention to our decision to choose CHELSA as our climate dataset in the Methods section:

“We chose CHELSA as the source for our climate data because, as a quasi-mechanical statistical downscaling product, it has a very fine grain size (1x1 km) and has been shown to outperform other interpolation-based climate products, and particularly to perform better predicting precipitation patterns^{84,88,89}.” (L676-680)

Referee #1 (Remarks to the Author):

1.1. The manuscript entitled “Plant diversity dynamics over space and time in a warming Arctic” by Garcia Criado et al. describes how vascular plant species richness and composition have changed over time, and it aims to identify the geographic, climatic, and biotic drivers behind such changes. I was quite pleased to read this manuscript, which is concise, straightforward, and very well written.

Response: Thank you so much for your assessment of our manuscript. We are very pleased with your positive feedback.

1.2. At first sight, the results obtained in this study with such a large sampling effort, seem very unspectacular. For example: no directional trend in changes in species richness across the Arctic. Or: no biome-wide homogenization across the Arctic (the plots shifted in their composition in all possible directions), etc. Among such results, the detection that shrubification out-compete other groups stands out for a reader with finally a clear statement (line 496-498).

However, in my opinion, this is the strength of this study. It confirms the complexity of the Arctic ecosystems, and the impressive resistance of Arctic plant communities to climatic and global changes. This is also highlighted by authors, by enumerating (from line 527) other processes that could contribute to a lack of detected diversity change.

Response: We are very happy that the reviewer found our communication of the underlying complexity and resistance of Arctic ecosystems as a strength of our study. We have revised the manuscript focusing on shrubification as a driver of Arctic plant biodiversity change, as highlighted by Reviewer 1, by adding new post-hoc analyses to further explore how shrub changes are related to warming temperatures (**Table S7**). We now demonstrate that shrub cover change is not directly related to latitude nor to warming rates (**Figure S14a, c**), but shrub cover is broadly sensitive to climate (**Figure S14b**), with dwarf shrubs responding negatively and non-dwarf shrubs responding positively to warming. Additionally, shrub cover increases are related to overall richness decreases, a trend which is particularly driven by tall shrubs (**Figure 2e**).

1.3. The study is very solid, based on more than 42,000 records, covering nearly 500 vascular plant species, more than 2,000 plots in 45 study areas across the Arctic (obtained from the International Tundra Experiment (ITEX+ database).

Response: Thank you. We agree and believe this newly expanded Arctic plant composition database provides an excellent compilation of data to test questions about Arctic plant diversity change.

1.4. However, several statements, such as the impressive numbers listed above, in the Abstract, could be misinterpreted. For example, “only” 64% of study plots were monitored at least twice, and thus only those could be use both for spatial and temporal analyses. And the time of monitoring was never 40 years (at average 8 years period). Thus, the statement of “four decades” is somewhat misleading and should be adapted/explained accordingly (e.g. line 136). Similarly, the authors are often using the term “across the Arctic”... although large part of the Arctic is not covered by the study (notably Russian Arctic, see Fig. 1, Fig. S2, etc.).

Response: We have revised the text to better communicate the parameters of the dataset and our analyses. We have made revisions throughout the manuscript to improve this precision, and we have ensured that we refer to “different intervals during the past four decades”. Specifically, we have also included the newly made **Figure 1d** to better clarify the temporal extent of our database, and rephrased the Abstract as:

“we quantified temporal changes in species richness and composition from repeat surveys conducted over different intervals between 1981 and 2022” (L148-149).

We have also revised the text to refer to the terrestrial Arctic and Arctic vascular plants rather than just using the word Arctic. While we fully acknowledge geographical gaps in the database, we prefer to retain the phrasing ‘across the Arctic’ in certain parts of the text as a general term, in order to emphasize that the patterns and trends we present are representative of many different Arctic locations spread across a broad geographic (6 countries) and climatic area (sub-Arctic to High Arctic; **Figure 1a, 2b**). Critically, our study sites incorporate the full spread of climate space experienced across the Arctic (**Figure 1b**), and climate space could be viewed as being more or equally important to represent than geographic space.

Unfortunately, currently available Russian data in the ITEX+ dataset did not meet the inclusion criteria (i.e., plot size $\leq 1\text{m}^2$, $> 60^\circ$ latitude, consistent sampling methods and plot sizes over time) to be retained for our analyses. We have reached out to Russian collaborators in the past, but as of yet there have not been further implementation of ITEX protocols at other Russian sites. There likely will not be any datasets collected in Russia in the near future that will be shared with our authorship team due to the current geopolitical situation. However, taken together, our study encompasses a lot of variation across Arctic sites and thus is relatively representative of the biome, with data gaps that are inevitable in any large-scale study.

1.5. The manuscript references previous literature appropriately. Eventually, two papers published very recently could be included:

Körner C. 2023. Concepts in Alpine plant ecology. *Plants* 12: 2666.

Zhang et al. 2023. Evolutionary history of the Arctic flora. *Nature Communications* 14: 4021.

Response: Thank you for providing these references. We have now included them both in the manuscript:

This expectation is further supported by observed climate-induced increases in vascular plant species richness across European mountain tops^{29,30}, whose elevational gradients mirror Arctic latitudinal climatic and richness gradients³¹. (L201-204)

“We defined biogeographic regions as Eurasia, Greenland-Iceland, Eastern North America and Western North America according to glaciation history⁸⁰⁻⁸².” (L633-635)

1.6. Concluding, the manuscript is very well written, all analyses and graphics, tables resulting from them, as well as the supplementary materials are of excellent quality. I find that the conclusions and data interpretation are robust, valid and reliable. The results are of large interest to many researchers from several disciplines.

Response: Thank you, we really appreciate the reviewer’s enthusiasm and feedback on our work.

Referee #2 (Remarks to the Author):

2.1. In their manuscript “Plant diversity dynamics over space and time in a warming Arctic” the authors address the question how plant species composition and diversity changed over time and why in a warming arctic using resurveyed plots ranging from Europe to Canada (excl Russia). They found no directional changes of plant species composition in response to climate. However, species richness at plot level increased most in plots with higher temperature and shrub encroachment leading to higher species losses and shifts in species abundances.

The manuscript touches an interesting and important topic; that is how vegetation is expected to change with progressing climate change. Also, the data is interesting allowing for such kind of analyses. The results were mixed but in general do mirror those obtained from other, similar studies from different regions. While this is interesting in itself, I found the manuscript lacked novelty in the sense to provide new and mechanistic insights into changes of vegetation over time.

Response: Thank you so much for your assessment of our manuscript, we are glad that you found it of interest.

We have now taken the time to clarify our hypotheses and predictions and to put them into the context of the results of our analysis to better highlight new findings and mechanistic insights into the drivers of vegetation change over time. Specifically, we have conducted numerous new analyses, generating a series of new findings (see revisions summary above).

Our study focuses on identifying the plant diversity changes and is an important first step to fully explore the climatic and biotic drivers of the observed plant diversity change. However, we have better highlighted the possible pathways of Arctic tundra responses to warming. We now describe discuss four pathways that could be leading to Arctic plant diversity change, including: 1) northward species migration, 2) colonisations from local species pools, 3) loss of cold-adapted species in favour of thermophilous (warm-loving) species, and 4) increased plant competition, that we hypothesise could be contributing to observed plant diversity change. This context is set out in the Introduction as follows:

“Arctic plant diversity change could be shaped by interacting processes following four pathways. 1) If species migrate northward with warming, we would expect a net increase in overall Arctic plant species richness^{2,18,19}. 2) Richness increases could also result from short-distance dispersal and colonization from species that are already present in local species pools²⁰. 3) Conversely, reduced Arctic floral diversity could result from losses of cold-adapted species²¹ that cannot cope with warming temperatures²². 4) These declines could be exacerbated by increased competition with colonising species originating from Low Arctic and boreal latitudes^{23,24}. Because these pathways may be acting in concert, it is possible that richness increases and decreases could occur simultaneously, resulting in no net richness change. Yet, the effects of these different pathways on current and future Arctic plant diversity trends remain poorly understood.” (L180-191)

These pathways could be affecting Arctic vascular plant communities at different scales and in different directions in different regions, but here we show that colonisations from local species pools, increased competition, and losses of cold-adapted species seem to be occurring. We have now weaved this narrative of complementary drivers and scales of influence throughout the Discussion as follows:

“Richness change was not greater towards southern Arctic edges (**Figure 2a**), where we hypothesised that northward migration from the boreal forest (i.e., borealization) might be a major driver of change. Instead, this lack of latitudinal change might indicate that, where diversity is changing, one of the main sources is colonisations by species present in local species pools that have not yet been recorded in long-term monitoring plots (referred to as ‘landscape’ or ‘dark’ diversity)^{20,54}. Species richness increases were not greater at sites with greater rates of warming over time (**Figure 2d**), but warming was associated with greater species gains and losses (**Figure 3e**), possibly due to cold-adapted species not coping with warming, and to warm-adapted species expanding to warmer areas and further displacing cold-adapted species^{52,55}.” (L477-486)

“Where shrub cover increased over time, plots experienced decreases in species richness, community evenness, and greater species losses (**Figure 2e, 3f, Table S1**). Lower species richness has been observed with greater shrub cover spatially, with shading and litter production leading to decreases in sun-loving plants under shrub canopies^{23,24}. Our Arctic-wide results corroborate site-level reports that increasing shrub cover over time may lead to less diverse plant communities and the displacement of less competitive species^{41,57,58}. Thus, Arctic diversity might be more at risk at sites with increasing shrub cover, particularly from tall shrubs (**Figure 2e**).” (L505-513)

“Overall, our findings indicate that likely pathways of Arctic plant diversity change include colonisations from local species pools²⁰, losses of cold-adapted species⁵⁵, gains of thermophilous species⁵² and increased competition with canopy-forming shrubs²³.” (L584-587)

We believe that through these revisions, we have now better highlighted the novel findings of this study and communicated the predicted mechanisms of Arctic plant biodiversity change, while specifically testing the contrasting influences of warming and shrubification.

2.2. While the authors incorporated many different diversity indices into their analyses it is not clear what they expect from this and why (please also see my comments on the introduction).

Response: We have undertaken major revisions on the Introduction and better spelled out the hypotheses that we considered for the different diversity metrics in this work. We have chosen to analyse common biodiversity metrics that capture species richness, species evenness and species compositional change. We have avoided any metrics that combine aspects of biodiversity into one aggregate index such as Shannon – Weiner or Simpsons, since we were interested in analysing the specific elements of biodiversity in isolation from each other. Taken together, with our revisions, we believe we have provided a clearer justification for our predictions of biodiversity change and the metrics that we have used in our biodiversity change analyses and the mechanisms driving the changes that we have seen.

We have added further clarification on our choice of metrics in the Methods section too:

“**Biodiversity metrics.** We chose to analyse common biodiversity metrics that capture species diversity, dominance, and composition change, rather than composite indices, in order to understand the specific elements of biodiversity in isolation from each other.” (L682-685)

2.3. More explanation and insights into their choice of indices could have helped to strengthen the manuscript. Other studies for instance (e.g., Jandt et al Nature 2022; not cited) showed clear trends in winners and losers at plot scale over time and hypothesised that this may be linked to

habitat types. Such could have been incorporated here. It seemed to me that the very generalistic approach chosen here (e.g., a lot of indices were correlated with many factors without formulating clear hypotheses why they matter and how) prevented insights into the real drivers of biodiversity change in the arctic.

Response: We have revised the manuscript throughout to better provide mechanistic insights into changes of vegetation over time (see 2.1 above). Specifically, we have better linked the possible pathways of diversity change to our results (see above). We have also included clearer hypotheses on what exactly we expected to find and why, and contrasted that with what we actually found.

We have not explored habitat types per se (as Jandt et al. 2022 did) as that was not one of our a priori research questions and because we felt that there were other more novel elements to focus on in this first study of this newly-compiled database. There are different habitats within the database, but there is some correspondence between latitude, floristic regions and habitats that complicate analyses of habitat type alone. However, we will certainly take this on board for upcoming analyses with this newly compiled dataset.

We have performed a few additional analyses to better understand the relationship of the two main drivers of richness change: temperature change and shrub cover change (see Figure below, now **Figure S14** in manuscript). Shrub cover change was not related to latitude nor rates of temperature increases, but shrub cover was broadly sensitive to climate.

Figure S14. Relationships between main drivers of diversity change (posthoc analyses), with model outputs in **Table S7**. **a)** Shrub cover change was not related to latitude. **b)** Shrub cover was sensitive to the mean MTWQ of the previous five years to the shrub cover measurement: non-dwarf shrub cover was greater at warmer temperatures, and dwarf shrub cover was lower at warmer temperatures. Mean temperatures are centred per subsite to account for variability and enable model convergence. **c)** Shrub cover change rates per plot were not related to temperature change rates over the 1987 - 2013 period. Lines represent predicted model fits and bands show the 95% credible intervals. Dashed lines indicate models whose credible intervals overlapped zero, and solid lines show models whose credible intervals did not overlap zero.

We refer to this new analysis in the Results section:

“Shrub cover was sensitive to temperature. Both warming (**Figure 3b, 3e**) and shrubification (**Figure 2e, 3f**) emerged as two main drivers of Arctic plant diversity change. We therefore conducted additional analyses to better understand how and where these drivers interact (**Table S7**). Overall, shrub cover did not increase significantly over time in our dataset (**Table S8, S9**). Shrub cover change was not associated with latitude (**Figure S14a**), and the

rate of long-term warming was not related to the rate of shrub cover change over time (**Figure S14c**). However, interannual variation in shrub cover was sensitive to temperature, indicating that dwarf shrubs respond negatively while non-dwarf shrubs respond positively to warming (**Figure S14b**)." (L415-423)

And in the Discussion:

"Shrub cover change has been widely linked to warming in previous site-level studies²⁻⁴. However, we did not find clear evidence for greater shrub change with greater rates of warming (**Figure S14c**), in agreement with previous pan-Arctic studies⁴. Instead, shrub cover was sensitive to temperature, with non-dwarf shrub cover increasing and dwarf shrub cover decreasing with warmer temperatures (**Figure S14b**)." (L500-505)

And in the Methods:

"Posthoc analyses. In order to understand the relationship between two of the main drivers of diversity change, shrub cover change and warming over time, we performed extra analyses (**Figure S14, Table S7**), given that previous literature suggests a positive relationship between them^{4,94}. First, we modelled shrub increases as a function of latitude, with a subsite random effect (**Figure S14a**). To identify whether shrubs exhibited sensitivity to temperature, we calculated the mean temperature of the past five years for each monitoring time point (**Figure S14b**). We centred temperatures per subsite prior to analyses in order to standardise magnitudes across regions and to enable model convergence. We modelled shrub cover at each time point as a function of mean temperature of the past five years, with a nested random effect structure of plot within subsite, and an interaction term of shrub type (dwarf versus non-dwarf). Additionally, we modelled shrub cover change per plot as a function of long-term temperature change (over the 1978 - 2013 period), with a random effect of subsite and an interactive term of shrub type (**Figure S14c**). To assign shrub categories, we followed the methodology from Garcia Criado et al.⁹⁵ and categorised shrubs as dwarf and non-dwarf (including low and tall shrubs), since we were interested in the ecological effects of species sprawling versus erect physiognomy." (L806-823)

More specific comments:

2.4. The Introduction was hard to follow. Many different concepts and ideas were mixed preventing a logical flow leading to the aims of this study.

Response: We have restructured the introduction to provide a better context for our study findings and to build the evidence for our hypothesised geographic, climatic and biotic drivers of change. We believe that the Introduction section is now stronger and the writing better communicates the predictions of what mechanisms could influence plant biodiversity change (see revision **2.1**), while also highlighting the big picture context for our study.

2.5. For instance, mechanistic explanation how and why species diversity in the arctic may change over time is only loosely touched or missing. The authors mention several processes but do not explain these in detail. This is problematic as it does not equip the reader with the knowledge needed to understand the difficult and complex results. For instance, in lines 161 to 172 the authors mention that species can be gained as well as lost in the arctic due to northward migration and extinction of cold-adapted species. However, this is poorly explained how.

Why do species go extinct? Is it due to increased competition or to eradication of their psychological climate niche? Also which species can be expected to migrate northwards? Those with good dispersal ability or those being highly competitive (maybe shrubs)? Further, I think that this is highly contingent on scale – which is also mixed in the intro (often it was not

clear whether a specific pattern was expected at local, regional or whole-arctic scale). Can we expect different patterns at plot, regional and whole-arctic scale?

Response: We agree that our hypothesised mechanisms explaining how Arctic plant diversity change were not previously clear. We have now provided a detailed explanation of our three hypothesised geographic, climatic and biotic drivers of change that we derive from literature evidence (see revision **2.1** above). We have then put our findings into the context of these hypothesised mechanisms in the discussion. We believe we have now greatly improved the clarity of the hypotheses and how our results do or do not support these hypotheses and what the implications are for a mechanistic understanding of the drivers of Arctic plant diversity change (see **2.1** above).

2.6. This all leads to poorly introduced research aims and questions.

Response: We have restructured the Introduction and its corresponding research aims and questions, which we now believe are better matched in the Results and Discussion section, with a clearer understanding of the different pathways of diversity change through the text.

2.7. The results looked interesting but it the choice of figures shown in the main text was difficult to apprehend.

Response: We have re-made the main manuscript figures and paired down from 5 to 4 to focus on the main messages (**Figure 1-4**). We have re-organised the subplots to ensure that they all thematically align better with each other, and included a few additional figures and tables to support our results (**Figure 1d, 2b, 2e, S4, S8, S9, S11, S14, Table S2, S7**). We have also updated our figures to reflect the variability inherent to random effects in model predictions, and have differentiated 'significant effects' (i.e., those whose credible intervals don't overlap zero) from 'non-significant effects' by adding solid and dashed lines in figures, respectively. We have also made substantial modifications in most figures in order to ensure that the aesthetics appropriately communicate our main messages.

2.8. Why mean temp of the warmest quarter as an explanatory variable? In the intro the authors mentioned that warming winters may lead to pronounced changes (line 209). This needs better justification.

Response: We extracted mean annual temperature, mean temperature of the warmest quarter and mean temperature of the coldest quarter from CHELSA, and found that they were correlated with each other. We chose temperature of the warmest quarter as our temperature fixed effect as it best represents the short growing season conditions experienced in Arctic ecosystems, and has been previously correlated with plant growth rates, biomass and reproductive rates, which are all relevant variables driving diversity change.

This is now better reflected in the Methods section:

"We extracted, at the subsite level, data from long-term climatologies at CHELSA (version 1.2.1)⁸⁴ including mean annual temperature, mean temperature of the warmest quarter (MTWQ) per year, mean temperature of the coldest quarter (MTCQ) per year and mean annual precipitation (MAP, hereafter 'precipitation') for the period 1979 - 2013. Upon examining correlations between the three temperature variables, we found that most were correlated with each other. Thus, for our temperature variable we used only MTWQ (hereafter 'temperature') as it best represents the growing season conditions and has been previously linked to plant biomass, growth and reproductive rates⁸⁵⁻⁸⁷, which are in turn relevant variables driving diversity change." (L666-674)

2.9. Summary statistics in the regression plots would be helpful.

Response: We have now added **Table S2** (Supplementary Information) with the model structure and summary statistics of all models. Variables highlighted in bold are those whose credible intervals don't overlap zero and can thus be considered as 'significant'. In the interest of keeping the figures less busy, we refer in the figure caption (and throughout the text) to the model number that belongs to each figure (or statement), with its corresponding summary statistics.

2.10. Further minor comments:

2.11. • Line 131: Referenced abstract needed – please provide. Please also note that max of 50 refs are allowed in the main text.

Response: We have now included references in the abstract. We defer to the editor on whether there is flexibility in the number of allowed references in the main text.

2.12. • Line 158: unprecise: there are arctic ecosystems that are not structured by plants. Do you mean those that are ice-free and terrestrial?

Response: Thank you for this suggestion. We have rephrased this to refer to the terrestrial Arctic:

“Plants are the foundation of Arctic terrestrial food webs, the carbon cycle and the livelihoods of Arctic people. Thus, in order to understand climate change impacts on Arctic ecosystems, we must first quantify impacts on terrestrial plant communities.” (L175-178)

2.13. • Line 167: How? Some more mechanistic explanations on effects of poleward species movements could help to understand better the rationale of this study.

Response: We have edited this section to better reflect the alternative pathways and the processes that we envisage could be shaping Arctic ecosystems (see **2.1**). This paragraph now reads:

“Arctic plant diversity change could be shaped by interacting processes following four pathways. 1) If species migrate northward with warming, we would expect a net increase in overall Arctic plant species richness^{2,18,19}. 2) Richness increases could also result from short-distance dispersal and colonization from species that are already present in local species pools²⁰. 3) Conversely, reduced Arctic floral diversity could result from losses of cold-adapted species²¹ that cannot cope with warming temperatures²². 4) These declines could be exacerbated by increased competition with colonising species originating from Low Arctic and boreal latitudes^{23,24}. Because these pathways may be acting in concert, it is possible that richness increases and decreases could occur simultaneously, resulting in no net richness change. Yet, the effects of these different pathways on current and future Arctic plant diversity trends remain poorly understood.” (L180-191)

2.14. • Line 174: This is, of course, greatly scale dependent. It is often unclear which spatial scale is considered and discussed in the manuscript.

Response: We have rephrased this section to:

“Species richness patterns at large scales are broadly driven by climatic gradients²⁵. Many taxa exhibit a latitudinal gradient in diversity, whereby species richness is greater at lower latitudes, which are generally warmer^{26,27}. Thus, Arctic vascular plant richness is expected to increase over time as rapid warming^{1,28} leads to new, warmer thermal niches for warm-adapted species at northern latitudes.” (L197-201).

See also **3.27** below for a detailed response on how we considered spatial and temporal scales throughout the manuscript. We have also gone through the manuscript and clarified the scale considered throughout.

2.15. • Line 176: Also here some reference to the LDG could be helpful. With climate warming, what species richness could we expect where and why (for instance)?

Response: As per above (**2.14**), we have made the reference to the LDG more explicit here, and addressed how we would expect warm-adapted species to expand northwards:

“Species richness patterns at large scales are broadly driven by climatic gradients²⁵. Many taxa exhibit a latitudinal gradient in diversity, whereby species richness is greater at lower latitudes, which are generally warmer^{26,27}. Thus, Arctic vascular plant richness is expected to increase over time as rapid warming^{1,28} leads to new, warmer thermal niches for warm-adapted species at northern latitudes.” (L197-201).

2.16. • Line 180: To what extent? This is a very simplistic assumption and may need more nuanced explanation.

Response: We have rephrased this sentence to:

“This expectation is further supported by observed climate-induced increases in vascular plant species richness across European mountain tops^{29,30}, whose elevational gradients mirror Arctic latitudinal climatic and richness gradients³¹.” (L201-203)

2.17. • Line 183: Unclear. Why would one assume that boreal species colonise low arctic habitats if they are more dissimilar than those of the high arctic?

Response: We have rephrased this to:

“Spatially, we would expect plant richness to increase at warmer, lower Arctic latitudes because of the potential influx from the species-rich boreal forest (“borealisation”)³²⁻³⁴ and because the dissimilarity between Low Arctic and boreal flora is more pronounced than the dissimilarity between High and Low Arctic flora³⁵. Overall, we expect richness increases where more warming has occurred, and at lower latitudes closer to the boreal zone.” (L204-209)

2.18. • Lines 174 to 189: Can we expect a delay of species migrating northwards with climate? Not all species are equally well adapted to travel northwards mainly due to different dispersal strategies and biotic dependencies. There is broad literature on species colonising recently deglaciated areas, which show that some species are much faster in colonising new terrain than others linked to their traits. This is briefly mentioned in the paragraph below but again depth is missing.

Response: There could indeed be delays in species migrations. We comment on this as one of the possible explanations for the lack of richness change that we found in our study in the Discussion section:

“Further research is required to determine whether Arctic ecosystems are indeed exhibiting resistance to warming⁶⁵, as other processes could contribute to a lack of detected diversity change. For example, the same species could be both lost and gained across plots over time due to stochastic dynamics or sampling effects (**Table S6**). Future change may not yet be detected due to extinction lags⁶⁶ and slow colonisation rates in communities of long-lived perennial species. Additionally, priority effects could cause heterogeneity in species responses to warming⁶⁷. Variation in topography, microclimate and nutrient availability could

mediate ecological responses and buffer against climate change impacts by providing microhabitats with suitable conditions⁶⁸⁻⁷¹. Thus, the integration of extinction lags, priority effects, and both micro- and macroclimate is an essential next step to better identify the mechanisms behind Arctic plant dynamics.” (L546-557)

We have also mentioned dispersal in the Introduction (see also **2.21** below):

“We hypothesise biotic homogenisation of plant communities (declining spatial β -diversity through time)⁴⁶. This homogenisation could be caused by an infilling of warmer thermal niches^{33,45,46} by the same increasingly dominant species with good dispersal and colonisation capacities⁵², which will outweigh species gains through borealisation.” (L281-285)

2.19. • Line 205 to 217: Can specific hypotheses/likely scenarios be included here? For instance, what may happen to those permafrost soils? Do they behave in a similar way as all other habitat types? As for now it sounds that everything can happen everywhere in the arctic completely neglecting previous studies that have made an effort understanding changes in the arctic.

Response: We have revised this paragraph and believe that the potential scenarios are better spelled out now:

“Warming-driven shifts in species composition are likely to lead to temporal changes in the spatial dissimilarity (i.e., spatial β -diversity changes over time) of plant communities across the Arctic. As observed across other biomes⁴³, Arctic vegetation might become spatially more homogeneous (i.e., lower β -diversity) due to the expansion of dominant and widespread species⁴⁴, such as dwarf shrubs across the High Arctic, as a result of reduced winter mortality and increased recruitment with warming^{45,46}. Similarly, shrub expansion at the forest-tundra ecotone could lead to biotic homogenisation as shrubs become more dominant⁴⁷. In contrast, Arctic landscapes could also become more spatially heterogeneous due to permafrost thaw and hydrology changes with warming, including the development of novel wetland plant communities^{48,49}. Furthermore, the borealisation of Arctic ecosystems close to the treeline could further differentiate Low and High Arctic plant communities⁵⁰. In sum, whether Arctic plant communities will become more or less similar to each other with climate change remains uncertain.” (L226-239)

2.20. • Lines 241 to 244: And what about all the other indices calculated? How do these change? In general, formulating clear hypotheses (and not research questions) that are introduced above would help to reader to better understand the rationale of this manuscript.

Response: We have rephrased the hypotheses section, and specifically included hypotheses for other biodiversity metrics, while specifying mechanism (L265-285).

2.21. • Line 254: This is the first-time dispersal is mentioned. It would be helpful to explain the effects of different dispersal capacities further up already.

Response: We have now added mention to dispersal capabilities earlier in the Introduction, in the context of potential pathways of diversity change:

“2) Richness increases could also result from short-distance dispersal and colonization from species that are already present in local species pools²⁰.” (L182-184)

2.22. • Line 596: Would be good to know the species richness of the final dataset used after cleaning.

Response: We have specified the species richness of the dataset (490 vascular plant species) after cleaning in lines 147, 247, and 597.

2.23. • Line 624: Was climate (temp and precip) from the year of the survey used (I gather from the results that you did; but this needs to be explained better in the methods)? And why did you decide on temp of the warmest quarter for the main results?

Response: We extracted climate data for the whole period available at CHELSA (1978 - 2013) for all plots. We wanted to explore the cumulative effect of warming over time across all plots, so we chose to use the whole available climatic data period for a more representative view of how these plots have changed over time. We have emphasized this in the text for clarity:

“We extracted, at the subsite level, data from long-term climatologies at CHELSA (version 1.2.1)⁸⁴ including mean annual temperature, mean temperature of the warmest quarter (MTWQ) per year, mean temperature of the coldest quarter (MTCQ) per year and mean annual precipitation (MAP, hereafter ‘precipitation’) for the period 1979 - 2013.” (L665-670)

We decided to use temperature of the warmest quarter for the main results because it best represents the growing season conditions in these highly seasonal environments, and thus might better reflect the biodiversity patterns that we explore in this manuscript. See response above (**2.8**) and corresponding text:

“Upon examining correlations between the three temperature variables, we found that most were correlated with each other. Thus, for our temperature variable we used only MTWQ (hereafter ‘temperature’) as it best represents the growing season conditions and has been previously linked to plant biomass, growth and reproductive rates⁸⁵⁻⁸⁷, which are in turn relevant variables driving diversity change.” (L670-674)

We have also included mention to our decision to choose CHELSA as our climate dataset here (see Response to Editor above too):

“We chose CHELSA as the source for our climate data because, as a quasi-mechanical statistical downscaling product, it has a very fine grain size (1x1 km) and has been shown to outperform other interpolation-based climate products, and particularly to perform better predicting precipitation patterns^{84,88,89}.” (L676-680)

Figures

2.24. • Please add summary statistics to all regressions in the figures.

Response: We have now added **Table S2** (Supplementary Information) with the model structure and summary statistics of all key models represented in **Table S1**. In the interest of keeping the figures less busy, we refer in the text and the figure caption to the model number that belongs to each figure.

2.25. • Fig 3: Can you please add the data to the figures? There seems to be a wide variance in the data, which would be interesting to see.

Response: We have added points to all subplots in **Figure 3** as requested.

Figure 3. Local climate, climate change and shrubification influenced species' temporal turnover and trajectories. **a)** Relationships between climate (mean temperature of the warmest quarter, MTWQ) and two temporal turnover metrics: Jaccard (presence-absence turnover) and Bray-Curtis (presence-absence and abundance turnover). Model outputs are in **Table S2.12, 20**, note that the significance of the Bray-Curtis models differed between the univariate and multivariate models (**Table S4**). **b)** Relationships between temperature change over time (slopes from linear models) and the two turnover metrics ($n=1,266$). Model outputs are in **Table S2.16 - 18, 24 - 26**; note that the significance of the Bray-Curtis models differed between the univariate and multivariate models (**Table S4**). The univariate model is represented here for visualization purposes. Nearly half of the plots (526 plots, 41.5%) did not change at all in terms of presence-absence turnover (Jaccard) and only 6 (0.4%) plots did not change at all when considering both presence-absence and abundance turnover (Bray-Curtis). **c)** Turnover metrics were not directly associated with shrub cover change over time (**Table S2.16, 21**) **d)** Relationships between MTWQ and species proportion for each trajectory (species gained and lost, persisting species are not displayed). Model outputs are in **Table S2.36, 44**. **e)** Relationships between MTWQ and species proportion for each trajectory. Model outputs are in **Table S2.40 - 42, 48 - 50**. **f)** Increases in shrub cover over time were associated with decreased species gains (though this effect was non-significant), and increased species losses (**Table S1, S2.40, 48, S4**). In all cases, lines represent predicted model fits and bands show the 95% credible intervals. Dashed lines indicate an overall model whose credible intervals overlapped zero, and solid lines indicate credible intervals that did not overlap zero. All analyses are Bayesian hierarchical models.

2.26. • Fig 4: In all regression plots is huge variation especially in the centre of the plots. What is happening here? The regression line seems mainly to be driven by the outliers to the left and right respectively. Could this have affected the overall results and conclusions?

Response: We have removed the previous Figure 4 and included the relevant subplots in **Figures 2, 3 and S8**, but this reviewer comment referred to the relationship between functional

group cover change and richness (now **Figure 2e, f**), and to shrub cover change and species gains and losses (**Figure 3f**).

There is indeed large variation and the majority of the plots converge towards the centre of the plot (between -5% and 5%). This matches our biological understanding of Arctic shrubs given that the average % cover change in shrub cover per year is within this range (e.g., 1.8% average shrub cover change per year, e.g., García Criado et al. 2020). Considering that these are long-lived, perennial species, their growth is slow and most values of annual cover (positive and negative) will not be too far from zero.

However, we have fitted these models again without the extreme values pointed by the reviewer (see Figure below, now **Figure S9**). Since we followed a rigorous data screening process, we are confident that these extreme values of shrub cover change are real and not the result of any errors. Values were removed when the slopes of functional group change were greater than 3 times the standard deviation. We found that the relationships hold up for shrub cover change (slope = -0.03, 95%CI = -0.04 to -0.02) and for forb cover change (slope = 0.06, 95%CI = 0.05 to 0.07). Graminoid change remains positive, but extreme values outliers this relationship turns non-significant (slope = 0.002, 95%CI = -0.007 to 0.01).

Figure S9. Models of richness change as a function of functional group change (without extreme values of cover change). Values were removed when the slopes of functional group change were greater than 3 times the standard deviation. We found that the relationships hold up for shrub cover change (slope = -0.03, 95%CI = -0.04 to -0.02) and for forb cover change (slope = 0.06, 95%CI = 0.05 to 0.07). Graminoid change remains non-significant (slope = 0.002, 95%CI = -0.007 to 0.01). **a)** Richness decreased as shrub cover increased over time, but increased when **b)** forb cover increased. **c)** There was a non-significant relationship between richness change and graminoid cover change. Scatterplots represent richness change over time as a function of changes in cover of shrubs, forbs and graminoids. Points represent slopes of linear models of change in richness and in functional group change per plot over time. Lines represent predicted model fits and bands show the 95% credible intervals. Dashed lines indicate models whose credible intervals overlapped zero, and solid lines show models whose credible intervals did not overlap zero.

We have included mention to this in the caption of **Figure 2**:

“To ensure that the relationships in e) and f) were not driven by the most extreme changes in functional group cover, we repeated these analyses by removing the extreme values, which yielded consistent results (**Figure S9**).” (L353-355)

And in the results section:

“The effects of shrub and forb change on richness change remained even when extreme values of change were removed from analyses (**Figure S9**).” (L329-331)

2.27 • Fig 4 e and f: Same here – the trend seems very weak (significant?) to justify the strong conclusions drawn from this.

Response: The figures that the Reviewer mentions here are now combined into **Figure 3f**, which shows a significant trend where shrub cover is associated with increased species losses, and a non-significant trend where sites with increased shrub cover experienced fewer species gains (see **Figure 3** above).

This is specified in the figure caption:

“**e**) Relationships between MTWQ and species proportion for each trajectory. Model outputs are in **Table S2.40 - 42, 48 – 50**. **f**) Increases in shrub cover over time were associated with decreased species gains (though this effect was non-significant), and increased species losses (**Table S1, S2.40, 48, S4**).” (L393-396).

We have also run the species losses ~ shrub cover change model without outliers (i.e., those values greater than 3 times the SD) and this relationship holds up (slope = 0.07, 95%CI = 0.04 to 0.11), see figure below. Since the figure without outliers is similar to the one in the text, we haven't included this new figure in the manuscript, but we have added the following text to reflect this:

“There were more species losses where shrubs had increased (**Figure 3f**; this relationship also held up when removing the most extreme values of change)” (L406-407)

Alternative to Figure 3e (losses only). Increases in shrub cover over time were associated with increased species losses (model without outliers). Points represent slopes of linear models of change in shrub cover and the proportion of species per trajectory and plot. Lines represent predicted model fits and bands show the 95% credible intervals.

Referee #3 (Remarks to the Author):

Plant diversity dynamics over space and time in a warming Arctic

3.1. In this paper the authors used an impressive data compilation to examine plant diversity

dynamics in the Arctic. With high latitudes experiencing among the fastest rates of climate change, these results represent an important advance for our understanding of climate impacts on biodiversity. The authors find lots of variation in diversity change, though no changes in local species richness on average. Richness increases were strongest in places that warmed the most, and shrub expansion was associated with the greatest richness declines. Turnover (i.e., species composition changes) and proportions of species gained and lost were related to temperature in the warmest quarter and temperature change. The authors fully embraced the complexity of biodiversity change, and I appreciated their efforts to combine multiple descriptions of the various aspects of diversity (richness, relative abundance, and composition).

Response: Thank you very much for your assessment of our manuscript.

3.2. That said, I did not find it easy to follow all of the results, and perhaps some simplification is in order. In particular, the spatial analyses (the latitudinal and temperature gradients) were not part of the aims set out at the end of the introduction, which all focused on diversity change.

Response: We have simplified the results section to improve flow. We have better integrated the spatial analyses into the context of the temporal analyses that are the focus of the study. While we briefly describe some spatial patterns, this is the set up for our expectations of temporal change. See for example the Results section:

“We found support for the extension of the latitudinal species richness gradient across the Arctic (**Figure 1a**), with higher spatial plot-level richness at lower latitudes (slope = $-0.03 \log(\text{species}) \text{ degree}^{-1}$, corresponding to a decrease of ~one species per every 5° increase at mid-range Arctic latitudes, 97.5% CI = -0.05 to -0.01 ; **Figure 1a, S7, Table S2.1**). Richness was also greater at warmer sites, with approximately one species gained on average for every 2°C increase in warmest quarter temperature (slope = $0.06 \log(\text{species})/^\circ\text{C}$, 97.5% CI = 0.03 to 0.1 , **Table S2.2**) and in plots with greater forb cover and lower graminoid cover (**Figure 1c, Table S1, S2.4 - 5**). Despite greater plant richness at lower latitudes and warmer sites, Arctic plant richness did not change directionally over time (slope = $0.0021 \log(\text{species}) \text{ year}^{-1}$, 95% CI = -0.0002 to 0.0043 , equating to 0.01 species gain per year; **Figure 2b, c, Table S3**).” (L311-321)

We have also specified this in the introduction:

“(1) We quantify spatial patterns in Arctic diversity across latitudinal and climatic gradients, in order to inform our expectations of diversity changes in response to warming.” (L247-249)

3.3. Moreover, the spatial analyses made some additional assumptions to deal with the heterogeneity in plot size that were not well explained. Indeed, how variation in plot size was handled for all results where plots of different sizes were compared (i.e., the spatial analyses and changes in spatial beta-diversity) needs better explanation, and may be sufficiently problematic in some cases to warrant reanalysis or removal.

Heterogeneity in plot sizes is something that we very much considered in our study. Prior to running any analyses, we performed extensive cleaning and constraining of the database in order to ensure comparability and consistency across Arctic plots. From a total of 88,010 records available on the ITEX+ database, our dataset was reduced to 42,234 records after applying our multiple restrictive filters (including plot sizes). We retained only plots $\leq 1\text{m}^2$ in order to minimise the sampling effort effect in our analyses (see Figure below, now **Figure S4**). We acknowledge that a small level of variability remains between plots of different sizes, but heterogeneity in plot size is ubiquitous in biodiversity synthesis studies at such a large scale and involving many different sites and data collectors (e.g., Antão et al. 2020, Blowes et al. 2019).

Additionally, there are different elements that contribute to surveyed areas. While there is indeed some degree of variability between plot sizes (mean = 0.57, range = 0.0484 to 1 m²), the numbers of plots per subsite also vary (mean = 14, range = 1 to 87). This adds another level of replication, so the total sampled area per subsite (where all plots have the same size) could be considered as the plot size * number of plots per subsite. As per the newly produced **Figure S4**, below, the range per subsite of total surveyed area is generally constricted under 20m² in total per subsite, with a few larger values. This might indicate that subsites with greater plot sizes compensate with those with smaller plot sizes if those have greater plot numbers.

We have included **Figure S4** in the Appendix and refer to it in the Methods:

“The total surveyed area per subsite (calculated as plot size * number of plots per subsite) is generally constrained under 20 m² (**Figure S4**).” (L619-621)

Figure S4. Variability in **a)** plot size, **b)** number of plots per subsite, and **c)** total surveyed area, calculated as plot size * plots per subsite.

Additionally, there was no overall relationship between plot size and spatial species richness (**Table S1, S2**), suggesting that sampling effort (i.e., plot size) does not have a strong impact on the spatial analyses. Interestingly, both the GEO and CLIM spatial richness models have a non-significant effect of plot size, while the FG model with functional group cover as covariates does have a positive plot size effect. This could be hinting at a stronger effect of plot-level covariates on the species-area relationship, rather than large-scale factors such as climate (e.g., GEO and CLIM models). Finally, we have also checked whether latitude was confounded with plot size and this relationship is non-significant (slope = 0.007, CI = -0.009 to 0.02). Thus, overall, we believe that there is no strong effect of variability in plot sizes that would affect the results and interpretation of our findings.

We believe that there is a lot of potential to explore species-area relationships in the context of biodiversity metrics, while taking into account scaling issues. While this is currently beyond the scope of our study and research questions, these suggestions have certainly strengthened these aspects of our manuscript.

3.4. The editor directed me to focus my attention on the statistics, which were frequently overly difficult to parse. Table S1 represents an aesthetically pleasing attempt to summarise all the many models fit, but it was difficult to read (e.g., new acronyms not featured in the methods section, coloured symbols, and missing link functions). Greater structure in the methods, e.g., subheadings and separate descriptions linking analyses to specific questions would greatly increase clarity. At the moment, the details of a single analysis emerge across multiple paragraphs in the methods, and some important details (e.g., link functions) are omitted entirely. Critically, the analytical decisions made to deal with the variation in plot size are not

completely described or justified. For example, what are the assumptions that accompany including $\log(\text{plot size})$ as a covariate in the different spatial models? Moreover, there was insufficient clarity in the description of the spatial beta-diversity analyses to understand exactly what was done.

Response: We have restructured the Methods section and added extra sub-headings and further detail into our analytical framework, including link functions. We have also included **Table S2** (Supplementary Information) with all model structure and summary statistics and hope this clarifies our analytical framework and results further. We have also made specific mention to the research questions related to each type of analyses:

“Both two-time point models and temporal models identify the main drivers behind temporal patterns of plant diversity change (cf. RQ 1, 2).” (L734-736)

“We performed ordination analyses to understand whether community homogenisation or differentiation had taken place at the subsite level (cf. RQ 3).” (L826-827)

See below (3.5) for a detailed response on the assumptions of including $\log(\text{plot size})$ in our models, and for the link functions.

3.5. With $\log(\text{plot size})$ included in regression models, I think questions (and inferences) change from species richness to species density (Gotelli & Colwell 2001). Whilst you have not simply divided richness by area, including this control or adjustment covariate is doing something similar.

For example, to generate a predicted richness value from a model with $\log(\text{plot size})$ as a covariate, you need to set plot size to some value, meaning you are predicting richness for a given area, which is a density. Gotelli & Colwell (2001) are not at all dogmatic about one or the other (richness or density) being right, but they do show that they are fundamentally different things.

What are the assumptions made when including the $\log(\text{plot size})$ term in the statistical model? Currently this cannot be fully discerned because the link functions are unknown. Assuming default link functions for the negative binomial (log link) and beta (logit link) error distributions for the richness, and evenness (and gains, losses and persistent proportion) models, respectively, result in fundamentally different assumptions about the scaling of diversity and plot size for the different response variables.

The $\log(\text{richness}) \sim f(\log[\text{plot size}])$ is possibly defensible as a model of species density, though the assumptions entailed need to be fully explicated in the methods. However, I'm not convinced that $\text{logit}(\text{evenness}) \sim f(\log[\text{plot size}])$ makes sense (with the same argument applying for the models of the proportions of species gained, lost and persisting).

Response: We included $\log(\text{plot size})$ as a model covariate to resemble the increasing but saturating shape of the species-area relationship theory. This way, we incorporate scale explicitly into our macroecological study, as recommended in the literature (Drakare et al. 2006). We show below the comparison of the richness ~ plot size relationship with and without the log-transformation. As can be seen from the plots, the $\log(\text{plot size})$ relationship most closely resembles the shape of species accumulation curves.

Figure comparing the richness-plot size relationships between plot size (raw values) and $\log(\text{plot size})$. Each point is a plot displaying current species richness (total number of species at the last survey point) as a function of the surveyed area. Darker points indicate overlap of multiple points. The line represents model fit.

We have also made reference to species richness versus density in the Methods (see below):

“Richness was defined as the total number of species co-occurring in a plot. We acknowledge that some authors refer to this term as ‘species density’ when it is based on an area metric⁹⁰, but hereafter we refer to ‘richness’ as a more common term in the literature.” (L685-688)

We have also specified the link functions for each data family in the Methods:

“**Data families.** We fitted hierarchical models with different response distributions depending on the structure of the response variable (**Table S1**). These included Gaussian with an identity link function (for response metrics with a normal distribution), negative binomial with a log link function (for count data where the variance is greater than the mean), beta with a logit link function (for values between 0 - 1, excluding 0 and 1), zero-inflated beta with a logit link function (for values between 0 and 0.99), and zero-one-inflated beta with a logit link function (for values between 0 - 1, including 0 and 1). For beta families, we included in our models ‘ $z_i - 1$ ’ (where z_i is the probability of being a zero), ‘ $z_{oi} - 1$ ’ (where z_{oi} is the probability of being a zero or a one), and ‘ $coi \sim 1$ ’ (where coi is the conditional probability of being a one, given that an observation is a zero or a one). We specified weakly informative priors for beta and negative binomial families.” (L792-804)

3.6. Unfortunately, I don’t understand how dissimilarity was calculated for the spatial beta-diversity analyses. And, given the complete absence of any mention of plot size, strongly suspect these calculations are confounded by the variation in sample effort (i.e., plot size). Either sample effort needs to be standardised, or compositional (dis)similarity can only be calculated between plots of the same size. Similarly, calculating the distance to a single centroid across plots of varying size seems likely to confound compositional variation with variation in sampling effort.

Response: We did not perform any strictly spatial beta-diversity analyses for this work. The two-time point turnover models (Jaccard and Bray-Curtis) shown in **Table S1** refer to temporal turnover per plot (dissimilarity), while the homogenisation analysis is based on temporal

changes in spatial turnover. We have added further detail to the dissimilarity analysis section (particularly on the plot-to-subsite aggregation) and hope this clarifies our process:

“In order to assess temporal changes in spatial turnover, we calculated spatial dissimilarity in species composition at the first time point for all subsites, and at the last time point separately. To aggregate plot-level data into subsite-level data, we calculated the mean cover per species across all plots in a subsite, both for the start timepoint and for the end timepoint.” (L826-829)

See above (**3.3**) for a detailed response on how we dealt with heterogeneity in plot sizes in general. In terms of its potential influence on the homogenisation/differentiation analyses, this should not affect the analyses showing changes in dissimilarity at the subsite level relative to each starting point, since all plots within each subsite have the same plot size (except one, representing 0.6% of subsites).

Some dissimilarity metrics might be more sensitive to sample completeness (e.g., species presence/absence), which might be in turn affected by plot size. Therefore, we calculated 6 different metrics of dissimilarity with varying degrees of emphasis on presence/absence versus abundance: Jaccard, Sorensen, Bray-Curtis, Modified Gower, Manhattan and Euclidian (**Figure S15**). We would expect to see differences in the dissimilarity metrics if plot size variation was driving sample completeness. Since we have found the same outcomes (i.e., no homogenization or differentiation) across all 6 metrics, we understand that plot size does not have a strong influence on our results.

However, we have tested this further to confirm the lack of influence on our results by plotting the two PCoAs in **Figure 4** and colouring them by total surveyed area per subsite (see **3.3**). Black points represent subsites where the total surveyed area (i.e., plot size * number of plots per subsite) was $> 20\text{m}^2$, as those were the clear extreme values in the plot in **3.3**. If sampling area was influencing dissimilarity, the subsites with greater surveyed areas would have more change or be grouped in the same graph area. As seen below in the graphs, there is no clear pattern for those subsites with greater surveyed areas, i.e., they are not driving the edges of the ellipses, they are not in a specific range of latitude if cross-checked with the original **Figure 4**, and they don't have the smallest or greatest rates of change relative to their starting point.

Overall, we think that there are a lot of research questions that can be tested with regards to homogenisation/differentiation at the Arctic scale. We hope that this first approach yields some interesting insights at this stage, and, as shown, plot size should not be influencing our results or their interpretation.

There was no patterning to subsites that had greater total surveyed areas. We calculated temporal change in spatial turnover (β -diversity) between the start (i.e., baseline) and end (i.e., final) time period for all subsites. Principal Coordinate Analyses (PCoAs) are shown with the **a)** Jaccard and **b)** Bray-Curtis β -diversity metrics. Triangles represent the start time point and circles represent the end time points for all subsites, joined by a line per subsite indicating the amount of compositional change. Points are coloured according to the total surveyed area (average are values $\leq 20\text{m}^2$ and large are values $> 20\text{m}^2$).

3.7. So, that leaves the temporal analyses. And I was reassured to read that only plots with consistent sampling methods and sizes over time were retained for analyses (L583-584). Technically, these are also models of species density, though I acknowledge that referring to this as species richness is very common in the ecological literature (though you might consider referring to something other than richness for accuracy).

I did get a bit confused between the “two-time point” and “temporal” models (terms that were introduced, but then not further described individually in detail in a single place). Table S3 says the model fit to calculate rates of species richness change assumed a negative binomial error distribution, though there is no mention of the link function. Assuming the default (log link), then the units of change are $\log(\text{richness})/\text{year}$; this needs to be checked and clarified throughout (e.g., Figure 2, Figure 4a-c, L440).

Note also that changes in evenness calculated assuming a beta error distribution are not linear in time (due to the default logit link). Regardless of the scale of the modelled responses (natural, logarithmic, logit-transformed), I think these temporal analyses are the strongest component of the paper, and the ones that I would focus the presentation on (due to their documenting changes in plots of consistent size).

Response: Thank you for your positive assessment of the temporal analyses of our manuscript. Indeed, since only plots with consistent sizes over time were retained for analyses, variability in plot size should not affect any of the temporal analyses.

See **3.5** for a response on species density vs richness, which is now reflected in the Methods section:

“Richness was defined as the total number of species co-occurring in a plot. We acknowledge that some authors refer to this term as ‘species density’ when it is based on an area metric⁹⁰, but hereafter we refer to ‘richness’ as a more common term in the literature.” (L685-688)

We have restructured the Methods section and clarified the distinction between two time-point and temporal models:

“Model types. We fitted three main types of models: spatial, two time-point and temporal (**Table S1**). 1) Spatial models refer to current biodiversity metrics across space, with one unique value of the response variable (richness, evenness) measured at the last monitoring timepoint. These models identify the main drivers behind spatial patterns of plant diversity. 2) Two-time point models use a response variable that has been derived from two points in time, with a single value providing the measure of change (temporal turnover via Jaccard and Bray-Curtis, species losses, gains and persisting species). 3) Temporal models reflect metrics whose response variable had multiple values over time, and at least start and end values (richness change, evenness change, models derived from the spatial homogenisation over time analyses). For these temporal models (richness change and evenness change), we followed a two-step modelling approach to examine biodiversity metrics over time. First, we calculated change over time by fitting linear models of richness and evenness per plot with sampling year as the fixed effect (one linear model per plot); these are referred to as ‘change over time models’ (CHG). Then, we extracted the slopes of change over time per plot and used them as a response variable in a second set of models to test the relationships between putative drivers of temporal diversity change, which were measured at the plot- or subsite-level (**Table S1**). Both two-time point models and temporal models identify the main drivers behind temporal patterns of plant diversity change (cf. RQ 1, 2).” (L716-736)

The link functions have also been described in the Methods section:

“Data families. We fitted hierarchical models with different response distributions depending on the structure of the response variable (**Table S1**). These included Gaussian with an identity link function (for response metrics with a normal distribution), negative binomial with a log link function (for count data where the variance is greater than the mean), beta with a logit link function (for values between 0 - 1, excluding 0 and 1), zero-inflated beta with a logit link function (for values between 0 and 0.99), and zero-one-inflated beta with a logit link function (for values between 0 - 1, including 0 and 1). For beta families, we included in our models ‘ $z_i \sim 1$ ’ (where z_i is the probability of being a zero), ‘ $z_{oi} \sim 1$ ’ (where z_{oi} is the probability of being a zero or a one), and ‘ $coi \sim 1$ ’ (where coi is the conditional probability of being a one, given that an observation is a zero or a one). We specified weakly informative priors for beta and negative binomial families.” (L792-804)”

Thank you so much for catching the issue with the model output units. You are correct that we were using the log link for negative binomial distribution, thus we have corrected the units throughout. We have done this as follows:

“We found support for the extension of the latitudinal species richness gradient across the Arctic (**Figure 1a**), with higher spatial plot-level richness at lower latitudes (slope = $-0.03 \log(\text{species}) \text{ degree}^{-1}$, corresponding to a decrease of ~one species per every 5° increase at mid-range Arctic latitudes, 97.5% CI = -0.05 to -0.01 ; **Figure 1a, S7, Table S2.1**). Richness was also greater at warmer sites, with approximately one species gained on average for every 2°C increase in warmest quarter temperature (slope = $0.06 \log(\text{species})/^\circ\text{C}$, 97.5% CI = 0.03 to 0.1 , **Table S2.2**) and in plots with greater forb cover and lower graminoid cover (**Figure 1c, Table S1, S2.4 - 5**). Despite greater plant richness at lower latitudes and warmer sites, Arctic plant richness did not change directionally over time (slope = $0.0021 \log(\text{species}) \text{ year}^{-1}$, 95% CI = -0.0002 to 0.0043 , equating to 0.01 species gain per year; **Figure 2b, c, Table S3**).” (L311-321)

Finally, we have corrected axis labels in **Figure 2c** to reflect this.

3.8. My final query regarding the temporal analyses, and in particular the link to temperature

change, is whether the authors looked at an interaction between temperature change and a “baseline” temperature for given subsite? Earlier work, e.g., Antao et al 2020 NEE, suggests at least the potential for the effect of temperature change to depend on average temperatures at a site, and the authors also speculate about greater species losses in plots that warmed the most being due to cold plots experiencing the greatest warming (L462). Is this something that they can test directly with these data?

Response: We did not explicitly model the potential interaction of baseline temperature on temperature change, since we focused on covariates indicating change (e.g., functional group change, temperature change, precipitation change) for the temporal analyses, and on baseline/average covariates for spatial analyses. However, you raise an interesting point and we have fitted an example model as Richness change ~ temperature change * temperature climatology + log(plot size) + duration + (1|subsite), as a Gaussian family, and the interaction is non-significant (slope = 0.06, 95%CI = -1.05 to 1.17). We have included this in the Results section:

“Species richness change was not related to latitude (**Figure 2a, Table S2.51**), nor to longterm warming trends (**Figure 2d, Table S4**). There was no interactive effect between temperature and temperature change on richness change (slope = 0.07, 95%CI = -0.65 to 0.78).” (L321-325)

Other, line referenced comments:

3.9. L180: elevational gradients mirror latitudinal gradients in what sense? Temperature? Diversity? Small edit for clarity.

Response: We have rephrased this sentence to:

“This expectation is further supported by observed climate-induced increases in vascular plant species richness across European mountain tops^{29,30}, whose elevational gradients mirror Arctic latitudinal climatic and richness gradients³¹.” (L201-204)

3.10. L242: I’m not sure what “reflecting” means in this sentence.

Response: We have added more context in the first part of the sentence and hope this reads a bit clearer:

“We expect an overall increase in plot-level richness (α -diversity) over recent decades across the Arctic. We expect greater richness increases in warmer sites and at lower latitudes, which are closer to boreal forest species pools, paralleling the latitudinal biodiversity gradient⁵¹.” (L267-270)

3.11. L267: random with respect to what?

Response: We defined a function to extract a random selection of plots across the Arctic, together with their respective climatic data. We have rephrased this to:

“Background grey points represent a selection of randomly extracted geographic coordinates from the Circumpolar Arctic Vegetation Map³⁵, including 1,189 locations across the Arctic for which climatic data were extracted.” (L294-296)

3.12. L279: distracting (bolded) topic sentence. This bolded text might work as a sub-heading; but the only sentence in this paragraph related to changes in species richness is the last one. The majority of this paragraph reports results not related to changes, and which were not part of the questions introduced at the end of the introduction.

Response: We have restructured the Results section and added a more general sub-heading for each major sub-section.

3.13. L283-284: I'm likely missing something simple, but how does a slope of 0.06 species/°C equate to one species gained for an increase of 2°C?

Response: The back-transformed relationship between species richness ~ temperature is not linear (since the negative binomial distribution has a log link), and thus this is an approximate estimate. This has been calculated using the model predictions as follows:

$(Y_2 - Y_1) / (X_2 - X_1) = (11.91 - 5.2) / (14.5 - 1.2) = 0.5$ species/degree celsius. Therefore, $1 * 1 / 0.5 = 2$ degrees for one species change.

This also applies to the richness ~ latitude relationship. This estimation has been calculated using the model predictions as follows:

$(Y_2 - Y_1) / (X_2 - X_1) = (4.92 - 8.83) / (80-60) = -0.19$ species/degree. Therefore, $1 * 1 / -0.19 = -5.26$ degrees for one species change.

The fact that this is an estimation from a non-linear, back-transformed relationship is emphasised in the text as:

“corresponding to a decrease of ~one species per every 5° increase at mid-range Arctic latitudes” (L313-314)

Richness was also greater at warmer sites, with approximately one species gained on average for every 2°C increase in warmest quarter temperature (L315-316)

3.14. L286: average (?) plot-level richness change was not different from zero?

Response: Thanks for this suggestion. We have trimmed down this paragraph and rephrased to:

“Despite greater plant richness at lower latitudes and warmer sites, Arctic plant richness did not change directionally over time (slope = 0.0021 log(species) year⁻¹, 95% CI = -0.0002 to 0.0043, equating to 0.01 species gain per year; **Figure 2b, c, Table S3**).” (L318-321)

3.15. Figure 2: do the estimates of richness change (i.e., slope estimates) on Fig. 2b and Fig. 2c come from the same model?

Response: the reviewer was referring to this comment to what are now Figure 2b and 2c. **Figure 2b** shows the estimate of richness change at the plot level, and **Figure 2d** shows the estimate at the subsite level. This is mentioned in the figure caption, but we have also edited the figure axis titles themselves for clarity.

3.16. L 321: difficult to see these mean values on Figure 3a, b. Are they the (unlabelled) density plots on the righthand side? Can the density plots be labelled (i.e., axes and a panel tag) so as a reader can see this as the distribution of turnover values easily. Alternately, remove this pointer to the figure.

Response: We have removed the density plots and included the data points for clarity in **Figure 3** (see response to **2.25** above).

3.17. Similarly, are the relationships between turnover and species richness (L323, L324) visible on the plot? I got confused looking, and couldn't find them.

Response: You are correct, the reference to **Figure 3b** related to the second part of the sentence (weaker warming trends), not the species richness statement. We have rephrased this to:

“Conversely, greater abundance-related temporal turnover (Bray-Curtis) occurred in warmer sites, regions with weaker warming trends (**Figure 3a, b, Table S2.24 - 26, S4**), species-rich plots (**Table S2.19**), and plots monitored over longer periods of time (**Figure S11**).” (L368371)

3.18. L403-405: does this refer to dissimilarity distance? It is worth making this explicit here, I was confused after the PCoA results (and distances to centroids).

Response: This refers to changes in community composition and abundance over time, relative to the start time point of each subsite, as calculated by Cartesian coordinates (see L819-834 for detailed methods). We have specified this here in the Results as follows:

“Mean shifts in distance between timepoints per subsite (as Cartesian coordinates, reflecting change in community composition relative to starting point) was 0.035 ± 0.03 (Jaccard) and 0.04 ± 0.03 (Bray-Curtis, **Figure 4e**).” (L433-436)

3.19. L453: I'd call this incomplete sampling (i.e., present but not captured due to rarity and the finite nature of biodiversity samples).

Response: Thank you for this suggestion. In this section, we are referring to the 'dark diversity' term coined by Meelis Pärtel and colleagues, among others. We are referring to species that are present in local species pools, but have not yet been found at the specific sampling location, inside long-term monitoring plots. So it's not necessarily incomplete sampling (where species are missed due to not enough sampling at a given location), rather species that might end up being reported if and when they colonise long-term plots. We have rephrased this to:

“Instead, this lack of latitudinal change might indicate that, where diversity is changing, one of the main sources is colonisations by species present in local species pools that have not yet been recorded in long-term monitoring plots (referred to as 'landscape' or 'dark' diversity)^{20,54}.” (L479-482)

3.20. L474: check language. “...fewer species losses and gains...” implies a count. However, the figures suggest that you modelled (and only report) proportional (or percentages).

Response: Correct, losses and gains are modelled as percentages. We have rephrased this to:

“We found some evidence for community resistance to rapid Arctic warming, with proportionally fewer species losses in plots that were more diverse and even (**Figure S10**).” (L463-465)

3.21. L476-478: the logic underpinning how the species pool size can influence greater abundance turnover is not clear to me. If larger species pools have more even SADs, then perhaps stronger changes in relative abundance are possible?

Response: Our logic here was that in colder, low richness plots, changes in species replacement will appear to be greater given that proportionally, species gains/losses will result

into a greater turnover value. Conversely, warmer plots with generally greater species richness seem to experience greater abundance turnover as there is greater potential for species gains and losses. We believe that this makes mechanistic sense as with more species present in a plot, these communities may experience greater change due to biotic interactions (facilitation, competition) between the different plant species. We have edited this sentence and integrated it with the turnover paragraph in the Supplementary Results:

“Turnover. A key difference between turnover metrics was that more species-rich plots experienced greater abundance-related turnover (Bray-Curtis), while species-poor plots had more presence-absence (Jaccard) change (**Table S1**). We contrasted these results with a null model that yielded no significant relationship between Jaccard and richness, and a negative relationship between Bray-Curtis and richness. Thus, this contrasting pattern is likely due to less diverse plots in colder sites dominated by a few species with higher abundance that do not experience as much proportional abundance change, but whose presence-absence change will be proportionally greater; and more diverse plots at lower and warmer latitudes that can experience more proportional abundance change potentially due to increased biotic interactions^{2,3}.”

3.22. L482: I cannot see shrubification on Figure 5.

Response: Thanks for catching that. We have completely rephrased the Discussion now and include a whole paragraph on the effects of shrubification, with reference to the right figures:

“Shrubification as a driver of diversity change. We found that shrubification was associated with richness and compositional change. Shrub expansion has been widely reported^{2,4,56}, though we found only a marginal increase (i.e., the credible intervals overlapped zero) in Arctic shrub cover at the plot scale within the ITEX+ dataset (**Table S9**). Shrub cover change has been widely linked to warming in previous site-level studies²⁻⁴. However, we did not find clear evidence for greater shrub change with greater rates of warming (**Figure S14c**), in agreement with previous pan-Arctic studies⁴. Instead, shrub cover was sensitive to temperature, with non-dwarf shrub cover increasing and dwarf shrub cover decreasing with warmer temperatures (**Figure S14b**). Where shrub cover increased over time, plots experienced decreases in species richness, community evenness, and greater species losses (**Figure 2e, 3f, Table S1**). Lower species richness has been observed with greater shrub cover spatially, with shading and litter production leading to decreases in sun-loving plants under shrub canopies^{23,24}. Our Arctic-wide results corroborate site-level reports that increasing shrub cover over time may lead to less diverse plant communities and the displacement of less competitive species^{41,57,58}. Thus, Arctic diversity might be more at risk at sites with increasing shrub cover, particularly from tall shrubs (**Figure 2e**). Conversely, both increasing graminoid and forb cover were associated with increased richness over time, and increasing graminoid cover was related to lower species losses (**Figure 2f, Table S1**). Graminoids were more likely to persist than forbs (**Table S1**), perhaps because graminoids are good competitors that can displace shallow-rooted forbs where they both co-occur due to their deeper root networks, faster nutrient uptake and greater height⁵⁹⁻⁶¹. Overall, our findings suggest that increased competition from shrubs^{61,62} is a main driver of Arctic diversity change.” (L496-520)

3.23. L491: does the tense change for the first half of this sentence?

Response: Corrected. The sentence now reads:

“Conversely, both increasing graminoid and forb cover were associated with increased richness over time, and increasing graminoid cover was related to lower species losses (**Figure 2f, Table S1**).” (L513-515)

3.24. L523: here (and other places in this paragraph) you refer to a lack of diversity change

(i.e., a statement about diversity in general). But haven't you found considerable changes, e.g., compositional. Does the "lack" refer to average richness change? If yes, be more precise.

Response: Yes, we were indeed referring to richness change. We have rephrased this sentence to:

"Our findings demonstrate some resistance of Arctic plant richness change (**Figure 2b, c**) despite continued warming." (L523-524)

3.25. L528: what is a slow lag?

Response: Corrected. The sentence now reads:

"Future change may not yet be detected due to extinction lags⁶⁶ and slow colonisation rates in communities of long-lived perennial species." (L550-551)

3.26. L614-617: sometimes cover or proportions were analysed with response distributions that respect the bounded nature of the data (i.e., 0-1 interval), other times not? This is not necessarily a problem, but you need to state the assumptions and give reasoning for why different choices were made.

Response: We were consistent in the way we treated our data, and chose the most appropriate data family depending on the spread and structure of the response variable. We describe our choice of response distributions in the Methods - Data families section, which now includes the link functions for each data family too:

"**Data families.** We fitted hierarchical models with different response distributions depending on the structure of the response variable (**Table S1**). These included Gaussian with an identity link function (for response metrics with a normal distribution), negative binomial with a log link function (for count data where the variance is greater than the mean), beta with a logit link function (for values between 0 - 1, excluding 0 and 1), zero-inflated beta with a logit link function (for values between 0 and 0.99), and zero-one-inflated beta with a logit link function (for values between 0 - 1, including 0 and 1). For beta families, we included in our models 'zi - 1' (where zi is the probability of being a zero), 'zoi - 1' (where zoi is the probability of being a zero or a one), and 'coi ~ 1' (where coi is the conditional probability of being a one, given that an observation is a zero or a one). We specified weakly informative priors for beta and negative binomial families." (L792-804)"

3.27. L638: if I understand correctly, you do not calculate a rate (i.e., changes per unit time). Turnover is species compositional dissimilarity between the baseline sample and the last sample (where duration between samples are highly heterogeneous [Figure S1]). It'd be good to see a plot of turnover - f(duration). Indeed, some examination of scale-dependencies in model results (see suggestions e.g., Spake et al. 2020 Ecology Letters) would be a great addition.

Response: Correct, turnover is a metric of change, and not a rate (see also response to comment above). We have scanned the text to ensure that we never refer to 'turnover rates' but 'values' instead (also see response about this below).

The results of the turnover - duration metric are in **Table S1** and the newly created **Table S2**, where the relationship between Bray-Curtis and study duration is positive, and the relationship between Jaccard and study duration is non-significant. This result shown in the newly created **Figure S11** and referred to in the Results:

“Conversely, greater abundance-related temporal turnover (Bray-Curtis) occurred in warmer sites, regions with weaker warming trends (Figure 3a, b, Table S2.24 - 26, S4), species-rich plots (Table S2.19), and plots monitored over longer periods of time (Figure S11).” (L368371)

And in the Supplementary information:

“We found that longer duration studies had increased abundance turnover and richness over time, but not increased species replacement (Figure S11).”

Figure S11. Relationship between turnover metrics and study duration. **a)** There is a non-significant relationship between Jaccard turnover and study duration and **b)** a positive relationship between Bray-Curtis turnover and study duration. Each point represents a plot, and the solid line represents model fit, with the semi-transparent bands representing the 95% credible intervals. A solid line represents a model estimate whose credible intervals do not overlap zero, and a dashed line represents a model estimate whose credible intervals overlap zero.

We have certainly thought about and incorporated scale context in our conceptual and methodological design. For instance, across all three model types (spatial, two time-point and temporal) and for each response variable, we fitted several models (geographical (GEO), climatic (CLIM), functional group composition (FG), change over time model (CHG), plot change over time (PCHG), subsite (SUBS)) depending on the scale at which the covariates affected the response variable, in order to avoid collinearity and obscuring patterns between fixed effects (Table S1). While certain patterns have a strong effect across a large area (i.e., climate), others are much more local and contribute to diversity patterns at a plot-level scale (i.e., functional group cover change over time). We have now explicitly mentioned scale in the Methods section:

“Across all three model types and for each response variable, we fitted several models (geographical (GEO), climatic (CLIM), functional group composition (FG), change over time model (CHG), plot change over time (PCHG), subsite (SUBS)) depending on the scale at which the covariates affected the response variable, in order to avoid collinearity and obscuring patterns between fixed effects (Table S1, S2). We used a hierarchical modelling approach by including a subsite random effect (as random intercepts) to account for non-independence of plots within subsites.” (L738-744)

In terms of temporal scaling, it would be expected that longer duration studies show greater change. Thus, we included duration (in years) as one of our ‘sampling design covariates’, together with the aforementioned plot size (**Table S1, S2**). Duration was only a significant factor in the above mentioned Bray-Curtis model (positive) and in the richness change model (positive).

We have rephrased the relevant text about this in the Appendix and included a new paragraph on the importance of scale:

“We found that longer duration studies had increased abundance turnover and richness over time, but not increased species replacement (**Figure S11**). There was an indication that shorter studies identified more species losses, possibly suggesting that local extinctions might be due to stochastic processes or sampling effects during monitoring⁸. Larger plots were more even and had experienced less richness change^{9,10}. Together, these findings indicate the importance of scale, local context, study design and sampling effort when drawing global inferences from local-scale monitoring”.

While implementing the suggested revisions on temporal scaling, we decided to incorporate a stricter cut-off of plots retained for temporal analyses and are now retaining only plots that have a minimum duration of 5 years of monitoring. Previous studies have used this cut-off (e.g., Bjorkman et al. 2018) due to the potential of short timeseries overrepresenting real change (Valdez et al. 2023). This is particularly evident in the tundra, as it takes a few years for changes to plant communities to change meaningfully beyond measurement error in response to global change drivers. After exploring trends of change at different durations, we uncovered that indeed those plots with <5 year durations had extremely high variability, likely because detection isn’t 100% accurate, thus leading to measurement errors that are not evident in timeseries of at least 5 years.

This new methodological step has resulted in a change to one of the main results of the manuscript, where temperature increases are no longer associated with richness increases over time. This particular analysis resulted to be particularly sensitive to duration, as we performed a sensitivity analysis with a minimum of 2 and 5 years, which showed different levels of significance. We have updated our results in the figures and text of the manuscript accordingly.

This decision is specified in the Methods:

“Our dataset contained 2,174 plots, and they were all retained for spatial analyses. For temporal analyses, we retained the 1,266 plots (58.2%) that had been surveyed at least twice over a minimum of five years, since shorter timeseries tend to overrepresent real change in Arctic communities^{17,62}. The remaining 908 plots (41.7%) were only used in the spatial analyses (**Figure 1d, S3**). Of all the plots that were surveyed more than once, 35.3% were surveyed twice, 21.5% were surveyed thrice, 19.7% were surveyed four times, 23.3% were surveyed five or more times, and 0.5% were surveyed ten or more times.” (L607-614)

The importance of scale cannot be discounted when interpreting our results. Here, we have retained only plots <1m² to ensure consistency and comparability across Arctic sites (see **3.3**). Thus, our findings and the patterns of change explored here show change in small-scale communities, and have to be interpreted in this context as these results would not necessarily be expected to scale up (Spake et al. 2020). This means that processes that can only be captured at larger scales might require datasets with larger grain size than these small-sized plots, which in turn allow us to explore questions related to climatic effects on plant communities, and biotic interactions taking place at smaller scales.

We have ensured that the scale at which we refer throughout the manuscript is clear, and signposted when referring to plot-level versus subsite-level change. One specific change we have made is to modify when we refer to “cold/warm plots” to “cold/warm sites”, given that climatic data is extracted at the subsite level, with all plots within a subsite having the same climatic data (which was accounted for with a subsite random effect). We refer here to ‘sites’ as a general term indicating location, rather to a specific term in the ITEX+ plot/subsite/study area hierarchy. We specify this terminology in the Methods:

“We use the terms ‘plant communities’ or ‘sites’ when referring more generally to groups of Arctic species at any scale or resolution.” (L621-623)

3.28. L648-653: this paragraph suggests that your losses and gains are counts of species. But the figures report percentages and Table S1 shows beta regression models. Needs clarification, i.e., what was the number used as the denominator to convert counts to proportions? Was it the same for gains and losses?

Response: We have added the following sentence, which we hope clarifies our methodology:

“These species trajectories were originally calculated as counts, and then transformed to proportions for modelling. Species proportions were calculated by dividing the number of species per trajectory in a plot by the total number of species in each plot (including losses, gains and persisting species).” (L704-708)

3.29. L662-664: this suggests that the two time point models are differences, i.e., they are not rates per unit time. Unless I am missing something, you need to check throughout that you have not described these as rates. Or are only the turnover results reported with these two-time point models? In general, the methods describe a lot of different models in a piecemeal fashion. Linking each question to the descriptions of their respective analyses in a single spot would increase clarity.

Response: This is correct, the two time-point models are differences (as derived from two different points in time) and not rates. We have checked throughout the manuscript to ensure that there is no mention of ‘turnover rates’. Instead, turnover is described as ‘turnover metrics’, ‘turnover values’ or simply ‘turnover’.

Further, we have re-structured the Methods section for clarity and included additional sub-headings to improve readability. We have also made the link between the main model types and the three main research questions (see **3.4**).

3.30. L669: what is the difference between a change over time model, and a plot change over time model? TableS1, whilst aesthetically appealing, is very difficult to read and doesn’t help me to answer this question.

Response: The change over time model is the first step of the temporal models, where we calculate change over time by fitting linear models per plot. We have signposted to this and hope it’s clearer now:

“For these temporal models (richness change and evenness change), we followed a two-step modelling approach to examine biodiversity metrics over time. First, we calculated change over time by fitting linear models of richness and evenness per plot with sampling year as the fixed effect (one linear model per plot); these are referred to as ‘change over time models’ (CHG). Then, we extracted the slopes of change over time per plot and used them as a response variable in a second set of models to test the relationships between putative drivers of temporal diversity change, which were measured at the plot- or subsite-level (**Table S1**).” (L726-734)

We have also signposted the model types in the Methods section so **Table S1** is easier to read (and it's also backed up by the newly created **Table S2** with all model outputs):

“Across all three model types and for each response variable, we fitted several models (geographical (GEO), climatic (CLIM), functional group composition (FG), change over time model (CHG), plot change over time (PCHG), subsite (SUBS)) depending on the scale at which the covariates affected the response variable, in order to avoid collinearity and obscuring patterns between fixed effects (**Table S1, S2**). We used a hierarchical modelling approach by including a subsite random effect (as random intercepts) to account for non-independence of plots within subsites.” (L738-744)

3.31. L675: are these two-time point models, or the temporal models (i.e., > 2 time points) as described above?

Response: This content applies to all temporal metrics (i.e., temporal models and two time-point models). We have clarified this now:

“For all temporal metrics (i.e., temporal models and two time-point models), we retained those plots with a minimum of two sampling points and at least five years of monitoring duration.” (L747-749)

3.32. L699-700: the question underlying the null model is not clear to me. And, though I possibly missed it, I couldn't find a description of the null model either, just a terse statement stating no relationship between turnover and richness in the supplementary results.

Response: We have included more detail on the null models in this section:

“To understand whether our temporal turnover versus richness models reflected a priori relationships or whether there was a meaningful biological relationship, we compared them with null models. To fit null models, we randomly removed 20% species per plot (to simulate species losses), and randomly included 20% species (to simulate species gains). We used this simulated dataset to calculate turnover values (Jaccard and Bray-Curtis). We fitted intercept-only null models with each metric, and modelled Jaccard and Bray-Curtis turnover as a function of species richness.” (L755-762)

3.33. L703-704: This is key. I was wondering what you did with the variation in the plot size. I think models that include log(plot size) effectively mean results describe species density, not species richness (Gotelli & Colwell 2001). Different link functions for the different response distributions are going to result in different scaling relationships, some of which might be hard to justify.

See above **3.5** for a detailed explanation of the implications of including log(plot size) and the description of the link functions, and **3.7** for species richness versus density.

3.34. L712-719: link functions? Do you include covariates for the zero or one components of the zero-inflated or zero-one-inflated beta regression models?

Response: We have specified all link functions in the following paragraph, together with clarification on the zoi and coi components in our beta models.

“**Data families.** We fitted hierarchical models with different response distributions depending on the structure of the response variable (**Table S1**). These included Gaussian with an identity link function (for response metrics with a normal distribution), negative binomial with a log link function (for count data where the variance is greater than the mean), beta with a logit link

function (for values between 0 - 1, excluding 0 and 1), zero-inflated beta with a logit link function (for values between 0 and 0.99), and zero-one-inflated beta with a logit link function (for values between 0 - 1, including 0 and 1). For beta families, we included in our models 'zi - 1' (where zi is the probability of being a zero), 'zoi - 1' (where zoi is the probability of being a zero or a one), and 'coi ~ 1' (where coi is the conditional probability of being a one, given that an observation is a zero or a one). We specified weakly informative priors for beta and negative binomial families." (L792-804)"

3.35. L726: I'd like to see all models fully described in the methods. I found Table S1 very difficult to read. e.g., the methods introduced the idea of spatial, two-time point models, and temporal models, but the table does not use this terminology, and introduces new acronyms for a different classification of models fit (GEO, CLIM, FG, CHG). There is no mention of link functions in the table or methods. Models with multiple likelihoods (e.g., zero-inflated or zero-one-inflated beta regression models) are not fully described (i.e., were covariates described in Table S1 used to predict the mean only?).

Response: We have now added **Table S2** (Supplementary Information) with the model structure and summary statistics of all models and hope this clarifies model structure and type. All link functions are now described in the Methods (see **3.34** and above) in lines 792-804.

3.36. L731: how was species composition (and relative abundance) determined for subsites? I understand the lowest resolution of the data to be plots. How was the aggregation to subsites done? And, again for emphasis, I think the spatial variation in plot size is going to confound these calculations of composition dissimilarity. Without some standardisation, I think you can only calculate (dis)similarity between plots of the same size. Similarly, calculating the distance to a single centroid using plots of varying size is confounded by the variation in plot size to me.

Response: See **3.3** and **3.6** for a detailed response on plot size heterogeneity, and specifically on how this wouldn't influence the results of the homogenisation/differentiation analyses. Further, we have clarified the aggregation from plot to subsite with the addition of the following text:

"In order to assess temporal changes in spatial turnover, we calculated spatial dissimilarity in species composition at the first time point for all subsites, and at the last time point separately. To aggregate plot-level data into subsite-level data, we calculated the mean cover per species across all plots in a subsite, both for the start timepoint and for the end timepoint." (L827-832)

Literature cited

L. H. Antão, A. E. Bates, S. A. Blowes, C. Waldoock, S. R. Supp, A. E. Magurran, M. Dornelas, A. M. Schipper, Temperature-related biodiversity change across temperate marine and terrestrial systems. *Nature ecology & evolution*. 4, 927–933 (2020).

N. J. Gotelli, R. K. Colwell, Quantifying biodiversity: procedures and pitfalls in the measurement and comparison of species richness. *Ecol Lett*. 4, 379–391 (2001).

R. Spake, A. S. Mori, M. Beckmann, P. A. Martin, A. P. Christie, M. C. Duguid, C. P. Doncaster, Implications of scale dependence for cross-study syntheses of biodiversity differences. *Ecology Letters*. 24, 374–390 (2021).

References cited

Blowes, S.A. et al. The geography of biodiversity change in marine and terrestrial assemblages. *Science*, 366(6463), 339-345 (2019).

Bjorkman, A.D. et al. Plant functional trait change across a warming tundra biome. *Nature* 562, 57 (2018).

Garcia Criado, M. et al. Woody plant encroachment intensifies under climate change across tundra and savanna biomes. *Global Ecology and Biogeography*, 29(5), 925-943. (2020).

Karger, D.N. et al. Climatologies at high resolution for the earth's land surface areas. *Scientific Data* 4, (2017).

Valdez, J.W. et al. The undetectability of global biodiversity trends using local species richness. *Ecography*, e06604 (2023).

Editor comments

Your manuscript, "Plant diversity dynamics over space and time in a warming Arctic", has now been seen by 3 of the previous referees, and a 4th referee who specifically commented on the climate modelling. Their comments are attached below. As you can see, while they continue to find your work of potential interest, they still raise important concerns that in our view need to be addressed before we can consider publication in Nature.

In particular, although Referee #1 is satisfied with the changes, Referee #2 is still not convinced that the study is sufficiently novel for Nature. Referee #3 continues to find the methods difficult to interpret, and still is not convinced that some aspects of the statistical approaches are sound. Referee #4 feels that additional analysis is needed to integrate the effect of snow cover, and the potential effects of proximity to ice melt sites.

Thank you very much for this assessment and for giving us the opportunity to revise and resubmit our consolidated work to Nature. We have made substantial revisions to address the reviewers' comments, and we believe that these have greatly improved the manuscript. We have numbered all referees' comments and provided our detailed answers in blue below. We provide both a version of the manuscript and supplementary information with track changes, and another version without track changes. Line numbers here refer to the version without track changes.

In particular, we are confident that we have now addressed all the remaining concerns from:

(i) Referee #2 about the novelty of our work. Specifically, by streamlining the key and novel messages of this study, clarifying the main drivers of directional change in Arctic plant communities, and communicating the advance of this study beyond previous research. We have also performed an additional analysis of species losses as a function of rarity, which showed that rare species are more likely to be lost from plots.

(ii) Referee #3 about providing clarification of the methods, by undergoing a major overhaul of the Methods section and clarifying statistical details. We have also removed previous mentions of community resistance, to instead emphasise the variety of direction in which we have observed plant diversity change. We have also performed a sensitivity analysis with only 1m² plots, which yielded similar results to our main analyses including all plots. We have also removed log(plot size) as a predictor variable in evenness models.

(iii) Referee #4 about the impact of snow cover. In particular, by performing additional analyses including trends in snow season length and associated variables. We found that trends in snow season length were not related to the diversity metrics calculated in our study (richness change, evenness change, turnover, and species trajectories), but whether this is a meaningful ecological effect or the result of a scale mismatch between the spatial resolution of diversity and snow data remains unclear.

Referees' comments:

Referee #1 (Remarks to the Author):

1.1. I read the revised 2nd version of the manuscript with interest, and I am very satisfied with the new version. All my comments and suggestions for improvement are now integrated (with new analyses and illustrations). The more so, I would find this new version very suitable for publication in Nature.

Thank you very much for your positive assessment of our manuscript. We are glad that the reviewer is satisfied with the new version of the manuscript and appreciate their comments.

Referee #2 (Remarks to the Author):

2.1. In their revised version of the manuscript “Plant diversity dynamics over space and time in a warming Arctic” the authors largely rewrote the text and reanalysed the data – this enormous effort needs to be congratulated.

Thank you very much, we appreciate this comment.

2.2. Overall, I think the manuscript is much improved with many previously critical or unclear sections addressed, changed or deleted. However, the revision couldn't resolve all my main criticisms, largely centred around the dataset and interpretation.

Thank you for this assessment of our revised manuscript. We are glad that the reviewer considers the manuscript much improved, and hope that we have now managed to resolve the remaining criticisms (see below for our detailed answers).

2.3. Specifically, I still wonder what the key message of this manuscript is. At some point (line 547) the authors say that the arctic communities are resistant against warming, but elsewhere is stated that communities lost species over time, mostly due to increasing cover of shrubs. Also it was mentioned that plot diversity is governed by idiosyncratic processes (I guess site-specific is a synonym here). This is not new at all and has been shown countless times before in nearly all ecosystems. Only when zooming out at larger scales more predictable patterns may emerge – which the authors couldn't detect because of the nature of the dataset. That said, I strongly suspect that plots <1m² are too small to address the specific question of vegetation change as response to warming across the whole arctic due to stronger local-scale specific effects (note that local scale richness is also greatly depended on pool size richness, which likely differed a lot depending on geographic location; also some islands were included with probably smaller pool sizes than the plots on the mainlands). May all this taken together add up leading to the weak ecological effects observed?

We have now streamlined the key messages of the manuscript so that they are now much clearer. In a nutshell, the key and novel message of this study is that despite no directional richness change, and no indication of plant community homogenization thus far, the magnitude of turnover in species composition is greater where temperatures have increased the most, while species richness has declined where shrub cover (particularly from erect shrubs) increased most.

Indeed, former studies have reported idiosyncratic changes in species richness in other ecosystems, leading to a net “no-change signal” on average. However, our results rather suggest that diversity change is far from being idiosyncratic. On the contrary, here we showed that this seemingly “no-change trend” in species richness is in fact hiding species turnover that are far more pronounced where it warmed the most and where shrub cover increased the most, something that would not be possible under random and idiosyncratic processes.

To better highlight the main results and this take-home message, we have undertaken a series of changes in the main text. We have done this by (1) moving evenness results to the Supplementary Materials (see **3.2**), (2) removing mentions of resistance of communities (see **3.18-20**), and (3) better emphasising the main message that plant communities have changed in richness and composition in different directions and magnitudes, but not in a idiosyncratic way given that we found more pronounced patterns of composition change where temperatures have warmed the most and where shrub cover increased the most (see **2.4-5**, **3.20-23**).

We have also emphasised the consequences of one of the main messages of our manuscript – that increasing erect shrub cover is correlated with decreasing species richness over time. This result is another novel outcome of our manuscript, given that this relationship hasn't been previously found at such a large scale using local plot data following a standardised protocol. We have also re-named “non-dwarf shrubs” or “tall shrubs” to “erect shrubs”, in order to better reflect the difference in physiognomy, throughout the text and in **Figures 2** and **S14**. Examples of this message being more prominent in the manuscript are:

“Overall, our findings suggest that species may be more at risk where taller shrubs are expected to increase due to aboveground competition for light^{64,65}.” (L556-558).

“Consistent with our hypotheses, where diversity changes do occur, they are mainly driven by the combined effects of warming and plant-plant competition, including erect shrub increases^{2,24}.” (L484-487).

“Thus, Arctic diversity might be more at risk at sites with increasing shrub cover, particularly from erect shrubs (**Figure 2e**).” (L459-550)

We have also fine-tuned the main messages in the first discussion paragraph and in the conclusion paragraph, as per below:

“Contrary to our hypotheses, there was no directional trend in plant richness change to date on average (α -diversity; **Figure 2b, c**), despite the Arctic experiencing the greatest rates of warming on Earth over the past decades¹ (**Figure S5b**). This result ran counter to literature predictions¹⁹, experimental observations of plant diversity declines at the local scale⁵⁴ and modelling studies predicting a regional declines of 15 to 47% in Arctic-alpine plant species richness²². We found that Arctic plant composition and richness change are decoupled, with no net richness change on average despite widespread composition change over time (**Figure 2, 3**). Consistent with our hypotheses, where diversity changes do occur, they are mainly driven by the combined effects of warming and plant-plant competition, including erect shrub increases^{2,24}. Despite the lack of a strong relationship between warming and richness change, both proportional species gains and losses were greater where temperatures increased the most (**Figure 2, 3**). We found a more consistent influence of shrub increases over time, with relatively greater species losses, thus leading to decreased species richness where shrub cover (particularly of erect shrubs) increased the most over time (**Figure 2, 3**). We did not find evidence of homogenisation of Arctic vascular plant communities over time, with no directional temporal changes in spatial dissimilarity of species composition (**Figure 4, S15**), indicating that plant communities changed in their composition in a variety of ways. Overall, we found that despite Arctic plant community composition changing to different assemblies based on local context, both climate warming and shrubification emerged as key factors influencing the magnitude of species turnover.” (L477-498)

“Overall, we found that changes in Arctic plant diversity and community composition depend on local context, with both warming and shrubification emerging as key factors influencing the magnitude of species turnover. Probable mechanisms underlying the observed diversity changes include colonisations from local species pools²⁰, gains of thermophilous species⁵³, losses of less competitive and/or rare species⁵⁶ and increased competition with canopy-forming shrubs²⁴. Our results indicate that we should not expect an overall loss or gain of vascular plant biodiversity with warming in the Arctic. Instead, directional change in plant communities will depend on the combination of changing environmental conditions and available species pools, with warming leading to greater plant community composition change and shrubification resulting in decreasing species richness over time. This research demonstrates the value of long-term *in situ* monitoring at local scales for the detection of

biodiversity change and improving our understanding of biome-wide responses or resistance to climate warming⁸⁰. The extensive reshuffling of Arctic vascular plant composition in recent decades observed in this study underscores the urgent need to explore the impacts of these shifts on ecosystem function, wildlife habitats, and livelihoods for Arctic peoples^{5,6}.” (L615-631)

As for the scaling point, we believe that 1m² plots are entirely appropriate for characterising Arctic plant communities and their changes, considering that Arctic plants are generally relatively small/short individuals. In particular, it's at these scales where the influence of dispersal capacity, environmental filtering and biotic filtering can be apparent, thus reflecting ecological assembly processes at the local scale (Vellend 2016). Importantly, study areas and subsites are represented by multiple plots (**Figure S4**), and the dataset overall includes > 2,000 plots, so the key inferences do not rely on any assumption of single 1m² plots representing entire sites or regions. We have good reasons to assume that these vegetation plots are sufficient to detect any effects of global change - and indeed we do see many responses, even if they are highly variable from place to place. However, these changes are not occurring randomly or idiosyncratically, given that we found directional diversity changes driven by warming and shrubification.

We have now improved the clarity of the explanation of the scaling considerations of this study as follows:

“We kept only plots whose surveyed area was $\leq 1 \text{ m}^2$ in order to ensure comparable richness values across plots, given that plant species richness tends to increase with plot size⁸⁶. Since Arctic plants are relatively small individuals, a plot size of 1m² is appropriate to reflect ecological assembly processes at the local scale⁸⁷. We included the natural log-transformation of plot size in all models (except for evenness) to account for variability among plot sizes to most closely resemble species-area relationship theory^{86,88}.” (L678-684)

Finally, we very much agree that local species pools will influence the magnitude of change that we see in our analyses. Hence, we included ‘mean richness’ as a sampling design covariate in all our models (except the richness and richness change models, see **Table S2**). This is explained in the methods as follows:

“**Sampling design covariates**

All multivariate models (**Table S2**) included a set of relevant sampling design variables to account for different surveying methods (‘plot size’), survey timing (‘duration’) and local context (‘mean richness’). We included the natural log-transformation of plot size in all models to account for variability among plot sizes and for the fact that different plot sizes may lead to different chances to detect changes over time^{17,88}. Mean richness was calculated as the mean values of richness across all years to reflect the most common conditions in a plot over time (**Table S2**). Duration was calculated as the difference between the last and the first years of surveying per plot. See **Supplementary Results** for an overview of the effects of the sampling design variables on biodiversity metrics.” (L847-857)

References:

Vellend, M. (2016). *The theory of ecological communities* (MPB-57). Princeton University Press.

2.4. In sum, what can the manuscript tell us new that hasn't been detected before (also by other studies led by the first author - García Criado et al GEB 2020; García Criado et al Nat Com 2023)? What is the major advancement and where do we need to go from here? All these questions still largely remain unanswered.

This manuscript presents research that is novel both in terms of its scope (pan-Arctic) and our understanding of Arctic ecosystem change. The first message of our study is that, in the region where warming is the most rapid (i.e., the Arctic), plant compositional changes are substantial and widespread, but are not happening idiosyncratically as they are clearly driven by global change drivers. While warming temperatures and increasing shrubification explained the magnitude of the species turnover, we found no directional changes in species richness on average over time among the studied sites, and there is no indication of homogenization between sites over time at a biome scale. This is a highly novel finding countering the widespread argument and prediction that richness is generally declining and that vegetation homogenization is increasing globally. Instead, using a carefully standardised approach, replicated across various spatio-temporal scales, we can draw strong conclusions that better inform our understanding of Arctic biodiversity change.

In particular, we found that the magnitude of change in species composition (temporal turnover in species composition) is predictable, given that compositional change was greater where it has warmed the most, however these compositional changes lead to different assemblies of species community composition (as per the homogenization analyses), which are likely the result of local contexts. This is critical knowledge for defining future approaches to mitigate the impact of global change on the tundra and to develop new monitoring protocols, for key species for instance. It also highlights that the future of Arctic plant biodiversity, in terms of species richness trend and the specific composition of plant communities, is hard to predict, but that compositional changes within Arctic plant communities are likely to be more pronounced under future climate change, which has important implications for understanding cascading effects on higher trophic levels as well as the climate change adaptation options available for people that rely on Arctic vegetation for their livelihoods.

The two other studies mentioned by the referee have different focuses than the current manuscript. García Criado et al *GEB* 2020 calculated rates of woody encroachment (trees and shrubs) across the tundra and savanna biomes, and characterised its relationship with climate. García Criado et al *Nat Com* 2023 showed that commonly measured plant traits (height, SLA, seed mass) in tundra shrubs were not strong predictors of future species range shifts or abundance changes. Amongst other differences in focus, there are two major factors distinguishing these two papers and the current manuscript: (i) García Criado et al., 2020 and García Criado et al., 2023 focused on shrubs, while the current manuscript considers the whole spectrum of vascular plant composition (shrubs, forbs and graminoids) and (ii) while the former two papers considered plant cover and species distributions only, the current manuscript quantifies diversity change through a series of diversity metrics (e.g., richness, evenness, turnover in species composition, species trajectories), thus providing both a deeper and more detailed study of Arctic vegetation change.

Additionally, this manuscript is based on an unusually large database, comprising 42,234 unique records from 2,174 plots within 155 subsites distributed across 45 study areas encompassing 490 vascular plant species, recorded during different intervals over the past four decades (1981 – 2022) across the Arctic. The wide coverage of this database across geographic space, climatic gradients and assemblages of tundra functional groups (**Figure 1**) is unique even beyond the Arctic biome and captures much of the variation across tundra landscapes, providing us with the opportunity to quantify diversity change across the Arctic in a representative manner.

We targeted our revisions to better communicate the main messages of this study and also to help communicate the advance of this study beyond previous studies. See also our responses to **2.3** above and the first paragraph of our response for a short summary of the main message we convey in this revised manuscript.

2.5. I am sorry not to be more optimistic at this point. But I do hope that the authors can use the comments to further refine their manuscript.

We thank the referee for their thorough assessment and for their constructive comments which have definitely improved our work. We hope that all revisions undertaken, and specifically the ones mentioned above (**2.1-2.4**), have addressed the reviewer's main concerns. We appreciate that different reviewers or readers will have different opinions on whether particular aspects of our study are novel or surprising, but we see unambiguous novelty in our combination of fine spatial resolution (detailed field-based vegetation surveys), wide spatial extent (pan-Arctic), temporal depth (four decades), and some clear, albeit complex, results that counter conventional wisdom.

Specific comments:

2.6. Introduction: I very much like how the authors have restructured the introduction making it much stronger now in my opinion.

Thank you very much, we are glad that the reviewer is satisfied with these changes.

2.7. Results: In general, it would be interesting to have the R^2 of the regressions mentioned in the results section as well. Knowing how much variance is explained by the predictors would help to better understand the results.

We have included both the conditional R^2 and the marginal R^2 from the Bayesian models where slope and CI were indicated in the text of both the Results section and the Supplementary Results. We have also included the Bayesian marginal R^2 for the models of the richness change as a function of shrub change and for richness change as a function of forb change (see the legend of **Figure 2e,f** as per comment **2.12** below). We have also included the marginal R^2 for the equivalent models without extreme values in the legend of **Figure S9**, and for the new model on the relationship of species losses as a function of rarity (see **2.15** and **2.18**).

2.8. Line 146: This study doesn't span the whole arctic. Writing it here misleads on scale of the dataset.

We have rephrased this to:

"However, the direction and magnitude of local plant diversity changes have not been quantified thus far at sites across the Arctic." (L147-148)

2.9. Line 224: But what about other herbaceous species traveling polewards as well? Wouldn't these lead to an increase of species richness? Or do the authors argue that shrubby environments are in general poorer in species richness due to their superior competition over herbs? This part is still not clear to me.

An increase of species richness due to herbaceous species (or from other functional groups) moving north could indeed be the case, with species from any functional groups moving northwards. We discuss this pathway, together with the other three main pathways of change in the Introduction, including: (1) increases in species richness as species migrate north, (2) increases in species richness from local neighbouring species pools, (3) decreases in species richness due to losses of cold-adapted species, and (4) decreases in species richness due to competition from colonising species. In this specific paragraph, we wanted to discuss the role of biotic interactions in plant community dynamics, and more specifically how we hypothesise shrubs to be one of the main biotic drivers of change.

The pathways of diversity change are discussed as follows:

“Arctic plant diversity change could be shaped by interacting processes following four pathways. (1) If species migrate northward to track climate warming, we would expect a net increase in overall Arctic plant species richness^{2,18,19}. (2) Richness increases could also result from short-distance dispersal and colonisation events from species that are already present in neighbouring local species pools, as growing conditions improve and communities are potentially able to support more species^{20,21}. (3) Conversely, reduced Arctic floral diversity could result from losses of cold-adapted species²² that cannot cope with increasing temperatures²³. (4) These declines could be exacerbated by increased competition with colonising species originating from Low Arctic and boreal latitudes^{24,25} or by local species becoming better competitors under warmer conditions⁴. Because these pathways may be acting in concert, it is possible and indeed likely that richness increases and decreases could occur simultaneously, resulting in no net richness change. Yet, the effects of these different pathways on current and future Arctic plant diversity trends remain poorly understood. We address this knowledge gap by quantifying the direction and magnitude of Arctic vascular plant diversity change over time at the local level (α -diversity) and temporal turnover in species composition (β -diversity), and investigating which geographic, climatic and biotic drivers are related to different aspects of diversity change in order to understand trends across the Arctic.” (L183-201)

2.10. Line 273: Again, what about shade-tolerant species travelling polewards? Could these compensate the losses?

See response to **2.9** above. Despite the presence of certain shade-tolerant species, we expect that competition from shrubs will displace more species than the ones that are gained. We have now mentioned shade-tolerant species in the hypothesis for RQ1:

“Despite the presence of some shade-tolerant species, we also hypothesize that plant species richness will decline overall where shrub cover increases over time, since sun-loving plants could be out-competed by shading and increased litter production from taller and denser shrub canopies, as per spatial analyses²⁴.” (L277-281)

2.11. Fig 2b: what does this plot show? It says it is richness per plot per year. But to assess directional change in richness over time wouldn't the gains and losses between consecutive sampling events of the same plot needed to be compared? As presented here it seems meaningless as different plots (also of different sizes with larger plots per se having higher richness) have been analysed together? If the model mentioned is accounting for this, it needs to be stated in the figure legend.

Figure 2b shows the overall direction of richness change over time, with each point representing the value of species richness at each plot at a given time. This is a visual representation that conveys the message that species richness has not increased or decreased on average over the past few decades. The model behind this figure indeed considers consecutive sampling events for each plot. The model structure (in **Table S3**) is Annual Richness \sim Year + (Year|Subsite/Plot), with plot as a nested random effect within subsite.

We have clarified this in the figure legend as suggested:

“b) Richness did not change directionally over time. Richness is presented per plot and per year, coloured according to the latitudinal gradient. The dashed line and grey band represent the output from the high-level model in **Table S1**, which includes a nested random effect of plot within year.” (L355-358)

2.12. Fig 2e,f: Thanks for looking specifically at the large variation in the middle of the plots in the rebuttal letter. However, may it be possible to add R² to the figure? Given so many data points a significant relationship can be expected but how much the relationship actually explains would be more informative to know in my opinion. It still looks like to me that the large variation in the middle is preventing any meaningful interpretation of trends.

See response to **2.7** above. We have included both the conditional and marginal R² for the models in the **Figure 2e,f** and their equivalent models without extreme values in the legend of **Figure S9**. We have included the values in the legend instead of inside the graph in order to facilitate visualisation and ease of interpretation of the figure.

2.13. Lines 468-471: This is an interesting finding/statement. Does it mean that changes in species composition are highly site-specific? Do the authors have any means for exploring this further? Do perhaps models with a nested design (random effects) pick up stronger signals with climate accounting for region specific idiosyncrasies? What may these local-scale specific factors be? Herbivory? Management practices? Fire? More discussion on this would be useful to add.

As highlighted in responses **2.3** and **2.4**, we found both decreases in species richness where shrubs had increased over time, and more pronounced species composition changes where it has warmed the most, suggesting that Arctic vegetation is not responding idiosyncratically to climate change. However, these compositional changes led to different assemblies of species community composition, which are likely the result of local context and processes and not so easily predictable. This message is now clearly stated in the revised abstract and we have incorporated a few potential local-scale processes (i.e., topography, microclimate) in the Discussion as follows – see also **4.4** below:

“A better understanding of the underlying mechanisms that drive local biodiversity change will be key to identifying future rates and hotspots of change under accelerating warming^{20,68}. Further research is required to determine whether Arctic plant communities are exhibiting resistance to warming⁶⁹, as additional processes could contribute to a lack of detected richness change on average. For example, the same species could be both lost and gained across plots over time due to stochastic dynamics or sampling effects (**Table S6**). Future change in species richness and composition may not yet be detected due to extinction lags⁷⁰ and slow colonisation rates in communities of long-lived perennial species. Additionally, priority effects could cause heterogeneity in species responses to warming⁷¹. Variation in topography, microclimate and nutrient availability could mediate ecological responses and buffer against climate change impacts by providing microhabitats with suitable conditions^{21,72–74}. Rising temperatures are projected to be accompanied by increasing precipitation leading to a warmer and wetter Arctic, which could ameliorate drought effects on plants⁵⁰. In addition, herbivory may mitigate warming-driven shrub expansion in certain regions⁵⁴. Thus, the integration of extinction lags, priority effects, local context, and both micro- and macroclimate is an essential next step to better identify the mechanisms behind Arctic plant dynamics.” (L583-600)

2.14. Line 485: That is an interesting statement. In general, it is believed that the warmer-edge boundary of a species is due to competition and not to physiological constraints. Can the authors reach any more specific conclusions in this direction? This paper is perhaps an interesting read in this context: Paquette, A. and Hargreaves, A. L. 2021. Biotic interactions are more often important at species' warm versus cool range edges. – *Ecol. Lett.* 24: 2427–2438.

Thank you, this is a great suggestion. We have rephrased this sentence to include this point and cite the suggested paper:

“While gains could represent warm-adapted species expanding into warmer areas, these could outcompete cold-adapted species^{53,56}, with biotic interactions usually being more relevant at species’ warm edges⁵⁷. This could be generating species losses, together with cold-adapted species being unable to cope physiologically with warming.” (L511-515)

2.15. Line 493: This would be an important finding. Would it be possible to test if especially rare species are those constituting the losses the authors found? Given that the dataset would allow testing this it feels a bit shallow just speculating here.

Thank you for this suggestion. We have modelled this relationship and found that species that had been proportionally lost more often were rarer geographically. This is now incorporated in the Results:

“Species that were more frequently lost across plots were generally rarer (i.e., were found at fewer study areas, slope = -0.13, CI = -0.17 to -0.09, conditional and marginal $R^2 = 0.18$).” (L398-400)

And in the Methods:

“To understand whether species losses were related to rarity, we modelled the proportional losses per species (as percent of losses relative to all trajectories across plots) as a function of the number of study areas where the species was present in our dataset.” (L883-886)

2.16. Line 507: well, if species richness is decreasing species must be lost. Maybe delete this last statement?

Yes, where species richness is decreasing, it means that the turnover in species composition (losses and gains) led to greater species losses than species gains (high turnover and low nestedness between the surveys). Since we have performed specific separate analyses for species gains and losses (and for persisting species), we just wanted to highlight this particular result to showcase the relationships with the different species trajectories. We have rephrased to:

“We found a more consistent influence of shrub increases over time, with relatively greater species losses, thus leading to decreased species richness where shrub cover (particularly of erect shrubs) increased the most over time (**Figure 2, 3**).” (L489-492)

2.17. Line 508: What does “Lower species richness has been observed with greater shrub cover spatially” mean?

We meant that the relationship between shrub cover and overall species richness has been formerly characterised across space, using space-for-time substitution, but not examined over time (i.e., with increasing shrub cover over time). We have reordered this section to include the expectation from the scientific literature using space-for-time substitution approaches, and the fact that we found and thus confirmed this pattern to be similar over time (i.e., with increasing shrub cover) in our analyses:

“Across space, lower species richness has been observed with greater shrub cover, with shading and litter production leading to decreases in sun-loving plants under shrub canopies^{24,25}. Using space-for-time approaches, studies have assumed a similar pattern to occur over time, without necessarily testing it. Here, we found and confirmed this pattern over time: where shrub cover increased over time, community evenness decreased and greater species losses occurred, leading to reduced species richness (**Figure 2e, 3f, Table S1**).” (L540-546)

2.18. Line 511: This is a strong conclusion but unsupported. So far, the authors didn't test if less competitive species are being lost.

See our response to **2.15** above. We have now addressed this point in our revised manuscript by modelling proportional losses per species as a function of the number of study areas where the species was present (as a metric of rarity). We found that species that had been proportionally lost more were rarer geographically. We have edited this sentence as follows:

“Our Arctic-wide results corroborate site-level reports that increasing shrub cover over time may lead to less diverse plant communities and the displacement of rare and/or less competitive species^{42,60,61}.” (L546-548)

2.19. Line 519: Or better resistance against herbivory?

Great suggestion. We have included this in the sentence:

“Graminoids were more likely to persist than forbs (**Table S2**), perhaps because graminoids are good competitors that can displace shallow-rooted forbs where they both co-occur due to their deeper root networks, faster nutrient uptake, greater height and better resistance against herbivory⁶²⁻⁶⁴.” (L553-556)

2.20. Line 542: Could you support this statement with data or at least a reference?

Indeed. We have included Bjorkman et al. 2018 and Wookey et al. 2009 as supporting references:

“Continued compositional change is likely to lead to additional shifts in plant traits and the functioning of Arctic ecosystems^{5,65}.” (L580-581)

References:

Bjorkman, A. D. et al. Plant functional trait change across a warming tundra biome. *Nature* **562**, 57 (2018).

Wookey, P. A. et al. Ecosystem feedbacks and cascade processes: understanding their role in the responses of Arctic and alpine ecosystems to environmental change. *Global Change Biology* **15**, 1153–1172 (2009).

2.21. Line 547: This statement comes as a surprise. Was the main conclusion of this piece that the arctic is resistant to warming?

This was poorly worded in our former version. We have now added some nuance to our previous interpretation of ‘resistance’ (see also **3.18**, **3.19**, **3.20** and **3.23**). We have rephrased this statement to:

“Further research is required to determine whether Arctic plant communities are exhibiting resistance to warming⁶⁹, as additional processes could contribute to a lack of detected richness change on average.” (L585-587)

2.22. Line 576: Note that this is the key prediction of MacArthur and Wilson’s Equilibrium Theory of Island Biogeography. There is a huge body of literature linked to that hypothesis and why there is no change in richness despite assumed community turnover. Maybe there are some further clues and ideas in the island literature why arctic communities seemingly didn’t respond to warming possibly also useful for this study?

Thank you for this suggestion. We have now included reference to MacArthur and Wilson's Equilibrium Theory of Island Biogeography as follows:

“This suggests that plant community composition is being influenced by warming (**Figure 3b**), but that species gains and losses within plant communities, on average, balance each other (**Figure 3e, S12**), consistent with some predictions of equilibrium theory⁵⁸ and thus resulting in the observed overall non-directional richness change (**Figure 2b**).” (L515-520)

In terms of species responses to warming, we have made further edits to the wording to better reflect the species responses to warming, and to better highlight the variability of directions and magnitudes of change across the Arctic (see **2.3-5**). We have also nuanced our previous interpretation of ‘resistance’ of these Arctic communities (see **3.18-20**).

2.23. Line 620 and line 637: Did the authors test for additional “plot size correction methods”? Comparing richness of differently sized plots, even at such small spatial differences, can introduce huge variation in itself. Rarefaction could be a solution or testing if same results were obtained if only including 1m² plots for instance.

Thank you for this suggestion. We have followed the reviewer's suggestion of testing whether the same results were obtained when only including 1m² plots in the models. We have re-run two of the models which support some of the main conclusions of the study with 1m² plots only: that species gains are greater where temperatures have warmed the most (model 1 below) and that species richness has decreased where shrub cover has increased the most (model 2 below).

Model 1: species gains. The temperature change estimate retains the same direction and significance in both the original model and the model with only 1m² plots.

Original structure: proportion of gains ~ temperature change + precipitation change + shrub cover change + mean richness + log(plot size) + duration + (1|subsite)

Sample size = 1,266

Temperature change estimate: 7.60, 95%CI = 1.03 to 14.34

Modified structure (only retaining 1m² plots): proportion of gains ~ temperature change + precipitation change + shrub cover change + mean richness + duration + (1|subsite)

Sample size = 631

Temperature change estimate: 10.86, 95%CI = 2.22 to 19.57

Model 2: species richness change. The shrub cover change estimate retains the same direction and significance in both the original model and the model with only 1m² plots.

Original structure: richness change ~ moisture + temperature change + precipitation change + shrub cover change + log(plot size) + duration + (1|subsite)

Sample size = 1,266

Shrub cover change estimate: -0.012, 95%CI = -0.023 to -0.001

Modified structure: richness change ~ moisture + temperature change + precipitation change + shrub cover change + duration + (1|subsite)

Sample size = 597

Shrub cover change estimate: -0.011, 95%CI = -0.017 to -0.004

Considering that both of these main messages from the manuscript are consistent even when only retaining 1m² plots, this indicates that the retaining of plots of different sizes (all under or equal to 1m²) and our method of including log(plot size) in the models is appropriate in this context. We have included mention of this additional correction in the Methods section:

“We tested an additional plot size sensitivity analysis by re-running models behind some of the main manuscript outcomes (**Table S2.45** and **S2.52**) but only with plots whose size was 1m^2 ($n = 631$ and 597 for the main analysis and the sensitivity analysis, respectively). Both estimates of temperature change and shrub cover change had the same direction and significance as their original model counterparts.” (L687-692)

Referee #3 (Remarks to the Author):

Plant diversity dynamics over space and time in a warming Arctic

I was Referee #3 in the previous round.

3.1. For the most part I think the authors have responded strongly to comments and issues identified by all reviewers. Unfortunately, I found the methods remained very hard to parse. And, if I understand the analysis correctly, I remain puzzled by some of the choices made, and find others to be without justification (and approaching untenable). Due to my continued uncertainty of the analyses, I will step through each in turn, before making some suggestions on other aspects of the presentation that the authors might consider.

We thank the referee for their thorough assessment of the manuscript and for all the suggestions. We have taken them on board, made the necessary changes with a particular focus on the Methods, and hope the reviewer finds the revised manuscript improved.

3.2. Spatial models: in the rebuttal the authors show two relationships between richness and plot size: 1) $S \sim \text{plot size}$; 2) $S \sim \log(\text{plot size})$. However, neither of these relationships reflect the one assumed by the richness models fit by the authors, which with the log link, mean you are assuming $\log(S) \sim \log(\text{plot size})$. As I stated in my initial review, I think this is a reasonable assumption, but it does need to be stated clearly in the methods. The other spatial models fit to the evenness metric assumed beta error, and my comment regarding the assumption underlying the inclusion of $\log(\text{plot size})$ in my initial review remained unaddressed. The models with beta error and logit links amount to a logistic (S -shaped) functional form for the relationship between evenness and \log plot size (i.e., $\text{logit}(\text{evenness}) \sim \log(\text{plot size})$). Again, this assumption is not stated clearly in the ms anywhere that I can see. And without further persuasion, I remain unconvinced that it is justifiable. The evenness results from the spatial models receive very little emphasis in the main text, and short of strong justification for either including them in their current form or modifying them somehow, I would recommend removing them. Unfortunately, for this bounded evenness metric no alternative solutions to including sample effort into a regression model come to mind.

In terms of the $\log(S) \sim \log(\text{plot size})$, we have now clearly stated this relationship in the Methods section, as follows:

“Data families

We fitted hierarchical models with different family distributions depending on the structure of the response variable (**Table S1, S2**). These included Gaussian family with an identity link function (for continuous response variables with a normal distribution), negative binomial family with a log link function (for count data where the variance is greater than the mean), beta family with a logit link function (for values ranging between 0 and 1, but excluding 0 and 1), zero-inflated beta family with a logit link function (for values ranging between 0 and 0.99), and zero-one-inflated beta family with a logit link function (for values between 0 and 1, including 0 and 1). For the beta family, we included in our models ‘ $z_i \sim 1$ ’ (where z_i is the probability of being a zero), ‘ $z_{oi} \sim 1$ ’ (where z_{oi} is the probability of being a zero or a one), and ‘ $coi \sim 1$ ’ (where coi is the conditional probability of being a one, given that an observation is a zero or a one). In the case of the spatial richness models (**Table S2.1-5**), the log link function with a negative binomial distribution assumes the relationship between

richness and plot size to be log-log: $\log(\text{richness}) \sim \log(\text{plot size})$. We specified weakly informative priors for beta and negative binomial families. Data families for each model are specified in **Table S1, S2.**" (L787-803)

With respect to the spatial evenness results, we have relegated all mention of the spatial evenness results to the Supplementary Results, as per **3.26** below. The only mention to spatial evenness in the results is:

"Spatial richness and evenness were correlated (**Table S1, Supplementary Results**)."
(L345-346)

Additionally, we have re-run the evenness models without the $\log(\text{plot size})$ term. As per the reviewer's suggestion and after scouting the literature, we found that the evidence of an effect between plot size and evenness is mixed, with studies finding positive, negative and no relationships (Otýpková & Chytrý 2006, Wilson et al. 1999). Thus, we cannot assume that a relationship can be expected between evenness and plot size. All evenness models without the plot size term have the same direction and significance for fixed effects (except for a couple of changes in moisture and richness terms), and we have updated the model outcomes on **Tables S2** and **S3**. We have also discussed this in the Methods:

"Since Arctic plants are relatively small individuals, a plot size of 1m^2 is appropriate to reflect ecological assembly processes at the local scale⁸⁷. We included the natural log-transformation of plot size in all models (except for evenness) to account for variability among plot sizes to most closely resemble species-area relationship theory^{86,88}. We did not include the plot size term as a fixed effect in evenness models as the evidence of a relationship between plot size and evenness is mixed, with studies finding positive, negative and no relationships⁸⁹, and thus there are no clear theoretical reasons to expect such a relationship." (L680-687)

References:

Otýpková Z. & Chytrý M. 2006. Effects of plot size and heterogeneity of vegetation data sets on assessment of evenness and β -diversity. Ms. (Ph.D. thesis, Depon. in: Dept. Bot. Zool., MU, Brno).

Wilson J.B., Steel J.B., King W.McG. & Gitay H. (1999): The effect of spatial scale on evenness. – J. Veg. Sci. 10: 463–468.

3.3. Two time point models: Because plot size is consistent through time, I am puzzled as to why the authors chose to include it in these models. Again, it entails some unexplained assumptions regarding the relationship between metrics and sample effort (i.e., $\text{logit}(\text{dissimilarity}) \sim \log(\text{plot size})$, and $\text{logit}(\text{proportion}) \sim \log(\text{plot size})$). Including plot size in these regressions impacts the estimates of the other parameters, and theoretical reasons for its inclusion are not made clear (nor can I think of any). In contrast, there are good reasons to think that richness and duration will be related to these measures of diversity change, and I like that the authors have included these in their analyses. A strong presentation of these analyses would make the reasoning for all inclusions (or omissions) clear in the methods, and report any sampling effects when describing the patterns found.

Indeed, plot size is consistent through time for each individual plot, and we ensured that all plots with inconsistent plot sizes over time were removed prior to analysis. However, there is variation in plot size among the plots in our database. We retained only plots smaller than 1m^2 in order to ensure comparability across Arctic plots (mean plot size = 0.57 m^2 , range = 0.048 to 1 m^2). See below for a breakdown of the plot sizes across our database (also in **Figure S4**).

Figure S4a. Variability in plot size across our database.

Each plot is the replication unit in most models (unless otherwise indicated, such as the homogenization analyses which are carried at the subsite level). Thus, each plot constitutes one record in the two-time point models, with a set of relevant covariates, and thus each specific plot only has one value of plot size attached to it. Since there is some variability in plot size across our dataset, we included $\log(\text{plot size})$ in all models in order to account for the fact that different plot sizes may lead to different chances to detect changes over time. For instance, larger plots may increase the chances to detect temporal changes in species composition than smaller plots. Indeed, diversity changes are likely to be context-dependent, and so the probability of detecting changes may differ with spatial scales (e.g., magnitudes of change might be lower in a 1m² plot compared to a 20m² plot, which might experience greater proportional losses due to a greater number of species, and greater rates of colonisations due its size). Thus, we should account for plot size in our models. See also response to **2.23** above for an additional plot size correction method. We have now added these points in the Method section to explain why we accounted for plot size in our models, even though plot size is constant over time but not across space.

“We kept only plots whose surveyed area was $\leq 1 \text{ m}^2$ in order to ensure comparable richness values across plots, given that plant species richness tends to increase with plot size⁸⁶. Since Arctic plants are relatively small individuals, a plot size of 1m² is appropriate to reflect ecological assembly processes at the local scale⁸⁷. We included the natural log-transformation of plot size in all models (except for evenness) to account for variability among plot sizes to most closely resemble species-area relationship theory^{86,88}. We did not include the plot size term as a fixed effect in evenness models as the evidence of a relationship between plot size and evenness is mixed, with studies finding positive, negative and no relationships⁸⁹, and thus there are no clear theoretical reasons to expect such a relationship. We tested an additional plot size sensitivity analysis by re-running models behind some of the main manuscript outcomes (**Table S2.45** and **S2.52**) but only with plots whose size was 1m² ($n = 631$ and 597 for the main analysis and the sensitivity analysis, respectively). Both estimates of temperature change and shrub cover change had the same direction and significance as their original model counterparts.” (L678-692)

We are glad that the reviewer agrees that richness and duration are relevant metrics to include in the models of diversity change, and we consider these to be ‘sampling design covariates’. In the same manner, we think that $\log(\text{plot size})$ is another relevant metric that can influence diversity metrics, and we wanted to account for this. We believe that this is an appropriate way to handle the variability found across the plot network, and are further confident in these results as per the additional plot size correction method in **2.23**. We have clarified this in the ‘Sampling design covariates’ section of the manuscript, which now reads:

“Sampling design covariates

All multivariate models (**Table S2**) included a set of relevant sampling design variables to account for different surveying methods ('plot size'), survey timing ('duration') and local context ('mean richness'). We included the natural log-transformation of plot size in all models to account for variability among plot sizes and for the fact that different plot sizes may lead to different chances to detect changes over time^{17,88}. Mean richness was calculated as the mean values of richness across all years to reflect the most common conditions in a plot over time (**Table S2**). Duration was calculated as the difference between the last and the first years of surveying per plot. See **Supplementary Results** for an overview of the effects of the sampling design variables on biodiversity metrics.” (L848-858)

3.4. Temporal models: these were the hardest to understand, and some reconsideration of how the models are presented in methods and supplement are needed if readers are to easily grasp the workflow. Critically, it was not clear to me whether plot size was included in the models to estimate rates of change or not. Table S2 shows the 'Temporal models' all included plot size. But Table S3 shows the 'change over time models' referred to in the methods (L730-731), and does not include plot size. If Table S2 is showing the second-stage analysis, then the inclusion of plot size is less problematic, though I would still question whether the authors want to include it.

We have undergone a major overhaul and re-organization of the Methods section, and hope that these are easier to understand now (see also **3.28**).

We have also better differentiated between the high-level models (section starting L805 in the Methods, **Table S1**) and the multivariate temporal models (section starting L815 in the Methods, **Table S2, S3**).

As per the inclusion of plot size, this is indeed included in the multivariate temporal models as a sampling design covariate (see **3.3**. above), which are indeed second-stage analyses. Plot size is not included as a covariate in the high-level models (**Table S1**).

3.5. If I am understanding correctly, Table S3 shows the first stage models and should precede the temporal models currently detailed in Table S2; relatedly, greater clarity is needed in the methods to clearly delineate the first and second stage analyses. Any inclusion of plot size in models used to estimate the rate of change will impact the estimated rate of change, and I don't believe that is what the authors want. A model to estimate the rate of change without plot size estimates the rate of change assuming that the plot size is constant (which is a fair assumption for these data - it is constant through time). But when plot size is included, the estimated rate of change is adjusted for the plot size (which didn't change, so seems to be an unwanted, unnecessary adjustment). Similarly, when plot size is included in the second stage analysis it will impact the other coefficient estimates, and I am unaware of any theoretical reason to think multiplicative (proportional or log-scale) rates of richness change should be grain size dependent. The highly non-linear model for temporal changes in evenness (due to the logit-link) is even harder to form an expectation for any theoretical relationship with grain size.

As per our responses to **3.3** and **3.4** above, plot size is not included as a covariate in the high-level models estimating rates of change (richness change over time and evenness change over time, **Table S1**). We hope that with the re-organization of the Methods section, the differentiation between high-level models and multivariate temporal models is clearer. We have also taken the reviewer's suggestion to better reflect this logic, and moved **Table S3** (now **Table S1**) before **Table S1** (now **Table S2**). **Table S2** is now **Table S3**. We have updated Table numbers throughout to reflect the new order.

3.6. Spatial beta-diversity: I was reassured to read that the analysis of spatial beta-diversity through time used plots of the same size (>99% of subsites). The authors also mentioned the variation among the spatial extent of the subsites (i.e., # of plots * size of plots), and they should confirm that the spatial extent of the plots within subsites (and the number of plots within subsites) was the same for each time point in the beta-diversity analysis.

This is indeed the case for both instances. Only plots with at least two timepoints and five years of monitoring duration were retained for the spatial beta-diversity analysis (and for all temporal analyses), meaning that every plot will be represented at the start and at the end for each subsite. All plots within a subsite have the same plot size, in general and over time. The size of all plots within a subsite is the same for each time point (start and end). See summary figure below:

Figure showing that the plot size within each subsite is the same for each timepoint. Each plot size has two columns: yellow for the starting time point and blue for the ending time point.

The number of plots within subsites is the same between start and end point for each subsite – see summary figure below:

Figure showing that the number of plots per subsite at each timepoint is the same. Each subsite has two columns: yellow for the starting timepoint and blue for the ending timepoint.

3.7. Stepping out of the weeds of the analyses, which with some further clarifications, adjustments, and a greatly improved presentation, should prove to serve the authors aims well, I remain convinced that this work has the potential to make an important contribution.

We thank the referee for this assessment. We hope that the additional clarifications and adjustments that we have provided in this newly revised version of our manuscript are satisfactory.

3.8. L152: do you mean no directional local richness changes *on average*? Looks to me that some locations go up, others go down, resulting in no directional trend on average. Indeed, you report greater proportional species losses and richness declines in locations where tall shrubs have increased in cover. And conversely, show a relationship between changes in forb cover and richness changes that implies species richness increases in locations where forb cover increased.

Indeed, this is a good point. We have rephrased this sentence to:

“We found greater species richness at lower latitudes and warmer sites, but no indication that local species richness was changing directionally over time, on average.” (L152-154)

3.9. Aside from my desire to see the authors clarify that they find no directional local richness trend *on average*, I think the authors do a good job of visualising and reporting uncertainty on their figures and throughout the manuscript.

Thank you very much, we are pleased that uncertainty comes across clearly throughout the manuscript.

3.10. L197: if only it were so simple! What about history and biogeography? E.g., Ricklefs & He 2016 PNAS.

Thank you for this suggestion. We have added the suggested reference and rephrased this sentence to:

“Apart from evolutionary history and biogeography, species richness patterns at large scales are broadly driven by climatic gradients²⁶.” (L203-204)

3.11. L320: richness did not change *on average*. Figure 2c suggests that tens, possibly hundreds, of plots had changes $\geq 20\%$ of their species (i.e., slopes of $|0.2|$ or greater) per year.

Corrected to:

“Despite greater plant richness at lower latitudes and warmer sites, Arctic plant richness did not change directionally over time, on average (slope = $0.0021 \log(\text{species}) \text{ year}^{-1}$, 95% CI = -0.0002 to 0.0043 , equating to 0.01 species gain per year, conditional $R^2 = 0.63$, marginal $R^2 = 0.003$; **Figure 2b, c, Table S1**).” (L329-333)

3.12. L362: not sure how to parse changes in species abundance in relation to figure 3. Do you mean changes in species relative abundance? Or in light of figure 3, would referring to changes in species composition be most accurate?

We have rephrased this section as follows:

“Nearly all (99%) of the plots experienced changes in species composition (Bray-Curtis), with 59% of plots either gaining or losing species (Jaccard, **Figure 3a-c**). Arctic communities experienced a mean temporal turnover of 0.22 (Jaccard) and 0.36 (Bray-Curtis) [data bounded between 0 and 1], representing presence-absence (Jaccard) and both presence-absence and abundance-related turnover at the plot level (hereafter, ‘abundance-related turnover’).” (L381-386)

We have also included more detail in the caption of **Figure 3**, so that the messages are better signposted:

“Nearly half of the plots (526 plots, 41.5%) did not change at all in terms of presence-absence turnover (Jaccard) and only 6 (0.4%) plots did not change at all when considering both presence-absence and abundance turnover (Bray-Curtis). These plots are indicated by a turnover value of 0 in **a-c**.” (L412-415)

See also **3.17** below.

3.13. L402-410: If I am reading Figures 4d-f correctly, they show the percentage of species being gained or lost. All results referring to these panels should describe proportions not numbers. All descriptions of these results should be carefully checked throughout.

We believe that the reviewer is referring to **Figures 3d-f** here. If that is the case, then this is correct, as we’re referring to proportions and not numbers. We have gone through the manuscript and ensured that gains and losses are always referred to as proportions (see **3.15, 3.16**).

3.14. L410-413: what is important to see here? For example, I expected negative relationships between duration and dissimilarity. But S11b shows the opposite for Bray-

Curtis! Help readers by unpacking some interesting relationships (or non-relationships) here.

We have expanded on this result in the Supplementary Results – Sampling realities section, as follows:

“We found that studies covering longer temporal extents had increased their abundance-related turnover (Bray-Curtis) and richness over time, but not increased species replacement (Jaccard) (**Figure S11**). There was an indication that shorter studies identified more species losses, possibly suggesting that local extinctions might be due to stochastic processes or sampling effects during monitoring⁸. These two results could confirm that long-term studies tend to reflect meaningful change in communities, while shorter timescales often overrepresent real changes in Arctic communities⁹. Finally, larger plots were more even and had experienced less richness change^{10,11}. Together, these findings indicate the importance of scale, local context, study design and sampling effort when drawing global inferences from local-scale monitoring.”

For the purposes of manuscript length, we have relegated to the Supplementary Materials those results that describe the general context of our plots (e.g., climate change rates), the database (e.g., functional group composition), or that deal with considerations when interpreting the results (e.g., sampling realities). These are signposted in the main text as follows:

“See **Supplementary Results** for the effects of geographic and sampling design variables, additional turnover and evenness results, overall functional group composition, and climate change context.” (L437-439)

3.15. L460-461: is it accurate to refer to your turnover estimate as a rate? And again, I think you mean higher proportions of species lost or gained, not rates of species losses and gains.

We have edited this section to better reflect the main messages of the manuscript, and this sentence is no longer relevant. We no longer refer to turnover as a rate in the new sentence:

“We found that Arctic plant composition and richness change are decoupled, with no net richness change on average despite widespread composition change over time (**Figure 2, 3**).” (L482-484)

3.16. L484: proportions of species gained or lost. This needs to be remedied throughout.

Rephrased to:

“Species richness increases were not greater at sites with greater rates of warming over time (**Figure 2d**), but warming was associated with proportionally greater species gains and losses (**Figure 3e**).” (L509-511)

We have also sorted this throughout and added the ‘proportional’ qualifier as required in lines 285, 295, 428, 432, 434, 435, 488, 529, 566, 571 and 885.

3.17. L490: this second reference to ‘changes in species abundance’ is no more illuminating than the first (L363). Do you mean changing composition by altered species relative abundances (as revealed by the Bray-Curtis analyses)?

Thank you for this suggestion. We have rephrased to:

“With 99% of plots experiencing composition changes via altered relative species abundance (Bray-Curtis > 0), and 66% of plots gaining and/or losing species (Jaccard > 0), composition change could begin to influence richness change over time.” (L520-522)

See also 3.12 above.

3.18. L523-530: I don't see any evidence for resistance on Figure 2b, c. I see that plot-scale richness did not change on average (Fig. 2b), because the number of plots where richness went up were approximately balanced by other plots where it went down (Fig. 2c).

We have edited this paragraph (including the header) to reflect that communities changed in a variety of ways, rather than them exhibiting resistance. See also 3.19, 3.20 and 3.23. This paragraph now reads:

“Plant diversity changed in multiple directions among rapid warming

Our findings demonstrate that Arctic plant richness changed in different directions (**Figure 2b, c**) among continued warming.” (L560-562).

3.19. I appreciated the acknowledgement that smaller proportions of species gained and lost in more species rich plots is expected by chance alone, However, it didn't silence my questioning the strength of the evidence for the resistance assertion. Indeed, I found the assertion of resistance weak and distracting.

We have removed mention of resistance throughout the Discussion (see our responses to 3.18, 3.20 and 3.23). In this particular instance, when discussing plots with greater species pools having proportionally fewer gains and losses, we have edited the text to reflect that resistance is a potential explanation for this pattern (as seen in the diversity-stability literature), but that it could also be due to a statistical artefact, so that both options are stated as potential explanations but resistance is not inferred throughout. The text now reads:

“Plots with high species richness and more even communities showed the least amount of change, with a lower proportion of species losses and gains (**Figure S10**). This pattern could be a statistical artefact due to smaller species pool sizes leading to proportionally greater gains and losses, or be a result of greater community resistance, due the reduced extinction risk derived from greater richness and lack of species dominance^{43,66}, as per the diversity-stability relationship⁶⁷.” (L563-569)

3.20. Any link between spatial patterns (i.e., homogenisation and differentiation) and resistance implied by the subheading were not clear to this reader. I think the homogenisation/differentiation result is interesting, but if you want to link it to resistance, much more is needed. Why not simply discuss it on its own terms? Finally, the inclusion of homogenisation/differentiation here mixes scales unnecessarily.

We agree with the reviewer and have removed mention to resistance as a result of lack of homogenisation/differentiation. See also our responses to 3.18, 3.19 and 3.23. We have also edited the sub-header of this Discussion section to remove resistance, and is now titled 'Plant diversity changed in multiple directions among rapid warming' (L560) and all content in this paragraph has been edited to reflect this message.

3.21. L544: I'd assert you detected a lot of changes, just no change in richness *on average*.

Corrected. This sentence now reads:

“Further research is required to determine whether Arctic plant communities are exhibiting resistance to warming⁶⁹, as additional processes could contribute to a lack of detected richness change on average.” (L585-587)

3.22. L572: Here and throughout, I think you’d be well served to say you detected no directional trend in plot-level vascular plant species richness *on average*.

Rephrased to:

“Contrary to our hypotheses, there was no directional trend in plant richness change to date on average (α -diversity; **Figure 2b, c**), despite the Arctic experiencing the greatest rates of warming on Earth over the past decades¹ (**Figure S5b**).” (L477-479)

3.23. L582-585: as stated earlier, I don’t see any development of a strong conceptual link between no change in spatial beta-diversity and resistance. Resistance is first mentioned in the discussion (L464), remains vaguely defined, and in addition to the undefined link with no change in spatial beta-diversity, appears to be additionally inferred by an empirical pattern expected solely by chance (i.e., proportionally fewer species gained and lost in more diverse plots).

We agree and have edited this sentence, which now reads:

“We did not find evidence of homogenisation of Arctic vascular plant communities over time, with no directional temporal changes in spatial dissimilarity of species composition (**Figure 4, S15**), indicating that plant communities changed in their composition in a variety of ways.” (L492-495)

See also our responses to **3.18-3.20** for related edits about resistance.

3.24. L683-685: This seems like sound reasoning, though it would be unwise to think (or imply) that these metrics are independent of each other, e.g., I expect richness and the chosen evenness metric are (strongly?) correlated for these data.

We thoroughly agree and this is indeed the case for some of the metrics (i.e., richness and evenness are indeed strongly correlated, see **3.2** and **3.26**). We have edited the wording here to ensure that this is not implied:

“We chose to analyse common biodiversity metrics that capture species diversity, dominance, and composition change, rather than composite indices, in order to examine the specific elements of biodiversity in isolation from each other.” (L750-752)

3.25. L706-708: First, why convert to proportions? And were all proportions calculated using the same total number of species in both time points combined. It could be argued that the proportion of colonising species (i.e., species gained) should be a proportion of the species in the second time point only (i.e., divided by the total number of species in the second sample). And conversely, the proportion of species lost would be a proportion of species in the initial sample (i.e., the total number of species in the first sample). More rationale and detail is required here.

The reason why the species trajectories (gains, losses and persisting) are converted to proportions is to account for the inherent variability in the number of species present in each plot. By converting to proportions, we can compare trajectories across plots spread across the Arctic in a meaningful way, while using species numbers wouldn’t yield comparable results.

Indeed, the proportions were calculated using the total number of species in both time points combined (i.e., the total number of unique species present in each plot across timepoints). While these metrics could certainly be calculated as the reviewer suggests, we decided to use this approach in order to obtain a full overview of the different trajectories of all the species present in a given plot, relative to the overall number of species. This allows for comparability of the different trajectories across plots (**Figure S12, S13**).

We have added further clarification to the Methods section:

“We considered species locally ‘lost’ if they were originally surveyed in a plot, but were not present in the last resurvey. Similarly, local ‘persisting’ species are those that were present at both the starting and ending year of the monitoring period. Species ‘gained’ are those absent during the baseline survey, but occurring in the last resurvey. These species trajectories were originally calculated as counts, and then transformed to proportions in order to account for the inherent variability in species richness across plots. Species proportions were calculated by dividing the number of species per trajectory in a plot by the total number of species in each plot at both time points combined (i.e., total number of unique species present at each plot in both timepoints, including losses, gains and persisting species). This approach allows for an overview of species trajectories per plot, and also for comparability across plots.” (L768-778)

3.26. L719: you don’t report any evenness results from the spatial models in the main text. And given the (from my perspective) problematic relationship assumed with plot size, I recommend omitting this analysis.

We have relegated all mention of the spatial evenness results to the Supplementary Results, for the reasons the reviewer mentions plus the fact that spatial richness and evenness were correlated (see **3.2, 3.24** above). This is now signposted as follows:

“Spatial richness and evenness were correlated (**Table S1, Supplementary Results**).” (L345-346)

The only mention to spatial evenness results in the Supplementary Results are as follows:

“Spatial richness and evenness were correlated, with more diverse plots also being more even (**Table S1**). Mean evenness (Pielou) across the Arctic was 0.7 [data bounded by 0 - 1]. Evenness was greater in more diverse plots with high forb cover and low shrub cover, and in Western North America relative to other regions (**Table S3.6 - 10**).”

3.27. L724-734: I found this description of the temporal models (in particular), and all the analyses more generally, very difficult to follow. And the lack of clarity means I still don’t know whether plot size was in the models used to estimate the rate of change. If I understand correctly, the temporal models shown in Table S2 do not have “multiple values over time” (L725), and instead show the second stage analysis of the rates of change previously estimated. This paragraph appears to mix single- and two-step analyses all together, and caused considerable confusion and frustration for this reader. Are the first stage models fit to estimate rates of change described under the next sub-heading “additional models”?

As per suggestions made by Reviewer #3 in **3.3, 3.4** and **3.5** above, and **3.28** below, we have modified the Methods section accordingly and we hope that the revised Methods section now reads clearer after this major re-organization as per the reviewer’s suggestions. We have emphasized differentiating between high-level models and multivariate temporal models, and signposted this throughout in the Methods, Results and Supplementary Materials.

3.28. The structure of methods needs an overhaul. Currently there are multiple “model’ sections. And the multi-section description of (the many) models is separate from the specification of their response distributions and link functions. This means that the reader has to identify which of the numerous models or response metrics previously (or yet to be [ordination and posthoc analyses]) described would have normally distributed response, etc. For example, the general statement of what the negative binomial is used for (count data with variance > mean) doesn’t much help the reader link it to your analyses of species richness. Most importantly, the logical flow of your analyses is broken; for example, *I think* the clearest description of first stage analyses quantifying rates of change in richness and evenness come after the second stage models are described.

We thank the referee for making this suggestion and we have now undergone a major overhaul of the Methods section as suggested by the reviewer, and hope this reads clearer now. The current order of the section is as follows: 1) plant composition data, 2) climate data, 3) biodiversity metrics, 4) statistical analyses, 5) data families, 6) high-level models, 7) multivariate models, 8) sampling design covariates, 9) *post hoc* analyses, 10) additional models, and 11) ordination analyses. We have also clarified certain sections and paired down when content was duplicated.

We have also more clearly signposted to the specific data family information included in **Table S1** and **S2** in the Methods section ‘Data families’:

“Data families for each model are specified in **Table S1, S2.**” (L802-803)

3.29. Fig. 2a: this is very hard to read, and the caption does not help much. It appears that these results are unrelated to the analyses, and it is hard to know exactly what is being shown. E.g., what is magnitude of change? Proportional change or something else? You state in the caption that the average changes reported are for visualisation only, but the citations to this figure report and infer patterns. Moreover, without latitudinal contours on the map, it was hard for me to see the patterns the authors allude to.

We have simplified both the legend and the figure caption for **Figure 2a**. We have also included the latitudinal contours in the map every 10 degrees latitude, and increased the font size of the legend. The figure now looks as follows:

FIGURE REDACTED

The magnitude of change (indicated now in the legend as “Mean plot richness change per study area (species/year)”) are the estimates of richness change over time (as number of species per year) in the linear models that constitute step 1 of the temporal models (see **3.31**). We have specified this unit in the Figure legend itself, next to the map, and we have clarified the caption of **Figure 2a** to reflect all these changes as follows:

“**Figure 2. There was no directional change in species richness across the Arctic on average. a)** There was no clear relationship between species richness change and latitude (**Table S3.51**). Richness change values were calculated as the slope estimate from the linear models of richness change over time per plot (number of species per year), and then averaged at the study area level ($n = 25$) to be represented in the map for visualization purposes only. Points are coloured according to their richness change value as the number of species per year (including positive and negative values) and sized according to their absolute values of richness change. Polar projection with latitudinal bands specified every 10° latitude.” (L348-355)

3.30. Fig. 2d: do these subsite level estimates come from the same model shown on Fig. 2b, c?

Indeed. We have clarified this in the caption of **Figure 2**, but also edited the previous sentences to better signpost that the estimates from **Figure 2b, c** and **d** come from the ‘high-level models’, as now described in the Methods (as per **3.4**):

“**b)** Richness did not change directionally over time. Richness is presented per plot and per year, coloured according to the latitudinal gradient. The dashed line and grey band represent the output from the high-level model in **Table S1**, which includes a nested random effect of plot within year. **c)** Mean richness changes across all plots that were surveyed at least twice over at least five years ($n = 1,266$ plots), calculated as the slope of richness over time per plot. The dashed blue line represents mean richness change. Histogram bin width is 0.1. Model structure and output is from the high-level model in **Table S1**. **d)** Richness did not increase at subsites where long-term warming trends were stronger (warmest quarter temperatures). Points represent richness change computed from slope estimates at the subsite level ($n = 90$), as extracted from the high-level model in **Table S1**, and are coloured according to climatology (long-term temperature means).” (L355-364)

3.31. Fig. 2e, f: A reader has no way of knowing where these estimates of richness change come from at this point. Are these the plot-level estimates (i.e., individual slopes aggregated on 2c), or subsite level estimates (as per 2d)? If yes, why did the units change? If no, where did they come from?

We have clarified where the estimates of richness change are obtained for Figure 2e, f analyses:

“**e)** Richness decreased where erect shrubs (but not dwarf shrubs) increased over time (conditional $R^2 = 0.16$ and marginal $R^2 = 0.05$ for model without shrub categories, and conditional $R^2 = 0.08$ and marginal $R^2 = 0.007$ for model with shrub categories, **Table S3.52, S3.52b**). Richness change estimates (species/year) are extracted from the richness over time linear model per plot. Points are coloured according to mean shrub cover per plot over time.” (L365-371)

The high-level models (**Figure 2b, c, Table S1**) are intended to quantify the mean richness and change estimate across the tundra. The subsite-level estimates from this model are then extracted for the temperature analysis (**Figure 2d**) given that climatic data is extracted at the subsite level (and not at the plot level) since we geographic coordinates are only available at the subsite level (and not at the plot level). The temporal analyses of richness change over time and evenness over time (indicated with the clock in **Table S2**) are performed at the plot level and follow a two-step approach: 1) a linear model is fit (richness ~ year for each plot), 2) the estimates act as response variables in the plot change (PCHG) models against different fixed effects (e.g., moisture, functional group change, etc.). We have added further detail to the ‘High-level models’ subsection in the Methods section so this information is better cross-referenced throughout the manuscript:

“High-level models

To obtain the mean richness and evenness change estimate across the tundra, we fitted hierarchical models of richness and evenness per year over time and included nested random slopes per plot within the subsite (**Table S1**). In these two models, the year covariate was centred as needed to achieve model convergence. From the richness change over time model, plot-level estimates were extracted to visualize overall richness change over time (**Figure 2b, c**) and subsite-level estimates were extracted to fit the richness change ~ temperature change model (**Figure 2d, Table S4**).” (L805-813)

Referee #4 (Remarks to the Author):

4.1. The authors report results from a large study of plant diversity observations, undertaken across the North American and European Arctic. Many of these observations represent a time series at specific locations, while others are from single points in time. The authors perform an impressive set of statistical analyses to assess if changes in plant diversity can be identified, if any changes have a particular direction and they relate these changes to climate in the Arctic. As plant biodiversity is not my specialist subject area, this review confines itself to assessing the climate part of the paper. Overall, and as a non-biology specialist, I found this a very interesting and informative study to read, particularly as the observations are widely spread spatially (the lack of Russian observations, for presumably obvious reasons is to be regretted, but not a reason to reject this study) and with a temporal component. Given the widespread interest in climate impacts in the Arctic, I find this a helpful contribution with the focus on ecosystems.

We thank Referee #4 for their positive assessment of our manuscript. We are glad that the manuscript is of interest, and particularly for climate scientists and those outside of the plant ecology discipline.

4.2. Regarding the climate analysis, the CHELSA dataset could be considered a slightly unusual choice in the climate community, but it is well established in the climate impacts community. Given the importance of well-resolved topography, and the lack of detailed observation networks in the Arctic, it is also a good compromise to use as a homogenous dataset especially as the main basis for CHELSA in this region is the ERA-5 reanalysis, which is well documented in the Arctic, though it does also include a well-known but small warm bias. The coming CARRA-2 high resolution reanalysis will be an even better choice in the future but is not yet available. I have two further points that I think should be included.

Thank you for this comment. We agree that CHELSA is well-established when researching climate impacts, and certainly in the scientific field of ecology – a quick Google Scholar search yields 6,600 studies using CHELSA climatic data. We will keep an eye out to incorporate the upcoming CARRA-2 products in future projects once it becomes available.

4.3. 1) The authors assess temperature and precipitation changes in the paper, but one aspect missing that I consider may be important is the impact of the snow season – that is both any systematic changes in snow depth as well as how long snow lies on the ground at each site. There are well-documented trends in snow season length and variability, specifically a shortening of the season due to earlier melt onset but also a later start in some regions. Given the importance of snow cover on ground temperatures, I would be surprised if this was not also significant for vegetation dynamics. As snow depth on the ground and ground temperatures are often subject to systematic biases in model products, I would suggest the authors look at the satellite-derived datasets as well as in-situ snow depth observations, rather than relying on model products for this data. There are a number of high resolution data products that would be relevant here including the ESA CCI snow products. Local temperature change gradients should also include this effect, but I am uncertain if it is included in CHELSA and the documentation does not mention it.

Thank you very much for this suggestion. We very much agree that snow plays an important ecological role in plant communities, and snow trends have certainly changed over time in recent decades across the Arctic.

While *in situ* snow depth data is not currently available for our sites, since many of them are remote and not accessible outside the growing season, we have looked into the high-resolution data products suggested by the reviewer, and specifically the ESA Snow Climate Change Initiative (Snow_cci): Daily global Snow Cover Fraction - snow on ground (SCFG) from AVHRR (1982 - 2018), version 2.0 data at 5 km by 5 km resolution.

Unfortunately, given the nature of remote sensing data in Arctic regions, we have found a large majority of missing values across our study region which don't allow us to be confident in our calculations of metrics such as starting melt date, first snowfall of the year, and overall snow season length.

Firstly, out of the 155 subsites present in our database, we had to remove 10 subsites for the snow analyses given that they were classified as permanent ice/snow or water, and thus no snow data was available for our coordinates for those 10 subsites.

For the 145 remaining subsites, there was a large number of missing values (mean = 250, range = 142 – 333 missing days per year and subsite) due to the polar night, cloud cover, input data errors and no satellite acquisition (see Figure below). This means that there is an average of 115 available days of snow data per year and subsite.

Figure showing an overview of missing days with snow cover data per year and subsite, as extracted from the ESA CCI AVHRR dataset. The dotted line represents the mean missing days per year and subsite.

Thus, considerable gap-filling would be required, featuring site-specific knowledge, in order to have confidence in the metrics calculated here. After consulting with climate scientist colleagues, we have learnt that gap-filled versions of these products are currently underway, but not publicly available yet nor ready to be cited.

However, we still attempted to calculate snow cover length as the difference between the first observed snow melt day of the following year (i.e., the first day when snow cover is <100% and data are available) and the first snowfall after the summer in the previous year (i.e., first day of available data when snow cover is > 0% after the last day when snow cover was 0%).

While this is the usual method to calculate snow season length, this protocol means that sites that never had recorded data of 100% snow cover (due to missing values) result in NA values, and the same is the case if a site was never recorded as having 0% cover during the summer. This NA creation drags throughout the process of the calculation of snowmelt date and snow length. After removing these values, we were left with a dataset with 4,040 site-per-year combinations (for 145 subsites). The timeseries available for each site varied widely (mean = 27, range = 7 to 36 years per subsite).

We have provided a figure for the two metrics per subsite below for information: first melting day and overall season length. As it can be observed, both first melting day and snow season length are widely variable across years and between subsites within the same site. This goes against expectations given that snow melt has been reported to be advancing (Foster et al. 2008), snow season length is generally decreasing (Cooper 2014), and the fact that snow distribution patterns are highly repeatable over time in Arctic, sub-Arctic and alpine environments (Crumley et al. 2024), which is not reflected when reordering locations based on their relative melt data and season length, giving us little confidence that this method reflects *in situ* conditions.

We believe that this wide variability across subsites and time is due to the missing values in the snow cover dataset, and do not reflect actual trends, considering expert knowledge of the sites within the co-authorship team, and the published literature.

Figure showing trends in first snow melting date per subsite. Lines reflect subsites and are coloured according to their study area.

Figure showing trends in snow season length per subsite. Lines reflect subsites and are coloured according to their study area.

While the ESA CCI products seem to be of high quality and resolution, it appears that the temporal resolution (i.e., number of days available) is not comprehensive enough to confidently detect trends, calculate snow-derived metrics and thus trust any potential results when linking it with our biodiversity metrics. We hope that peer-reviewed gap-filled versions of these data, together with an increase in site-specific snow depth and cover data collection, are made available soon in order to enable these kinds of analyses to take place in the future, which as the reviewer points out, are particularly relevant for Arctic plant ecology.

However, we wanted to take the reviewer's suggestion forward and proceeded to extract data from another dataset available across the Arctic: the 'Bioclimatic atlas of the terrestrial Arctic' (ARCLIM, Rantanen et al. 2023). This database includes a variety of climatic indices

calculated from the hourly ERA5-Land reanalysis data for the period 1950-2021 and available as spatial grids of 0.1 degree resolution (~9 km resolution). ARCLIM contains bioclimatic variables of interest that have been tailored for the Arctic with ecological impacts in mind. Here, we have extracted from ARCLIM the temporal trends over the long-term period of 1950-2021 for three variables: “snow season length”, “onset of snow season” and “end of snow season”. Out of the 155 subsites in our database, snow trend data was available for 144 subsites for snow season length, and for 142 subsites for both snow season onset and end of the snow season.

The ARCLIM dataset is derived from ERA5-Land, which is a state-of-the-art global dataset for various land applications. As ERA5-Land is still a relatively new dataset, only a few studies so far have validated its performance against in-situ measurements of snow cover or snow season length (see the “Technical Validation” section in Rantanen et al. (2023) for more information on the quality of the ARCLIM dataset regarding snow-related variables). On average, snow season length had shortened by 0.14 days per year during 1950-2021, with the onset of snow season starting 0.09 later per year, and with snow season ending 0.04 days earlier per year (see Figure below).

Histogram showing the temporal trends, over the period 1950-2021, in three snow-related variables for a total of 144 subsites for snow season length, and for 142 subsites for both snow season onset and end of the snow season. Dotted lines indicate the average value of change for each variable.

Despite comparatively more data available both spatially and temporally in ARCLIM than in CCI, we are cognizant of a scale mismatch between our plot sizes. The scale at which diversity metrics were calculated in our study (1m² or smaller) is much smaller than the scale at which snow data is available in ARCLIM (9 km by 9 km resolution). Thus, it remains uncertain to which extent these snow data actually reflect the processes observed inside the plant monitoring plots.

However, we ran a selection of models using these three snow variables as fixed effects variables (i.e., temporal change in snow season length, temporal change in the onset of the snow season, and temporal change in the end of the snow season) to explain temporal changes in our biodiversity metrics. Overall, none of these three snow variables was significant (i.e., their 95% credible intervals overlapped zero). A summary can be found below:

- Richness change ~ Change in snow season length + Change in snow season onset + Change in end of snow season + log(Plot size) + Duration + (1|SiteSubsite). Gaussian family. All snow parameter estimates overlapped zero.

- Evenness change ~ Change in snow season length + Change in snow season onset + Change in end of snow season + Mean richness + Duration + (1|SiteSubsite). Gaussian family. All snow parameter estimates overlapped zero.
- Jaccard ~ Change in snow season length + Change in snow season onset + Change in end of snow season + Mean richness + log(Plot size) + Duration + (1|SiteSubsite). Zero-one inflated beta family. All snow parameter estimates overlapped zero.
- Bray-Curtis ~ Change in snow season length + Change in snow season onset + Change in end of snow season + Mean richness + log(Plot size) + Duration + (1|SiteSubsite). Zero-one inflated beta family. All snow parameter estimates overlapped zero.
- Persisters ~ Change in snow season length + Change in snow season onset + Change in end of snow season + Mean richness + log(Plot size) + Duration + (1|SiteSubsite). Zero-one inflated beta family. All snow parameter estimates overlapped zero.
- Species losses ~ Change in snow season length + Change in snow season onset + Change in end of snow season + Mean richness + log(Plot size) + Duration + (1|SiteSubsite). Zero-inflated beta family. All snow parameter estimates overlapped zero.
- Species gains ~ Change in snow season length + Change in snow season onset + Change in end of snow season + Mean richness + log(Plot size) + Duration + (1|SiteSubsite). Zero-inflated beta family. All snow parameter estimates overlapped zero.

Whether this lack of a relationship between snow season length trends and plant diversity is the result of true ecological processes, or of a scale mismatch, remains unclear. Thus, this precludes us from making ecological inferences on the effect of snow season length and plant diversity change. We have captured this in the Methods - Additional models section (see below), but we defer to the editor and/or referee on whether this new section improves the manuscript, otherwise we are happy to move or remove it.

“Snow is another important driver of tundra plant composition. However, analyses of satellite remote sensing products providing snow cover variables¹⁰¹ showed that gridded layers of snow-related variables contained too many spatial and temporal gaps to generate a reliable time series of snow-cover duration at our sites. Instead, we extracted data on temporal trends, over the period 1950-2021, for three snow-related variables: snow season length, onset of snow season and end of snow season. These three variables were downloaded from the ‘Bioclimatic atlas of the terrestrial Arctic’ database (ARCLIM)¹⁰², at a spatial resolution of ~9 km by 9 km. We fitted a selection of mixed-effects models to analyse temporal changes for a series of biodiversity variables (richness change, Jaccard turnover, Bray-Curtis turnover, persisters, gains, losses, evenness change) with these three snow-related variables as fixed effects, together with sampling design variables (plot size, duration, mean richness). None of the snow variables were significant in either of these models. This might be due to a non-significant ecological effect of snow season length on diversity trends, or instead the result of a scale mismatch. The spatial resolution at which diversity metrics were calculated is 1m² or smaller, while the spatial resolution at which snow data are available is 9 km. Thus, this scale mismatch precludes us from making any ecological inferences on the effect of temporal trends in snow season length on plant diversity change.” (L896-914)

References:

Cooper, E. J. (2014). Warmer shorter winters disrupt Arctic terrestrial ecosystems. *Annual review of ecology, evolution, and systematics*, 45(1), 271-295.

Crumley, R. L., Bachand, C. L., & Bennett, K. E. (2024). Snow distribution patterns revisited: A physics-based and machine learning hybrid approach to snow distribution mapping in the sub-Arctic. *Water Resources Research*, 60(9), e2023WR036180.

Foster, J. L., Robinson, D. A., Hall, D. K., & Estilow, T. W. (2008). Spring snow melt timing and changes over Arctic lands. *Polar Geography*, 31(3–4), 145–157.

Naegeli, K.; Neuhaus, C.; Salberg, A.-B.; Schwaizer, G.; Weber, H.; Wiesmann, A.; Wunderle, S.; Nagler, T. (2022): ESA Snow Climate Change Initiative (Snow_cci): Daily global Snow Cover Fraction - snow on ground (SCFG) from AVHRR (1982 - 2018), version 2.0. NERC EDS Centre for Environmental Data Analysis, 17 March 2022. <https://dx.doi.org/10.5285/3f034f4a08854eb59d58e1fa92d207b6>

Rantanen, M., Kämäräinen, M., Niittynen, P. et al. Bioclimatic atlas of the terrestrial Arctic. *Scientific Data* 10, 40 (2023). <https://doi.org/10.1038/s41597-023-01959-w>

4.4. 2) Much of the recorded warming in the Arctic occurs during the winter season. In the summer, increasing temperatures have been partly dampened by enhanced ice melt of both sea ice and glaciers. It is probably worth mentioning this effect in the discussion when considering how changes in plant diversity will develop in the future as well as why there is not as strong signal at present as might be expected when considering annual rates of warming compared to warmest quarter warming rates. I am unsure if the plant data has sufficient granularity to be able to compare sites close to melting ice and sites further away, but the existing analysis does also include different rates of warming at observation sites. I think it likely these differential warming rates are partly related to proximity to local ice sources, so I do not consider it necessary to do extra analysis on this latter point.

Thank you for this observation. As the reviewer rightly mentions, unfortunately we don't have enough granularity to identify closeness to melting ice sites, given that coordinates are only available at the subsite level (and not at the plot level which is our replication unit for most of the analyses). The direct links between sea ice loss and plant productivity are still debated, with most of the effects being indirect through climate-related variables (Post et al. 2013), but we take the reviewer's point about amelioration and have mentioned instead the projections of a wetter climate and their potential effects on plants:

“Future change in species richness and composition may not yet be detected due to extinction lags⁷⁰ and slow colonisation rates in communities of long-lived perennial species. Additionally, priority effects could cause heterogeneity in species responses to warming⁷¹. Variation in topography, microclimate and nutrient availability could mediate ecological responses and buffer against climate change impacts by providing microhabitats with suitable conditions^{21,72–74}. Rising temperatures are projected to be accompanied by increasing precipitation leading to a warmer and wetter Arctic, which could ameliorate warming-derived drought effects on plants⁵⁰. In addition, herbivory may mitigate warming-driven shrub expansion in certain regions⁵⁴. Thus, the integration of extinction lags, priority effects, local context, and both micro- and macroclimate is an essential next step to better identify the mechanisms behind Arctic plant dynamics.” (L589-600)

References:

Post, E., Bhatt, U. S., Bitz, C. M., Brodie, J. F., Fulton, T. L., Hebblewhite, M., ... & Walker, D. A. (2013). Ecological consequences of sea-ice decline. *Science*, 341(6145), 519-524.

Response to reviewers

Referee #2 (Remarks to the Author):

I would like to thank the authors for their thorough revision that addressed all my comments. We thank Reviewer #2 for their very useful feedback through the review process, it is greatly appreciated.

Referee #3 (Remarks to the Author):

Plant diversity dynamics over space and time in a warming Arctic

I think the authors have done a comprehensive job responding to the reviewer comments, and find the manuscript much improved. With a bit of effort, an interested reader should now be able to follow, and glean the rationale, for the analytical decisions and methods.

I've only minor comments and queries that the authors might consider as they prepare this work for publication. Congratulations to all involved in this important contribution to the biodiversity change literature.

We are grateful to Reviewer #3 for their thorough feedback and believe it has greatly strengthened the manuscript.

Line numbers refer to the clean (i.e., without tracked changes) version.

L272: do you really need an acronym for Research Questions?

Following previous reviews, we thought it would be useful to link the specific methods to their specific research questions, and thus we signpost the research questions in the hypothesis section, which then map on directly to the Methods section as 'RQ1, 2, 3'. Thus, we'd prefer to keep it if that is agreeable with the editor and reviewer.

L281: does this treeline idea come up again? I'm a bit confused by the exact meaning of treeline in this context, and if it comes up again, I missed it if it did. Consider removing.

Reference to the boreal forest is made previously in the Introduction in the context of potential sources of biodiversity change:

"Moreover, the borealization of Arctic ecosystems close to the treeline could further differentiate Low and High Arctic plant communities⁵¹." (L202-203)

This idea is further referenced again in the discussion:

"Richness change was not greater towards southern Arctic edges (**Figure 2a**), where we hypothesised that northward migration from the boreal forest (i.e., borealization) might be a major driver of change. Instead, this lack of latitudinal change might indicate that, where diversity is changing, one of the main sources is colonisations by species present in local species pools that have not yet been recorded in long-term monitoring plots (i.e., 'landscape' or 'dark' diversity)^{20,55}." (L386-391)

L295: I think you need something extra here for this to make sense, e.g., proportional gains of low occupancy species. Note also that dominant here needs to refer to high occupancy (as opposed to

local relative abundance or competitive superiority).

Rephrased to:

“This homogenisation could be caused by an infilling of warmer thermal niches^{34,46,47} by the same increasingly dominant, high occupancy species with higher growth rates, good dispersal and colonisation capacities⁵³, which will outweigh proportional species gains of low occupancy species.” (L252-255)

L382: would this be more accurate if you said nearly all plots experienced changes in species composition due to altered relative abundances (Bray-Curtis), and 59% showed compositional changes due to species gains and losses (Jaccard)? I’m not sure the reference to Jaccard makes sense as phrased currently.

Thank you for this suggestion. Rephrased as suggested (L293-295).

L398-400: this loss of low occupancy species is a little inconsistent with the finding of no biotic homogenisation reported below.

We believe that this might be the case because even if low-occupancy species are being lost more often, these might not be the same low-occupancy species being lost across sites, thus leading to no overall homogenisation.

L450: “associated with” is probably more appropriate here than “driven by”

Rephrased as suggested.

L481-482: I’d urge a little bit of caution regarding richness changes at larger scales here. If I understand correctly, you’ve only looked at local (indeed, quadrat) scale changes (with the exception of the spatial beta-diversity, which does not consider richness). Too many biodiversity scientists continue to assume that changes at one (smaller) scale will be consistent at another (larger) scale.

We have rephrased to:

“This result based on local scales ran counter to literature predictions¹⁹ and experimental observations of plant diversity declines at the landscape scale⁵⁵, and modelling studies predicting a regional decline of 15 to 47% in Arctic-alpine plant species richness²².” (L361-364)

L495: not sure about despite in this sentence, nor of the word choice “assemblies” - perhaps assemblages, or just delete: “...composition changed based on local context, and both climate...”. More generally, the presentation would benefit from an edit with a focus on making the writing more direct.

We have rephrased to:

“Overall, we found that Arctic plant community composition changed to different assemblages based on local context, and both climate warming and shrubification emerged as key factors influencing the magnitude of species turnover.” (L377-380)

L511-514: this sentence reads awkwardly to me, and I’m not sure of the intended meaning. Can you rephrase to more directly say what you are trying to say?

We have rephrased to:

“Given the importance of biotic interactions at species’ warm edges⁵⁷, gains could represent expansions of warm-adapted species, which could outcompete cold-adapted species^{53,58}.” (L394-396)

L520-522: some logic is needed to underpin this assertion that composition changes might eventually drive richness changes (e.g., altered biotic interactions resulting greater competitive exclusion). At the moment it reads as contradictory (and confusing for this reader at least) given the preceding discussion of your results (gains and losses balanced, consistent with equilibrium theory). Similarly, (L522-523), why might compositional changes results in further reshuffling?
Great suggestion, we have rephrased to:

“Overall, these compositional changes could result in further species reshuffling due to altered biotic interactions, and thus associated losses of rare and ecologically important species, and potential changes to ecosystem function.” (L405-408)

L560, 562: word choice: is amidst better than among here?

Sentenced rephrased to:

“Our findings demonstrate that Arctic plant richness changed in different directions (**Figure 2b, c**) amidst continued warming.” (L446-447)

Fig. 2b: Change Years to Year on the x-axis.

Corrected.

Referee #4 (Remarks to the Author):

2nd Review of Criado et al., Plant Diversity Dynamics over space and time in a warming Arctic

This is my second review of the Criado et al paper and although again I will confine my remarks to my area of expertise, namely Arctic climate, I would like to note that I found the revised paper again interesting and I think better written than the first version. As a non-biologist with a keen interest in the Arctic, I found it a very enlightening read.

The additional analyses done by the authors on the role of snow is very convincing, it is obviously disappointing that the EO datasets are not yet up to scratch for this kind of analysis and I will take this as feedback to the relevant CCI group. Nevertheless, the ARCLIM dataset is likely a good substitute and the results of this analysis seem to show fairly convincingly that the effects of climate change on snow cover and duration are still small. This may also perhaps play into the extinction lag that the authors mention elsewhere. I do also think the observation of scale effects is important and I am therefore happy that the additional paragraph on this analysis is sufficient to show that snow has been taken into account.

We thank Reviewer #4 for their suggestions regarding the climate data, and have left the additional paragraph outlining the snow analysis in the Methods section as recommended.